# Additive Decoders for Latent Variables Identification and Cartesian-Product Extrapolation

**Sébastien Lachapelle**[*,1]

**Divyat Mahajan**[*]

**Ioannis Mitliagkas**[†]

**Simon Lacoste-Julien**[†,1]

Mila & DIRO, Université de Montréal
[1]Samsung - SAIT AI Lab, Montreal

## Abstract

We tackle the problems of latent variables identification and "out-of-support" image generation in representation learning. We show that both are possible for a class of decoders that we call *additive*, which are reminiscent of decoders used for object-centric representation learning (OCRL) and well suited for images that can be decomposed as a sum of object-specific images. We provide conditions under which exactly solving the reconstruction problem using an additive decoder is guaranteed to identify the blocks of latent variables up to permutation and block-wise invertible transformations. This guarantee relies only on very weak assumptions about the distribution of the latent factors, which might present statistical dependencies and have an almost arbitrarily shaped support. Our result provides a new setting where nonlinear independent component analysis (ICA) is possible and adds to our theoretical understanding of OCRL methods. We also show theoretically that additive decoders can generate novel images by recombining observed factors of variations in novel ways, an ability we refer to as *Cartesian-product extrapolation*. We show empirically that additivity is crucial for both identifiability and extrapolation on simulated data.

## 1 Introduction

The integration of connectionist and symbolic approaches to artificial intelligence has been proposed as a solution to the lack of robustness, transferability, systematic generalization and interpretability of current deep learning algorithms [53, 4, 13, 25, 21] with justifications rooted in cognitive sciences [20, 28, 43] and causality [57, 63]. However, the problem of extracting meaningful symbols grounded in low-level observations, e.g. images, is still open. This problem is sometime referred to as *disentanglement* [4, 48] or *causal representation learning* [63]. The question of *identifiability* in representation learning, which originated in works on *nonlinear independent component analysis* (ICA) [65, 31, 33, 36], has been the focus of many recent efforts [49, 66, 26, 47, 3, 9, 41]. The mathematical results of these works provide rigorous explanations for when and why symbolic representations can be extracted from low-level observations. In a similar spirit, *Object-centric representation learning* (OCRL) aims to learn a representation in which the information about different objects are encoded separately [19, 22, 11, 24, 18, 51, 14]. These approaches have shown impressive results empirically, but the exact reason why they can perform this form of segmentation without any supervision is poorly understood.

---

[*] Equal contribution. [†] Canada CIFAR AI Chair.
Correspondence to: {`lachaseb, divyat.mahajan`}@mila.quebec

37th Conference on Neural Information Processing Systems (NeurIPS 2023).

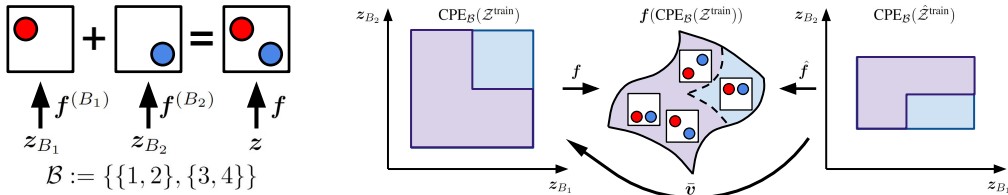

Figure 1: **Left:** Additive decoders model the additive structure of scenes composed of multiple objects. **Right:** Additive decoders allow to generate novel images never seen during training via Cartesian-product extrapolation (Corollary 3). Purple regions correspond to latents/observations seen during training. The blue regions correspond to the Cartesian-product extension. The middle set is the manifold of images of balls. In this example, the learner never saw both balls high, but these can be generated nevertheless thanks to the additive nature of the scene. Details in Section 3.2.

## 1.1 Contributions

Our first contribution is an analysis of the identifiability of a class of decoders we call *additive* (Definition 1). Essentially, a decoder $\boldsymbol{f}(\boldsymbol{z})$ acting on a latent vector $\boldsymbol{z} \in \mathbb{R}^{d_z}$ to produce an observation $\boldsymbol{x}$ is said to be additive if it can be written as $\boldsymbol{f}(\boldsymbol{z}) = \sum_{B \in \mathcal{B}} \boldsymbol{f}^{(B)}(\boldsymbol{z}_B)$ where $\mathcal{B}$ is a partition of $\{1, \ldots, d_z\}$, $\boldsymbol{f}^{(B)}(\boldsymbol{z}_B)$ are "block-specific" decoders and the $\boldsymbol{z}_B$ are non-overlapping subvectors of $\boldsymbol{z}$. This class of decoder is particularly well suited for images $\boldsymbol{x}$ that can be expressed as a sum of images corresponding to different objects (left of Figure 1). Unsurprisingly, this class of decoder bears similarity with the decoding architectures used in OCRL (Section 2), which already showed important successes at disentangling objects without any supervision. Our identifiability results provide conditions under which exactly solving the reconstruction problem with an additive decoder identifies the latent blocks $\boldsymbol{z}_B$ up to permutation and block-wise transformations (Theorems 1 & 2). We believe these results will be of interest to both the OCRL community, as they partly explain the empirical success of these approaches, and to the nonlinear ICA and disentanglement community, as it provides an important special case where identifiability holds. This result relies on the block-specific decoders being "sufficiently nonlinear" (Assumption 2) and requires only very weak assumptions on the distribution of the ground-truth latent factors of variations. In particular, these factors can be statistically dependent and their support can be (almost) arbitrary.

Our second contribution is to show theoretically that additive decoders can generate images never seen during training by recombining observed factors of variations in novel ways (Corollary 3). To describe this ability, we coin the term "Cartesian-product extrapolation" (right of Figure 1). We believe the type of identifiability analysis laid out in this work to understand "out-of-support" generation is novel and could be applied to other function classes or learning algorithms such as DALLE-2 [59] and Stable Diffusion [61] to understand their apparent creativity and hopefully improve it.

Both latent variables identification and Cartesian-product extrapolation are validated experimentally on simulated data (Section 4). More specifically, we observe that additivity is crucial for both by comparing against a non-additive decoder which fails to disentangle and extrapolate.

**Notation.** Scalars are denoted in lower-case and vectors in lower-case bold, e.g. $x \in \mathbb{R}$ and $\boldsymbol{x} \in \mathbb{R}^n$. We maintain an analogous notation for scalar-valued and vector-valued functions, e.g. $f$ and $\boldsymbol{f}$. The $i$th coordinate of the vector $\boldsymbol{x}$ is denoted by $\boldsymbol{x}_i$. The set containing the first $n$ integers excluding 0 is denoted by $[n]$. Given a subset of indices $S \subseteq [n]$, $\boldsymbol{x}_S$ denotes the subvector consisting of entries $\boldsymbol{x}_i$ for $i \in S$. Given a function $\boldsymbol{f}(\boldsymbol{x}_S) \in \mathbb{R}^m$ with input $\boldsymbol{x}_S$, the derivative of $\boldsymbol{f}$ w.r.t. $\boldsymbol{x}_i$ is denoted by $D_i \boldsymbol{f}(\boldsymbol{x}_S) \in \mathbb{R}^m$ and the second derivative w.r.t. $\boldsymbol{x}_i$ and $\boldsymbol{x}_{i'}$ is $D^2_{i,i'} \boldsymbol{f}(\boldsymbol{x}_S) \in \mathbb{R}^m$. See Table 2 in appendix for more.

**Code:** Our code repository can be found at this link.

## 2 Background & Literature review

**Identifiability of latent variable models.** The problem of latent variables identification can be best explained with a simple example. Suppose observations $\boldsymbol{x} \in \mathbb{R}^{d_x}$ are generated i.i.d. by first sampling a latent vector $\boldsymbol{z} \in \mathbb{R}^{d_z}$ from a distribution $\mathbb{P}_{\boldsymbol{z}}$ and feeding it into a decoder function $\boldsymbol{f} : \mathbb{R}^{d_z} \to \mathbb{R}^{d_x}$,

i.e. $\boldsymbol{x} = \boldsymbol{f}(\boldsymbol{z})$. By choosing an alternative model defined as $\hat{\boldsymbol{f}} := \boldsymbol{f} \circ \boldsymbol{v}$ and $\hat{\boldsymbol{z}} := \boldsymbol{v}^{-1}(\boldsymbol{z})$ where $\boldsymbol{v} : \mathbb{R}^{d_z} \to \mathbb{R}^{d_z}$ is some bijective transformation, it is easy to see that the distributions of $\hat{\boldsymbol{x}} = \hat{\boldsymbol{f}}(\hat{\boldsymbol{z}})$ and $\boldsymbol{x}$ are the same since $\hat{\boldsymbol{f}}(\hat{\boldsymbol{z}}) = \boldsymbol{f} \circ \boldsymbol{v}(\boldsymbol{v}^{-1}(\boldsymbol{z})) = \boldsymbol{f}(\boldsymbol{z})$. The problem of identifiability is that, given only the distribution over $\boldsymbol{x}$, it is impossible to distinguish between the two models $(\boldsymbol{f}, \boldsymbol{z})$ and $(\hat{\boldsymbol{f}}, \hat{\boldsymbol{z}})$. This is problematic when one wants to discover interpretable factors of variations since $\boldsymbol{z}$ and $\hat{\boldsymbol{z}}$ could be drastically different. There are essentially two strategies to go around this problem: (i) restricting the hypothesis class of decoders $\hat{\boldsymbol{f}}$ [65, 26, 44, 54, 9, 73], and/or (ii) restricting/adding structure to the distribution of $\hat{\boldsymbol{z}}$ [33, 50, 42, 47]. By doing so, the hope is that the only bijective mappings $\boldsymbol{v}$ keeping $\hat{\boldsymbol{f}}$ and $\hat{\boldsymbol{z}}$ into their respective hypothesis classes will be trivial indeterminacies such as permutations and element-wise rescalings. Our contribution, which is to restrict the decoder function $\hat{\boldsymbol{f}}$ to be additive (Definition 1), falls into the first category. Other restricted function classes for $\boldsymbol{f}$ proposed in the literature include post-nonlinear mixtures [65], local isometries [16, 15, 29], conformal and orthogonal maps [26, 60, 9] as well as various restrictions on the sparsity of $\boldsymbol{f}$ [54, 73, 7, 71]. Methods that do not restrict the decoder must instead restrict/structure the distribution of the latent factors by assuming, e.g., sparse temporal dependencies [31, 38, 42, 40], conditionally independent latent variables given an observed auxiliary variable [33, 36], that interventions targeting the latent factors are observed [42, 47, 46, 8, 2, 3, 64, 10, 67, 72, 34], or that the support of the latents is a Cartesian-product [68, 62]. In contrast, our result makes very mild assumptions about the distribution of the latent factors, which can present statistical dependencies, have an almost arbitrarily shaped support and does not require any interventions. Additionally, none of these works provide extrapolation guarantees as we do in Section 3.2.

**Relation to nonlinear ICA.** Hyvärinen and Pajunen [32] showed that the standard nonlinear ICA problem where the decoder $\boldsymbol{f}$ is nonlinear and the latent factors $\boldsymbol{z}_i$ are *statistically independent* is unidentifiable. This motivated various extensions of nonlinear ICA where more structure on the factors is assumed [30, 31, 33, 36, 37, 27]. Our approach departs from the standard nonlinear ICA problem along three axes: (i) we restrict the mixing function to be additive, (ii) the factors do not have to be necessarily independent, and (iii) we can identify only the blocks $\boldsymbol{z}_B$ as opposed to each $\boldsymbol{z}_i$ individually up to element-wise transformations, unless $\mathcal{B} = \{\{1\}, ..., \{d_z\}\}$ (see Section 3.1).

**Object-centric representation learning (OCRL).** Lin et al. [45] classified OCRL methods in two categories: *scene mixture models* [22, 23, 24, 51] & *spatial-attention models* [19, 12, 11, 18]. Additive decoders can be seen as an approximation to the decoding architectures used in the former category, which typically consist of an object-specific decoder $\boldsymbol{f}^{(\mathrm{obj})}$ acting on object-specific latent blocks $\boldsymbol{z}_B$ and "mixed" together via a masking mechanism $\boldsymbol{m}^{(B)}(\boldsymbol{z})$ which selects which pixel belongs to which object. More precisely,

$$\boldsymbol{f}(\boldsymbol{z}) = \sum_{B \in \mathcal{B}} \boldsymbol{m}^{(B)}(\boldsymbol{z}) \odot \boldsymbol{f}^{(\mathrm{obj})}(\boldsymbol{z}_B) \,, \text{ where } \boldsymbol{m}_k^{(B)}(\boldsymbol{z}) = \frac{\exp(\boldsymbol{a}_k(\boldsymbol{z}_B))}{\sum_{B' \in \mathcal{B}} \exp(\boldsymbol{a}_k(\boldsymbol{z}_{B'}))} \,, \tag{1}$$

and where $\mathcal{B}$ is a partition of $[d_z]$ made of equal-size blocks $B$ and $\boldsymbol{a} : \mathbb{R}^{|B|} \to \mathbb{R}^{d_x}$ outputs a score that is normalized via a softmax operation to obtain the masks $\boldsymbol{m}^{(B)}(\boldsymbol{z})$. Many of these works also present some mechanism to select dynamically how many objects are present in the scene and thus have a variable-size representation $\boldsymbol{z}$, an important technical aspect we omit in our analysis. Empirically, training these decoders based on some form of reconstruction objective, probabilistic or not, yields latent blocks $\boldsymbol{z}_B$ that represent the information of individual objects separately. We believe our work constitutes a step towards providing a mathematically grounded explanation for why these approaches can perform this form of disentanglement without supervision (Theorems 1 & 2). Many architectural innovations in scene mixture models concern the encoder, but our analysis focuses solely on the structure of the decoder $\boldsymbol{f}(\boldsymbol{z})$, which is a shared aspect across multiple methods. Generalization capabilities of object-centric representations were studied empirically by Dittadi et al. [14] but did not cover Cartesian-product extrapolation (Corollary 3) on which we focus here.

**Diagonal Hessian penalty [58].** Additive decoders are also closely related to the penalty introduced by Peebles et al. [58] which consists in regularizing the Hessian of the decoder to be diagonal. In Appendix A.2, we show that "additivity" and "diagonal Hessian" are equivalent properties. They showed empirically that this penalty can induce disentanglement on datasets such as CLEVR [35], which is a standard benchmark for OCRL, but did not provide any formal justification. Our work provides a rigorous explanation for these successes and highlights the link between the diagonal Hessian penalty and OCRL.

**Compositional decoders [7].** Compositional decoders were recently introduced by Brady et al. [7] as a model for OCRL methods with identifiability guarantees. A decoder $\boldsymbol{f}$ is said to be *compositional* when its Jacobian $D\boldsymbol{f}$ satisfies the following property everywhere: For all $i \in [d_z]$ and $B \in \mathcal{B}$, $D_B \boldsymbol{f}_i(\boldsymbol{z}) \neq \boldsymbol{0} \implies D_{B^c} \boldsymbol{f}_i(\boldsymbol{z}) = \boldsymbol{0}$, where $B^c := [d_z] \setminus B$. In other words, each $\boldsymbol{x}_i$ can *locally* depend solely on one block $\boldsymbol{z}_B$ (this block can change for different $\boldsymbol{z}$). In Appendix A.3, we show that compositional $C^2$ decoders are additive. Furthermore, Example 3 shows a decoder that is additive but not compositional, which means that additive $C^2$ decoders are strictly more expressive than compositional $C^2$ decoders. Another important distinction with our work is that we consider more general supports for $\boldsymbol{z}$ and provide a novel extrapolation analysis. That being said, our identifiability result does not supersede theirs since they assume only $C^1$ decoders while our theory assumes $C^2$.

**Extrapolation.** Du and Mordatch [17] studied empirically how one can combine energy-based models for what they call *compositional generalization*, which is similar to our notion of Cartesian-product extrapolation, but suppose access to datasets in which only one latent factor varies and do not provide any theory. Webb et al. [70] studied extrapolation empirically and proposed a novel benchmark which does not have an additive structure. Besserve et al. [5] proposed a theoretical framework in which out-of-distribution samples are obtained by applying a transformation to a single hidden layer inside the decoder network. Krueger et al. [39] introduced a domain generalization method which is trained to be robust to tasks falling outside the convex hull of training distributions. Extrapolation in text-conditioned image generation was recently discussed by Wang et al. [69].

## 3 Additive decoders for disentanglement & extrapolation

Our theoretical results assume the existence of some data-generating process describing how the observations $\boldsymbol{x}$ are generated and, importantly, what are the "natural" factors of variations.

**Assumption 1** (Data-generating process)**.** *The set of possible observations is given by a lower dimensional manifold $\boldsymbol{f}(\mathcal{Z}^{\text{test}})$ embedded in $\mathbb{R}^{d_x}$ where $\mathcal{Z}^{\text{test}}$ is an open set of $\mathbb{R}^{d_z}$ and $\boldsymbol{f} : \mathcal{Z}^{\text{test}} \to \mathbb{R}^{d_x}$ is a $C^2$-diffeomorphism onto its image. We will refer to $\boldsymbol{f}$ as the ground-truth decoder. At training time, the observations are i.i.d. samples given by $\boldsymbol{x} = \boldsymbol{f}(\boldsymbol{z})$ where $\boldsymbol{z}$ is distributed according to the probability measure $\mathbb{P}_{\boldsymbol{z}}^{\text{train}}$ with support $\mathcal{Z}^{\text{train}} \subseteq \mathcal{Z}^{\text{test}}$. Throughout, we assume that $\mathcal{Z}^{\text{train}}$ is regularly closed (Definition 6).*

Intuitively, the ground-truth decoder $\boldsymbol{f}$ is effectively relating the "natural factors of variations" $\boldsymbol{z}$ to the observations $\boldsymbol{x}$ in a one-to-one fashion. The map $\boldsymbol{f}$ is a $C^2$-diffeomorphism onto its image, which means that it is $C^2$ (has continuous second derivative) and that its inverse (restricted to the image of $\boldsymbol{f}$) is also $C^2$. Analogous assumptions are very common in the literature on nonlinear ICA and disentanglement [33, 36, 42, 1]. Mansouri et al. [52] pointed out that the injectivity of $\boldsymbol{f}$ is violated when images show two objects that are indistinguishable, an important practical case that is not covered by our theory.

We emphasize the distinction between $\mathcal{Z}^{\text{train}}$, which corresponds to the observations seen during training, and $\mathcal{Z}^{\text{test}}$, which corresponds to the set of all possible images. The case where $\mathcal{Z}^{\text{train}} \neq \mathcal{Z}^{\text{test}}$ will be of particular interest when discussing extrapolation in Section 3.2. The "regularly closed" condition on $\mathcal{Z}^{\text{train}}$ is mild, as it is satisfied as soon as the distribution of $\boldsymbol{z}$ has a density w.r.t. the Lebesgue measure on $\mathbb{R}^{d_z}$. It is violated, for example, when $\boldsymbol{z}$ is a discrete random vector. Figure 2 illustrates this assumption with simple examples.

**Objective.** Our analysis is based on the simple objective of reconstructing the observations $\boldsymbol{x}$ by learning an encoder $\hat{\boldsymbol{g}} : \mathbb{R}^{d_x} \to \mathbb{R}^{d_z}$ and a decoder $\hat{\boldsymbol{f}} : \mathbb{R}^{d_z} \to \mathbb{R}^{d_x}$. Note that we assumed implicitly that the dimensionality of the learned representation matches the dimensionality of the ground-truth. We define the set of latent codes the encoder can output when evaluated on the training distribution:

$$\hat{\mathcal{Z}}^{\text{train}} := \hat{\boldsymbol{g}}(\boldsymbol{f}(\mathcal{Z}^{\text{train}})) \,. \tag{2}$$

When the images of the ground-truth and learned decoders match, i.e. $\boldsymbol{f}(\mathcal{Z}^{\text{train}}) = \hat{\boldsymbol{f}}(\hat{\mathcal{Z}}^{\text{train}})$, which happens when the reconstruction task is solved exactly, one can define the map $\boldsymbol{v} : \hat{\mathcal{Z}}^{\text{train}} \to \mathcal{Z}^{\text{train}}$ as

$$\boldsymbol{v} := \boldsymbol{f}^{-1} \circ \hat{\boldsymbol{f}} \,. \tag{3}$$

This function is going to be crucial throughout the work, especially to define $\mathcal{B}$-disentanglement (Definition 3), as it relates the learned representation to the ground-truth representation.

Before introducing our formal definition of additive decoders, we introduce the following notation: Given a set $\mathcal{Z} \subseteq \mathbb{R}^{d_z}$ and a subset of indices $B \subseteq [d_z]$, let us define $\mathcal{Z}_B$ to be the projection of $\mathcal{Z}$ onto dimensions labelled by the index set $B$. More formally,

$$\mathcal{Z}_B := \{\boldsymbol{z}_B \mid \boldsymbol{z} \in \mathcal{Z}\} \subseteq \mathbb{R}^{|B|}. \tag{4}$$

Intuitively, we will say that a decoder is *additive* when its output is the summation of the outputs of "object-specific" decoders that depend only on each latent block $\boldsymbol{z}_B$. This captures the idea that an image can be seen as the juxtaposition of multiple images which individually correspond to objects in the scene or natural factors of variations (left of Figure 1).

**Definition 1** (Additive functions). *Let $\mathcal{B}$ be a partition of $[d_z]$[1]. A function $\boldsymbol{f} : \mathcal{Z} \to \mathbb{R}^{d_x}$ is said to be **additive** if there exist functions $\boldsymbol{f}^{(B)} : \mathcal{Z}_B \to \mathbb{R}^{d_x}$ for all $B \in \mathcal{B}$ such that*

$$\forall \boldsymbol{z} \in \mathcal{Z}, \boldsymbol{f}(\boldsymbol{z}) = \sum_{B \in \mathcal{B}} \boldsymbol{f}^{(B)}(\boldsymbol{z}_B). \tag{5}$$

This additivity property will be central to our analysis as it will be the driving force of identifiability (Theorem 1 & 2) and Cartesian-product extrapolation (Corollary 3).

**Remark 1.** *Suppose we have $\boldsymbol{x} = \sigma(\sum_{B \in \mathcal{B}} \boldsymbol{f}^{(B)}(\boldsymbol{z}_B))$ where $\sigma$ is a known bijective function. For example, if $\sigma(\boldsymbol{y}) := \exp(\boldsymbol{y})$ (component-wise), the decoder can be thought of as being multiplicative. Our results still apply since we can simply transform the data doing $\tilde{\boldsymbol{x}} := \sigma^{-1}(\boldsymbol{x})$ to recover the additive form $\tilde{\boldsymbol{x}} = \sum_{B \in \mathcal{B}} \boldsymbol{f}^{(B)}(\boldsymbol{z}_B)$.*

**Differences with OCRL in practice.** We point out that, although the additive decoders make intuitive sense for OCRL, they are not expressive enough to represent the "masked decoders" typically used in practice (Equation (1)). The lack of additivity stems from the normalization in the masks $\boldsymbol{m}^{(B)}(\boldsymbol{z})$. We hypothesize that studying the simpler additive decoders might still reveal interesting phenomena present in modern OCRL approaches due to their resemblance. Another difference is that, in practice, the same object-specific decoder $\boldsymbol{f}^{(\text{obj})}$ is applied to every latent block $\boldsymbol{z}_B$. Our theory allows for these functions to be different, but also applies when functions are the same. Additionally, this parameter sharing across $\boldsymbol{f}^{(B)}$ enables modern methods to have a variable number of objects across samples, an important practical point our theory does not cover.

### 3.1 Identifiability analysis

We now study the identifiability of additive decoders and show how they can yield disentanglement. Our definition of disentanglement will rely on *partition-respecting permutations*:

**Definition 2** (Partition-respecting permutations). *Let $\mathcal{B}$ be a partition of $\{1, ..., d_z\}$. A permutation $\pi$ over $\{1, ..., d_z\}$ respects $\mathcal{B}$ if, for all $B \in \mathcal{B}$, $\pi(B) \in \mathcal{B}$.*

Essentially, a permutation that respects $\mathcal{B}$ is one which can permute blocks of $\mathcal{B}$ and permute elements within a block, but cannot "mix" blocks together. We now introduce $\mathcal{B}$-disentanglement.

**Definition 3** ($\mathcal{B}$-disentanglement). *A learned decoder $\hat{\boldsymbol{f}} : \mathbb{R}^{d_z} \to \mathbb{R}^{d_x}$ is said to be $\mathcal{B}$-**disentangled** w.r.t. the ground-truth decoder $\boldsymbol{f}$ when $\boldsymbol{f}(\mathcal{Z}^{\text{train}}) = \hat{\boldsymbol{f}}(\hat{\mathcal{Z}}^{\text{train}})$ and the mapping $\boldsymbol{v} := \boldsymbol{f}^{-1} \circ \hat{\boldsymbol{f}}$ is a diffeomorphism from $\hat{\mathcal{Z}}^{\text{train}}$ to $\mathcal{Z}^{\text{train}}$ satisfying the following property: there exists a permutation $\pi$ respecting $\mathcal{B}$ such that, for all $B \in \mathcal{B}$, there exists a function $\bar{\boldsymbol{v}}_{\pi(B)} : \hat{\mathcal{Z}}_B^{\text{train}} \to \mathcal{Z}_{\pi(B)}^{\text{train}}$ such that, for all $\boldsymbol{z} \in \hat{\mathcal{Z}}^{\text{train}}$, $\boldsymbol{v}_{\pi(B)}(\boldsymbol{z}) = \bar{\boldsymbol{v}}_{\pi(B)}(\boldsymbol{z}_B)$. In other words, $\boldsymbol{v}_{\pi(B)}(\boldsymbol{z})$ depends only on $\boldsymbol{z}_B$.*

Thus, $\mathcal{B}$-disentanglement means that the blocks of latent dimensions $\boldsymbol{z}_B$ are disentangled from one another, but that variables within a given block might remain entangled. Note that, unless the partition is $\mathcal{B} = \{\{1\}, \ldots, \{d_z\}\}$, this corresponds to a weaker form of disentanglement than what is typically sought in nonlinear ICA, i.e. recovering each variable individually.

**Example 1.** *To illustrate $\mathcal{B}$-disentanglement, imagine a scene consisting of two balls moving around in 2D where the "ground-truth" representation is given by $\boldsymbol{z} = (x^1, y^1, x^2, y^2)$ where $\boldsymbol{z}_{B_1} = (x^1, y^1)$ and $\boldsymbol{z}_{B_2} = (x^2, y^2)$ are the coordinates of each ball (here, $\mathcal{B} := \{\{1, 2\}, \{3, 4\}\}$). In that case, a learned representation is $\mathcal{B}$-disentangled when the balls are disentangled from one another. However, the basis in which the position of each ball is represented might differ in both representations.*

---

[1]Without loss of generality, we assume that the partition $\mathcal{B}$ is contiguous, i.e. each $B \in \mathcal{B}$ can be written as $B = \{i + 1, i + 2, \ldots, i + |B|\}$.

Our first result (Theorem 1) shows a weaker form of disentanglement we call *local $\mathcal{B}$-disentanglement*. This means the Jacobian matrix of $\boldsymbol{v}$, $D\boldsymbol{v}$, has a "block-permutation" structure everywhere.

**Definition 4** (Local $\mathcal{B}$-disentanglement). *A learned decoder $\hat{\boldsymbol{f}} : \mathbb{R}^{d_z} \to \mathbb{R}^{d_x}$ is said to be **locally $\mathcal{B}$-disentangled** w.r.t. the ground-truth decoder $\boldsymbol{f}$ when $\boldsymbol{f}(\mathcal{Z}^{\text{train}}) = \hat{\boldsymbol{f}}(\hat{\mathcal{Z}}^{\text{train}})$ and the mapping $\boldsymbol{v} := \boldsymbol{f}^{-1} \circ \hat{\boldsymbol{f}}$ is a diffeomorphism from $\hat{\mathcal{Z}}^{\text{train}}$ to $\mathcal{Z}^{\text{train}}$ with a mapping $\boldsymbol{v} : \hat{\mathcal{Z}}^{\text{train}} \to \mathcal{Z}^{\text{train}}$ satisfying the following property: for all $\boldsymbol{z} \in \hat{\mathcal{Z}}^{\text{train}}$, there exists a permutation $\pi$ respecting $\mathcal{B}$ such that, for all $B \in \mathcal{B}$, the columns of $D\boldsymbol{v}_{\pi(B)}(\boldsymbol{z}) \in \mathbb{R}^{|B| \times d_z}$ outside block $B$ are zero.*

In Appendix A.4, we provide three examples where local disentanglement holds but not global disentanglement. The first one illustrates how having a disconnected support can allow for a permutation $\pi$ (from Definition 4) that changes between disconnected regions of the support. The last two examples show how, even if the permutation stays the same throughout the support, we can still violate global disentanglement, even with a connected support.

We now state the main identifiability result of this work which provides conditions to guarantee *local* disentanglement. We will then see how to go from local to *global* disentanglement in the subsequent Theorem 2. For pedagogical reasons, we delay the formalization of the sufficient nonlinearity Assumption 2 on which the result crucially relies.

**Theorem 1** (Local disentanglement via additive decoders). *Suppose that the data-generating process satisfies Assumption 1, that the learned decoder $\hat{\boldsymbol{f}} : \mathbb{R}^{d_z} \to \mathbb{R}^{d_x}$ is a $C^2$-diffeomorphism, that the encoder $\hat{\boldsymbol{g}} : \mathbb{R}^{d_x} \to \mathbb{R}^{d_z}$ is continuous, that both $\boldsymbol{f}$ and $\hat{\boldsymbol{f}}$ are additive (Definition 1) and that $\boldsymbol{f}$ is sufficiently nonlinear as formalized by Assumption 2. Then, if $\hat{\boldsymbol{f}}$ and $\hat{\boldsymbol{g}}$ solve the reconstruction problem on the training distribution, i.e. $\mathbb{E}^{\text{train}}||\boldsymbol{x} - \hat{\boldsymbol{f}}(\hat{\boldsymbol{g}}(\boldsymbol{x}))||^2 = 0$, we have that $\hat{\boldsymbol{f}}$ is locally $\mathcal{B}$-disentangled w.r.t. $\boldsymbol{f}$ (Definition 4) .*

The proof of Theorem 1, which can be found in Appendix A.5, is inspired from Hyvärinen et al. [33]. The essential differences are that (i) they leverage the additivity of the conditional log-density of $\boldsymbol{z}$ given an auxiliary variable $\boldsymbol{u}$ (i.e. conditional independence) instead of the additivity of the decoder function $\boldsymbol{f}$, (ii) we extend their proof techniques to allow for "block" disentanglement, i.e. when $\mathcal{B}$ is not the trivial partition $\{\{1\}, \ldots, \{d_z\}\}$, (iii) the asssumption "sufficient variability" of the prior $p(\boldsymbol{z} \mid \boldsymbol{u})$ of Hyvärinen et al. [33] is replaced by an analogous assumption of "sufficient nonlinearity" of the decoder $\boldsymbol{f}$ (Assumption 2), and (iv) we consider much more general supports $\mathcal{Z}^{\text{train}}$ which makes the jump from local to global disentanglement less direct in our case.

**The identifiability-expressivity trade-off.** The level of granularity of the partition $\mathcal{B}$ controls the trade-off between identifiability and expressivity: the finer the partition, the tighter the identifiability guarantee but the less expressive is the function class. The optimal level of granularity is going to dependent on the application at hand. Whether $\mathcal{B}$ could be learned from data is left for future work.

**Sufficient nonlinearity.** The following assumption is key in proving Theorem 2, as it requires that the ground-truth decoder is "sufficiently nonlinear". This is reminiscent of the "sufficient variability" assumptions found in the nonlinear ICA litterature, which usually concerns the distribution of the latent variable $\boldsymbol{z}$ as opposed to the decoder $\boldsymbol{f}$ [30, 31, 33, 36, 37, 42, 73]. We clarify this link in Appendix A.6 and provide intuitions why sufficient nonlinearity can be satisfied when $d_x \gg d_z$.

**Assumption 2** (Sufficient nonlinearity of $\boldsymbol{f}$). *Let $q := d_z + \sum_{B \in \mathcal{B}} \frac{|B|(|B|+1)}{2}$. For all $\boldsymbol{z} \in \mathcal{Z}^{\text{train}}$, $\boldsymbol{f}$ is such that the following matrix has linearly independent columns (i.e. full column-rank):*

$$\boldsymbol{W}(\boldsymbol{z}) := \left[ \left[ D_i \boldsymbol{f}^{(B)}(\boldsymbol{z}_B) \right]_{i \in B} \left[ D^2_{i,i'} \boldsymbol{f}^{(B)}(\boldsymbol{z}_B) \right]_{(i,i') \in B^2_{\le}} \right]_{B \in \mathcal{B}} \in \mathbb{R}^{d_x \times q}, \qquad (6)$$

*where $B^2_{\le} := B^2 \cap \{(i, i') \mid i' \le i\}$. Note this implies $d_x \ge q$.*

The following example shows that Theorem 1 does not apply if the ground-truth decoder $\boldsymbol{f}$ is linear. If that was the case, it would contradict the well known fact that linear ICA with independent Gaussian factors is unidentifiable.

**Example 2** (Importance of Assumption 2). *Suppose $\boldsymbol{x} = \boldsymbol{f}(\boldsymbol{z}) = \boldsymbol{A}\boldsymbol{z}$ where $\boldsymbol{A} \in \mathbb{R}^{d_x \times d_z}$ is full rank. Take $\hat{\boldsymbol{f}}(\boldsymbol{z}) := \boldsymbol{A}\boldsymbol{V}\boldsymbol{z}$ and $\hat{\boldsymbol{g}}(\boldsymbol{x}) := \boldsymbol{V}^{-1}\boldsymbol{A}^{\dagger}\boldsymbol{x}$ where $\boldsymbol{V} \in \mathbb{R}^{d_z \times d_z}$ is invertible and $\boldsymbol{A}^{\dagger}$ is the left pseudo inverse of $\boldsymbol{A}$. By construction, we have that $\mathbb{E}[\boldsymbol{x} - \hat{\boldsymbol{f}}(\hat{\boldsymbol{g}}(\boldsymbol{x}))] = 0$ and $\boldsymbol{f}$ and $\hat{\boldsymbol{f}}$ are*

$\mathcal{B}$-additive because $\boldsymbol{f}(\boldsymbol{z}) = \sum_{B\in\mathcal{B}} \boldsymbol{A}_{\cdot,B}\boldsymbol{z}_B$ and $\hat{\boldsymbol{f}}(\boldsymbol{z}) = \sum_{B\in\mathcal{B}}(\boldsymbol{AV})_{\cdot,B}\boldsymbol{z}_B$. *However, we still have that* $\boldsymbol{v}(\boldsymbol{z}) := \boldsymbol{f}^{-1} \circ \hat{\boldsymbol{f}}(\boldsymbol{z}) = \boldsymbol{V}\boldsymbol{z}$ *where* $\boldsymbol{V}$ *does not necessarily have a block-permutation structure, i.e. no disentanglement. The reason we cannot apply Theorem 1 here is because Assumption 2 is not satisfied. Indeed, the second derivatives of* $\boldsymbol{f}^{(B)}(\boldsymbol{z}_B) := \boldsymbol{A}_{\cdot,B}\boldsymbol{z}_B$ *are all zero and hence* $\boldsymbol{W}(\boldsymbol{z})$ *cannot have full column-rank.*

**Example 3** (A sufficiently nonlinear $\boldsymbol{f}$). *In Appendix A.7 we show numerically that the function*

$$\boldsymbol{f}(\boldsymbol{z}) := [\boldsymbol{z}_1, \boldsymbol{z}_1^2, \boldsymbol{z}_1^3, \boldsymbol{z}_1^4]^\top + [(\boldsymbol{z}_2 + 1), (\boldsymbol{z}_2 + 1)^2, (\boldsymbol{z}_2 + 1)^3, (\boldsymbol{z}_2 + 1)^4]^\top \tag{7}$$

*is a diffeomorphism from the square* $[-1, 0] \times [0, 1]$ *to its image that satisfies Assumption 2.*

**Example 4** (Smooth balls dataset is sufficiently nonlinear). *In Appendix A.7 we present a simple synthetic dataset consisting of images of two colored balls moving up and down. We also verify numerically that its underlying ground-truth decoder* $\boldsymbol{f}$ *is sufficiently nonlinear.*

### 3.1.1 From local to global disentanglement

The following result provides additional assumptions to guarantee *global* disentanglement (Definition 3) as opposed to only local disentanglement (Definition 4). See Appendix A.8 for its proof.

**Theorem 2** (From local to global disentanglement). *Suppose that all the assumptions of Theorem 1 hold. Additionally, assume* $\mathcal{Z}^{\mathrm{train}}$ *is path-connected (Definition 8) and that the block-specific decoders* $\boldsymbol{f}^{(B)}$ *and* $\hat{\boldsymbol{f}}^{(B)}$ *are injective for all blocks* $B \in \mathcal{B}$. *Then, if* $\hat{\boldsymbol{f}}$ *and* $\hat{\boldsymbol{g}}$ *solve the reconstruction problem on the training distribution, i.e.* $\mathbb{E}^{\mathrm{train}}||\boldsymbol{x} - \hat{\boldsymbol{f}}(\hat{\boldsymbol{g}}(\boldsymbol{x}))||^2 = 0$, *we have that* $\hat{\boldsymbol{f}}$ *is (globally)* $\mathcal{B}$-*disentangled w.r.t.* $\boldsymbol{f}$ *(Definition 3) and, for all* $B \in \mathcal{B}$,

$$\hat{\boldsymbol{f}}^{(B)}(\boldsymbol{z}_B) = \boldsymbol{f}^{(\pi(B))}(\bar{\boldsymbol{v}}_{\pi(B)}(\boldsymbol{z}_B)) + \boldsymbol{c}^{(B)}, \text{ for all } \boldsymbol{z}_B \in \hat{\mathcal{Z}}_B^{\mathrm{train}}, \tag{8}$$

*where the functions* $\bar{\boldsymbol{v}}_{\pi(B)}$ *are from Defintion 3 and the vectors* $\boldsymbol{c}^{(B)} \in \mathbb{R}^{d_x}$ *are constants such that* $\sum_{B\in\mathcal{B}} \boldsymbol{c}^{(B)} = 0$. *We also have that the functions* $\bar{\boldsymbol{v}}_{\pi(B)} : \hat{\mathcal{Z}}_B^{\mathrm{train}} \to \mathcal{Z}_{\pi(B)}^{\mathrm{train}}$ *are* $C^2$-*diffeomorphisms and have the following form:*

$$\bar{\boldsymbol{v}}_{\pi(B)}(\boldsymbol{z}_B) = (\boldsymbol{f}^{\pi(B)})^{-1}(\hat{\boldsymbol{f}}^{(B)}(\boldsymbol{z}_B) - \boldsymbol{c}^{(B)}), \text{ for all } \boldsymbol{z}_B \in \hat{\mathcal{Z}}_B^{\mathrm{train}}. \tag{9}$$

Equation (8) in the above result shows that each block-specific learned decoder $\hat{\boldsymbol{f}}^{(B)}$ is "imitating" a block-specific ground-truth decoder $\boldsymbol{f}^{\pi(B)}$. Indeed, the "object-specific" image outputted by the decoder $\hat{\boldsymbol{f}}^{(B)}$ evaluated at some $\boldsymbol{z}_B \in \hat{\mathcal{Z}}_B^{\mathrm{train}}$ is the same as the image outputted by $\boldsymbol{f}^{(B)}$ evaluated at $\boldsymbol{v}(\boldsymbol{z}_B) \in \mathcal{Z}_B^{\mathrm{train}}$, *up to an additive constant vector* $\boldsymbol{c}^{(B)}$. These constants cancel each other out when taking the sum of the block-specific decoders.

Equation (9) provides an explicit form for the function $\bar{\boldsymbol{v}}_{\pi(B)}$, which is essentially the learned block-specific decoder composed with the inverse of the ground-truth block-specific decoder.

**Additional assumptions to go from local to global.** Assuming that the support of $\mathbb{P}_{\boldsymbol{z}}^{\mathrm{train}}$, $\mathcal{Z}^{\mathrm{train}}$, is **path-connected** (see Definition 8 in appendix) is useful since it prevents the permutation $\pi$ of Definition 4 from changing between two disconnected regions of $\hat{\mathcal{Z}}^{\mathrm{train}}$. See Figure 2 for an illustration. In Appendix A.9, we discuss the additional assumption that each $\boldsymbol{f}^{(B)}$ must be injective and show that, in general, it is not equivalent to the assumption that $\sum_{B\in\mathcal{B}} \boldsymbol{f}^{(B)}$ is injective.

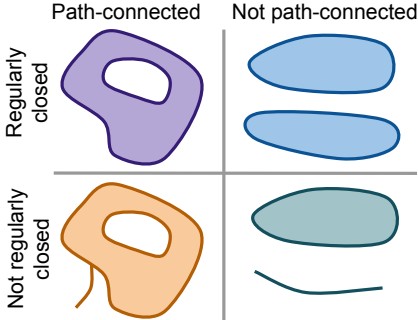

Figure 2: Illustrating regularly closed sets (Definition 6) and path-connected sets (Definition 8). Theorem 2 requires $\mathcal{Z}^{\mathrm{train}}$ to satisfy both properties.

### 3.2 Cartesian-product extrapolation

In this section, we show how a learned additive decoder can be used to generate images $\boldsymbol{x}$ that are "out of support" in the sense that $\boldsymbol{x} \notin \boldsymbol{f}(\mathcal{Z}^{\mathrm{train}})$, but that are still on the manifold of "reasonable" images, i.e. $\boldsymbol{x} \in \boldsymbol{f}(\mathcal{Z}^{\mathrm{test}})$. To characterize the set of images the learned decoder can generate, we will rely on the notion of "cartesian-product extension", which we define next.

**Definition 5** (Cartesian-product extension). *Given a set $\mathcal{Z} \subseteq \mathbb{R}^{d_z}$ and partition $\mathcal{B}$ of $[d_z]$, we define the Cartesian-product extension of $\mathcal{Z}$ as*

$$\mathrm{CPE}_{\mathcal{B}}(\mathcal{Z}) := \prod_{B \in \mathcal{B}} \mathcal{Z}_B \,, where\, \mathcal{Z}_B := \{\boldsymbol{z}_B \mid \boldsymbol{z} \in \mathcal{Z}\}.$$

*It is indeed an extension of $\mathcal{Z}$ since $\mathcal{Z} \subseteq \prod_{B \in \mathcal{B}} \mathcal{Z}_B$.*

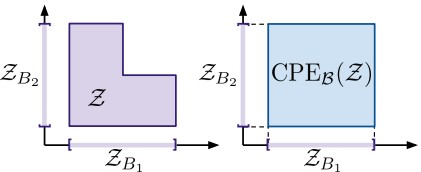

Figure 3: Illustration of Definition 5.

Let us define $\bar{\boldsymbol{v}} : \mathrm{CPE}_{\mathcal{B}}(\hat{\mathcal{Z}}^{\mathrm{train}}) \to \mathrm{CPE}_{\mathcal{B}}(\mathcal{Z}^{\mathrm{train}})$ to be the natural extension of the function $\boldsymbol{v} : \hat{\mathcal{Z}}^{\mathrm{train}} \to \mathcal{Z}^{\mathrm{train}}$. More explicitly, $\bar{\boldsymbol{v}}$ is the "concatenation" of the functions $\bar{\boldsymbol{v}}_B$ given in Definition 3:

$$\bar{\boldsymbol{v}}(\boldsymbol{z})^\top := [\bar{\boldsymbol{v}}_{B_1}(\boldsymbol{z}_{\pi^{-1}(B_1)})^\top \cdots \bar{\boldsymbol{v}}_{B_\ell}(\boldsymbol{z}_{\pi^{-1}(B_\ell)})^\top] \,, \tag{10}$$

where $\ell$ is the number of blocks in $\mathcal{B}$. This map is a diffeomorphism because each $\bar{\boldsymbol{v}}_{\pi(B)}$ is a diffeomorphism from $\hat{\mathcal{Z}}_B^{\mathrm{train}}$ to $\mathcal{Z}_{\pi(B)}^{\mathrm{train}}$ by Theorem 2.

We already know that $\hat{\boldsymbol{f}}(\boldsymbol{z}) = \boldsymbol{f} \circ \bar{\boldsymbol{v}}(\boldsymbol{z})$ for all $\boldsymbol{z} \in \hat{\mathcal{Z}}^{\mathrm{train}}$. The following result shows that this equality holds in fact on the larger set $\mathrm{CPE}_{\mathcal{B}}(\hat{\mathcal{Z}}^{\mathrm{train}})$, the Cartesian-product extension of $\hat{\mathcal{Z}}^{\mathrm{train}}$. See right of Figure 1 for an illustration of the following corollary.

**Corollary 3** (Cartesian-product extrapolation). *Suppose the assumptions of Theorem 2 holds. Then,*

$$for\ all\ \boldsymbol{z} \in \mathrm{CPE}_{\mathcal{B}}(\hat{\mathcal{Z}}^{\mathrm{train}}), \ \sum_{B \in \mathcal{B}} \hat{\boldsymbol{f}}^{(B)}(\boldsymbol{z}_B) = \sum_{B \in \mathcal{B}} \boldsymbol{f}^{(\pi(B))}(\bar{\boldsymbol{v}}_{\pi(B)}(\boldsymbol{z}_B)) \,. \tag{11}$$

*Furthermore, if $\mathrm{CPE}_{\mathcal{B}}(\mathcal{Z}^{\mathrm{train}}) \subseteq \mathcal{Z}^{\mathrm{test}}$, then $\hat{\boldsymbol{f}}(\mathrm{CPE}_{\mathcal{B}}(\hat{\mathcal{Z}}^{\mathrm{train}})) \subseteq \boldsymbol{f}(\mathcal{Z}^{\mathrm{test}})$.*

Equation (11) tells us that the learned decoder $\hat{\boldsymbol{f}}$ "imitates" the ground-truth $\boldsymbol{f}$ not just over $\hat{\mathcal{Z}}^{\mathrm{train}}$, but also over its Cartesian-product extension. This is important since it guarantees that we can generate observations never seen during training as follows: Choose a latent vector $\boldsymbol{z}^{\mathrm{new}}$ that is in the Cartesian-product extension of $\hat{\mathcal{Z}}^{\mathrm{train}}$, but not in $\hat{\mathcal{Z}}^{\mathrm{train}}$ itself, i.e. $\boldsymbol{z}^{\mathrm{new}} \in \mathrm{CPE}_{\mathcal{B}}(\hat{\mathcal{Z}}^{\mathrm{train}}) \setminus \hat{\mathcal{Z}}^{\mathrm{train}}$. Then, evaluate the learned decoder on $\boldsymbol{z}^{\mathrm{new}}$ to get $\boldsymbol{x}^{\mathrm{new}} := \hat{\boldsymbol{f}}(\boldsymbol{z}^{\mathrm{new}})$. By Corollary 3, we know that $\boldsymbol{x}^{\mathrm{new}} = \boldsymbol{f} \circ \bar{\boldsymbol{v}}(\boldsymbol{z}^{\mathrm{new}})$, i.e. it is the observation one would have obtain by evaluating the ground-truth decoder $\boldsymbol{f}$ on the point $\bar{\boldsymbol{v}}(\boldsymbol{z}^{\mathrm{new}}) \in \mathrm{CPE}_{\mathcal{B}}(\mathcal{Z}^{\mathrm{train}})$. In addition, this $\boldsymbol{x}^{\mathrm{new}}$ has never been seen during training since $\bar{\boldsymbol{v}}(\boldsymbol{z}^{\mathrm{new}}) \notin \bar{\boldsymbol{v}}(\hat{\mathcal{Z}}^{\mathrm{train}}) = \mathcal{Z}^{\mathrm{train}}$. The experiment of Figure 4 illustrates this procedure.

**About the extra assumption "$\mathrm{CPE}_{\mathcal{B}}(\mathcal{Z}^{\mathrm{train}}) \subseteq \mathcal{Z}^{\mathrm{test}}$".** Recall that, in Assumption 1, we interpreted $\boldsymbol{f}(\mathcal{Z}^{\mathrm{test}})$ to be the set of "reasonable" observations $\boldsymbol{x}$, of which we only observe a subset $\boldsymbol{f}(\mathcal{Z}^{\mathrm{train}})$. Under this interpretation, $\mathcal{Z}^{\mathrm{test}}$ is the set of reasonable values for the vector $\boldsymbol{z}$ and the additional assumption that $\mathrm{CPE}_{\mathcal{B}}(\mathcal{Z}^{\mathrm{train}}) \subseteq \mathcal{Z}^{\mathrm{test}}$ in Corollary 3 requires that the Cartesian-product extension of $\mathcal{Z}^{\mathrm{train}}$ consists only of reasonable values of $\boldsymbol{z}$. From this assumption, we can easily conclude that $\hat{\boldsymbol{f}}(\mathrm{CPE}_{\mathcal{B}}(\hat{\mathcal{Z}}^{\mathrm{train}})) \subseteq \boldsymbol{f}(\mathcal{Z}^{\mathrm{test}})$, which can be interpreted as: "The novel observations $\boldsymbol{x}^{\mathrm{new}}$ obtained via Cartesian-product extrapolation are *reasonable*". Appendix A.11 describes an example where the assumption is violated, i.e. $\mathrm{CPE}_{\mathcal{B}}(\mathcal{Z}^{\mathrm{train}}) \nsubseteq \mathcal{Z}^{\mathrm{test}}$. The practical implication of this is that the new observations $\boldsymbol{x}^{\mathrm{new}}$ obtained via Cartesian-product extrapolation might not always be reasonable.

**Disentanglement is not enough for extrapolation.** To the best of our knowledge, Corollary 3 is the first result that formalizes how disentanglement can induce extrapolation. We believe it illustrates the fact that disentanglement alone is not sufficient to enable extrapolation and that one needs to restrict the hypothesis class of decoders in some way. Indeed, given a learned decoder $\hat{\boldsymbol{f}}$ that is disentangled w.r.t. $\boldsymbol{f}$ on the training support $\mathcal{Z}^{\mathrm{train}}$, one cannot guarantee both decoders will "agree" outside the training domain without further restricting $\hat{\boldsymbol{f}}$ and $\boldsymbol{f}$. This work has focused on "additivity", but we believe other types of restriction could correspond to other types of extrapolation.

## 4   Experiments

We now present empirical validations of the theoretical results presented earlier. To achieve this, we compare the ability of additive and non-additive decoders to both identify ground-truth latent factors (Theorems 1 & 2) and extrapolate (Corollary 3) when trained to solve the reconstruction task on simple images ($64 \times 64 \times 3$) consisting of two balls moving in space [2]. See Appendix B.1

| | ScalarLatents | | | | BlockLatents (independent $z$) | | BlockLatents (dependent $z$) | |
|---|---|---|---|---|---|---|---|---|
| Decoders | RMSE | $\text{LMS}_{\text{Spear}}$ | $\text{RMSE}^{\text{OOS}}$ | $\text{LMS}^{\text{OOS}}_{\text{Spear}}$ | RMSE | $\text{LMS}_{\text{Tree}}$ | RMSE | $\text{LMS}_{\text{Tree}}$ |
| Non-add. | .06 $\pm$.002 | 70.6$\pm$5.21 | .18$\pm$.012 | 73.7$\pm$4.64 | .02$\pm$.001 | 53.9$\pm$7.58 | .02$\pm$.001 | 78.1$\pm$2.92 |
| Additive | .06$\pm$.002 | **91.5$\pm$3.57** | .11$\pm$.018 | **89.5$\pm$5.02** | .03$\pm$.012 | **92.2$\pm$4.91** | .01$\pm$.002 | **99.9$\pm$0.02** |

Table 1: Reporting reconstruction mean squared error (RMSE ↓) and the Latent Matching Score (LMS ↑) for the three datasets considered: **ScalarLatents** and **BlockLatents** with independent and dependent latents. Runs were repeated with 10 random initializations. $\text{RMSE}^{\text{OOS}}$ and $\text{LMS}^{\text{OOS}}_{\text{Spear}}$ are the same metric but evaluated out of support (see Appendix B.3 for details). While the standard error is high, the differences are still clear as can be seen in their box plot version in Appendix B.4.

for training details. We consider two datasets: one where the two ball positions can only vary along the $y$-axis (**ScalarLatents**) and one where the positions can vary along both the $x$ and $y$ axes (**BlockLatents**).

**ScalarLatents:** The ground-truth latent vector $z \in \mathbb{R}^2$ is such that $z_1$ and $z_2$ corresponds to the height (y-coordinate) of the first and second ball, respectively. Thus the partition is simply $\mathcal{B} = \{\{1\}, \{2\}\}$ (each object has only one latent factor). This simple setting is interesting to study since the low dimensionality of the latent space ($d_z = 2$) allows for exhaustive visualizations like Figure 4. To study Cartesian-product extrapolation (Corollary 3), we sample $z$ from a distribution with a L-shaped support given by $\mathcal{Z}^{\text{train}} := [0, 1] \times [0, 1] \setminus [0.5, 1] \times [0.5, 1]$, so that the training set does not contain images where both balls appear in the upper half of the image (see Appendix B.2).

**BlockLatents:** The ground-truth latent vector $z \in \mathbb{R}^4$ is such that $z_{\{1,2\}}$ and $z_{\{3,4\}}$ correspond to the $x, y$ position of the first and second ball, respectively (the partition is simply $\mathcal{B} = \{\{1, 2\}, \{3, 4\}\}$, i.e. each object has two latent factors). Thus, this more challenging setting illustrates "block-disentanglement". The latent $z$ is sampled uniformly from the hypercube $[0, 1]^4$ but the images presenting occlusion (when a ball is behind another) are rejected from the dataset. We discuss how additive decoders cannot model images presenting occlusion in Appendix A.12. We also present an additional version of this dataset where we sample from the hypercube $[0, 1]^4$ with dependencies. See Appendix B.2 for more details about data generation.

**Evaluation metrics:** To evaluate disentanglement, we compute a matrix of scores $(s_{B,B'}) \in \mathbb{R}^{\ell \times \ell}$ where $\ell$ is the number of blocks in $\mathcal{B}$ and $s_{B,B'}$ is a score measuring how well we can predict the ground-truth block $z_B$ from the learned latent block $\hat{z}_{B'} = \hat{g}_{B'}(x)$ outputted by the encoder. The final Latent Matching Score (LMS) is computed as $\text{LMS} = \arg\max_{\pi \in \mathfrak{S}_{\mathcal{B}}} \frac{1}{\ell} \sum_{B \in \mathcal{B}} s_{B, \pi(B)}$, where $\mathfrak{S}_{\mathcal{B}}$ is the set of permutations respecting $\mathcal{B}$ (Definition 2). When $\mathcal{B} := \{\{1\}, \ldots, \{d_z\}\}$ and the score used is the absolute value of the correlation, LMS is simply the *mean correlation coefficient* (MCC), which is widely used in the nonlinear ICA literature [30, 31, 33, 36, 42]. Because our theory guarantees recovery of the latents only up to invertible and potentially nonlinear transformations, we use the Spearman correlation, which can capture nonlinear relationships unlike the Pearson correlation. We denote this score by $\text{LMS}_{\text{Spear}}$ and will use it in the dataset **ScalarLatents**. For the **BlockLatents** dataset, we cannot use Spearman correlation (because $z_B$ are two dimensional). Instead, we take the score $s_{B,B'}$ to be the $R^2$ score of a regression tree. We denote this score by $\text{LMS}_{\text{tree}}$. There are subtleties to take care of when one wants to evaluate $\text{LMS}_{\text{tree}}$ on a non-additive model due to the fact that the learned representation does not have a natural partition $\mathcal{B}$. We must thus search over partitions. We discuss this and provide further details on the metrics in Appendix B.3.

## 4.1 Results

**Additivity is important for disentanglement.** Table 1 shows that the additive decoder obtains a much higher $\text{LMS}_{\text{Spear}}$ & $\text{LMS}_{\text{Tree}}$ than its non-additive counterpart on all three datasets considered, even if both decoders have very small reconstruction errors. This is corroborated by the visualizations of Figures 4 & 5. Appendix B.5 additionally shows object-specific reconstructions for the **BlockLatents** dataset. We emphasize that disentanglement is possible even when the latent factors are dependent (or causally related), as shown on the **ScalarLatents** dataset (L-shaped support implies dependencies) and on the **BlockLatents** dataset with dependencies (Table 1). Note that prior works have relied on interventions [3, 2, 8] or Cartesian-product supports [68, 62] to deal with dependencies.

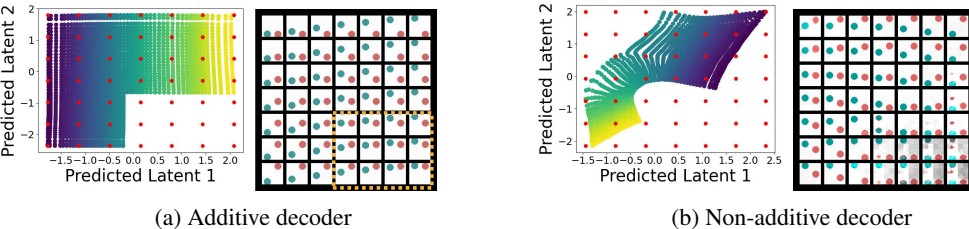

(a) Additive decoder           (b) Non-additive decoder

Figure 4: Figure (a) shows latent representation outputted by the encoder $\hat{g}(x)$ over the *training* dataset, and the corresponding reconstructed images of the additive decoder with median LMS$_{\text{Spear}}$ among runs performed on the **ScalarLatents** dataset. Figure (b) shows the same thing for the non-additive decoder. The color gradient corresponds to the value of one of the ground-truth factor, the red dots correspond to factors used to generate the images and the yellow dashed square highlights extrapolated images.

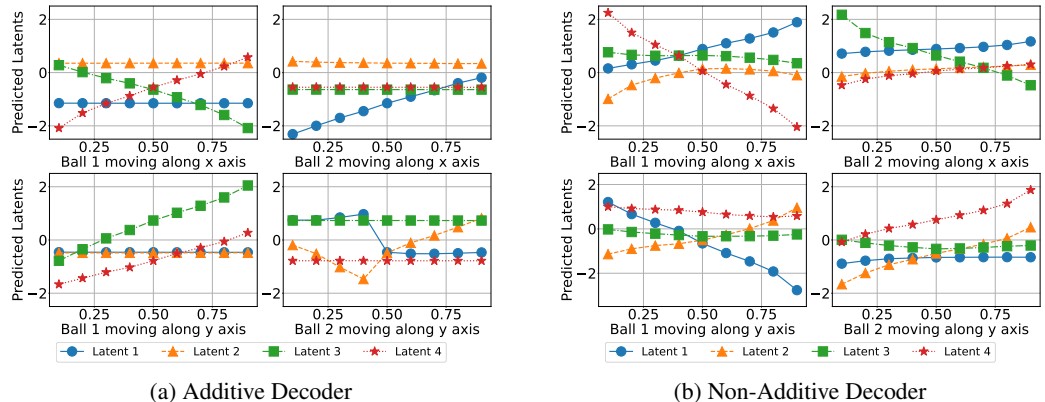

(a) Additive Decoder           (b) Non-Additive Decoder

Figure 5: Latent responses for the case of independent latents in the **BlockLatent** dataset. In each plot, we report the latent factors predicted from multiple images where one ball moves along only one axis at a time. For the additive case, at most two latents change, as it should, while more than two latents change for the non-additive case. See Appendix B.5 for details.

**Additivity is important for Cartesian-product extrapolation.** Figure 4 illustrates that the additive decoder can generate images that are outside the training domain (both balls in upper half of the image) while its non-additive counterpart cannot. Furthermore, Table 1 also corroborates this showing that the "out-of-support" (OOS) reconstruction MSE and LMS$_{\text{Spear}}$ (evaluated only on the samples never seen during training) are significantly better for the additive than for the non-additive decoder.

**Importance of connected support.** Theorem 2 required that the support of the latent factors, $\mathcal{Z}^{\text{train}}$, was path-connected. Appendix B.6 shows experiments where this assumption is violated, which yields lower LMS$_{\text{Spear}}$ for the additive decoder, thus highlighting the importance of this assumption.

# 5 Conclusion

We provided an in-depth identifiability analysis of *additive decoders*, which bears resemblance to standard decoders used in OCRL, and introduced a novel theoretical framework showing how this architecture can generate reasonable images never seen during training via "Cartesian-product extrapolation". We validated empirically both of these results and confirmed that additivity was indeed crucial. By studying rigorously how disentanglement can induce extrapolation, our work highlighted the necessity of restricting the decoder to extrapolate and set the stage for future works to explore disentanglement and extrapolation in other function classes such as masked decoders typically used in OCRL. We postulate that the type of identifiability analysis introduced in this work has the potential of expanding our understanding of creativity in generative models, ultimately resulting in representations that generalize better.

## Acknowledgements

This research was partially supported by the Canada CIFAR AI Chair Program, by an IVADO excellence PhD scholarship and by Samsung Electronics Co., Ldt. The experiments were in part enabled by computational resources provided by Calcul Québec (`calculquebec.ca`) and the Digital Research Alliance of Canada (`alliancecan.ca`). Simon Lacoste-Julien is a CIFAR Associate Fellow in the Learning in Machines & Brains program.

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

# Appendix

## Table of Contents

### Calligraphic & indexing conventions

$$
\begin{aligned}
[n] &:= \{1, 2, \ldots, n\} \\
x & \quad \text{Scalar (random or not, depending on context)} \\
\boldsymbol{x} & \quad \text{Vector (random or not, depending on context)} \\
\boldsymbol{X} & \quad \text{Matrix} \\
\mathcal{X} & \quad \text{Set/Support} \\
f & \quad \text{Scalar-valued function} \\
\boldsymbol{f} & \quad \text{Vector-valued function} \\
f|_A & \quad \text{Restriction of } f \text{ to the set } A \\
Df, D\boldsymbol{f} & \quad \text{Jacobian of } f \text{ and } \boldsymbol{f} \\
D^2 f & \quad \text{Hessian of } f \\
B \subseteq [n] & \quad \text{Subset of indices} \\
|B| & \quad \text{Cardinality of the set } B \\
\boldsymbol{x}_B & \quad \text{Vector formed with the } i\text{th coordinates of } \boldsymbol{x}, \text{ for all } i \in B \\
\boldsymbol{X}_{B,B'} & \quad \text{Matrix formed with the entries } (i, j) \in B \times B' \text{ of } \boldsymbol{X}.
\end{aligned}
$$

Given $\mathcal{X} \subseteq \mathbb{R}^n$, $\mathcal{X}_B := \{\boldsymbol{x}_B \mid \boldsymbol{x} \in \mathcal{X}\}$ (projection of $\mathcal{X}$)

### Recurrent notation

$$
\begin{aligned}
\boldsymbol{x} \in \mathbb{R}^{d_x} & \quad \text{Observation} \\
\boldsymbol{z} \in \mathbb{R}^{d_z} & \quad \text{Vector of latent factors of variations} \\
\mathcal{Z} \subseteq \mathbb{R}^{d_z} & \quad \text{Support of } \boldsymbol{z} \\
\boldsymbol{f} & \quad \text{Ground-truth decoder function} \\
\hat{\boldsymbol{f}} & \quad \text{Learned decoder function} \\
\mathcal{B} & \quad \text{A partition of } [d_z] \text{ (assumed contiguous w.l.o.g.)} \\
B \in \mathcal{B} & \quad \text{A block of the partition } \mathcal{B} \\
B(i) \in \mathcal{B} & \quad \text{The unique block of } \mathcal{B} \text{ that contains } i \\
\pi : [d_z] \to [d_z] & \quad \text{A permutation} \\
S_{\mathcal{B}} &:= \bigcup_{B \in \mathcal{B}} B^2 \\
S_{\mathcal{B}}^c &:= [d_z]^2 \setminus S_{\mathcal{B}} \\
\mathbb{R}^{d_z \times d_z}_{S_{\mathcal{B}}} &:= \{\boldsymbol{M} \in \mathbb{R}^{d_z \times d_z} \mid (i, j) \notin S_{\mathcal{B}} \implies \boldsymbol{M}_{i,j} = 0\}
\end{aligned}
$$

### General topology

$$
\begin{aligned}
\overline{\mathcal{X}} & \quad \text{Closure of the subset } \mathcal{X} \subseteq \mathbb{R}^n \text{ in the standard topology of } \mathbb{R}^n \\
\mathcal{X}^\circ & \quad \text{Interior of the subset } \mathcal{X} \subseteq \mathbb{R}^n \text{ in the standard topology of } \mathbb{R}^n
\end{aligned}
$$

## A  Identifiability and Extrapolation Analysis

### A.1  Useful definitions and lemmas

We start by recalling some notions of general topology that are going to be used later on. For a proper introduction to these concepts, see for example Munkres [56].

**Definition 6** (Regularly closed sets). *A set $\mathcal{Z} \subseteq \mathbb{R}^{d_z}$ is regularly closed if $\mathcal{Z} = \overline{\mathcal{Z}^\circ}$, i.e. if it is equal to the closure of its interior (in the standard topology of $\mathbb{R}^n$).*

**Definition 7** (Connected sets). *A set $\mathcal{Z} \subseteq \mathbb{R}^{d_z}$ is connected if it cannot be written as a union of non-empty and disjoint open sets (in the subspace topology).*

**Definition 8** (Path-connected sets). *A set $\mathcal{Z} \subseteq \mathbb{R}^{d_z}$ is path-connected if for all pair of points $\boldsymbol{z}^0, \boldsymbol{z}^1 \in \mathcal{Z}$, there exists a continuous map $\boldsymbol{\phi} : [0, 1] \to \mathcal{Z}$ such that $\boldsymbol{\phi}(0) = \boldsymbol{z}^0$ and $\boldsymbol{\phi}(1) = \boldsymbol{z}^1$. Such a map is called a path between $\boldsymbol{z}^0$ and $\boldsymbol{z}^1$.*

**Definition 9** (Homeomorphism)**.** *Let $A$ and $B$ be subsets of $\mathbb{R}^n$ equipped with the subspace topology. A function $\boldsymbol{f} : A \to B$ is an homeomorphism if it is bijective, continuous and its inverse is continuous.*

The following technical lemma will be useful in the proof of Theorem 1. For it, we will need additional notation: Let $S \subseteq A \subseteq \mathbb{R}^n$. We already saw that $\overline{S}$ refers to the closure $S$ in the $\mathbb{R}^n$ topology. We will denote by $\mathrm{cl}_A(S)$ the closure of $S$ in the subspace topology of $A$ induced by $\mathbb{R}^n$, which is not necessarily the same as $\overline{S}$. In fact, both can be related via $\mathrm{cl}_A = \overline{S} \cap A$ (see Munkres [56, Theorem 17.4, p.95]).

**Lemma 4.** *Let $A, B \subseteq \mathbb{R}^n$ and suppose there exists an homeomorphism $\boldsymbol{f} : A \to B$. If $A$ is regularly closed in $\mathbb{R}^n$, we have that $B \subseteq \overline{B^\circ}$.*

*Proof.* Note that $\boldsymbol{f}\big|_{A^\circ}$ is a continuous injective function from the open set $A^\circ$ to $\boldsymbol{f}(A^\circ)$. By the "invariance of domain" theorem [56, p.381], we have that $\boldsymbol{f}(A^\circ)$ must be open in $\mathbb{R}^n$. Of course, we have that $\boldsymbol{f}(A^\circ) \subseteq B$, and thus $\boldsymbol{f}(A^\circ) \subseteq B^\circ$ (the interior of $B$ is the largest open set contained in $B$). Analogously, $\boldsymbol{f}^{-1}\big|_{B^\circ}$ is a continuous injective function from the open set $B^\circ$ to $\boldsymbol{f}^{-1}(B^\circ)$. Again, by "invariance of domain", $\boldsymbol{f}^{-1}(B^\circ)$ must be open in $\mathbb{R}^n$ and thus $\boldsymbol{f}^{-1}(B^\circ) \subseteq A^\circ$. We can conclude that $\boldsymbol{f}(A^\circ) = B^\circ$.

We can conclude as follow:

$$B = \boldsymbol{f}(A) = \boldsymbol{f}(\overline{A^\circ}) = \boldsymbol{f}(\overline{A^\circ} \cap A) = \boldsymbol{f}(\mathrm{cl}_A(A^\circ)) \subseteq \mathrm{cl}_B(\boldsymbol{f}(A^\circ)) = \mathrm{cl}_B(B^\circ) = \overline{B^\circ} \cap B \subseteq \overline{B^\circ},$$

where the first inclusion holds by continuity of $\boldsymbol{f}$ [56, Thm.18.1 p.104]. $\qquad\square$

This lemma is taken from [42].

**Lemma 5** (Sparsity pattern of an invertible matrix contains a permutation)**.** *Let $\boldsymbol{L} \in \mathbb{R}^{m \times m}$ be an invertible matrix. Then, there exists a permutation $\sigma$ such that $\boldsymbol{L}_{i,\sigma(i)} \neq 0$ for all $i$.*

*Proof.* Since the matrix $\boldsymbol{L}$ is invertible, its determinant is non-zero, i.e.

$$\det(\boldsymbol{L}) := \sum_{\pi \in \mathfrak{S}_m} \mathrm{sign}(\pi) \prod_{i=1}^{m} \boldsymbol{L}_{i,\pi(i)} \neq 0 \,, \tag{12}$$

where $\mathfrak{S}_m$ is the set of $m$-permutations. This equation implies that at least one term of the sum is non-zero, meaning there exists $\pi \in \mathfrak{S}_m$ such that for all $i \in [m]$, $\boldsymbol{L}_{i,\pi(i)} \neq 0$. $\qquad\square$

**Definition 10** (Aligned subspaces of $\mathbb{R}^{m \times n}$)**.** *Given a subset $S \subseteq \{1, ..., m\} \times \{1, ..., n\}$, we define*

$$\mathbb{R}_S^{m \times n} := \{\boldsymbol{M} \in \mathbb{R}^{m \times n} \mid (i,j) \notin S \implies \boldsymbol{M}_{i,j} = 0\} \,. \tag{13}$$

**Definition 11** (Useful sets)**.** *Given a partition $\mathcal{B}$ of $[d]$, we define*

$$S_\mathcal{B} := \bigcup_{B \in \mathcal{B}} B^2 \qquad S_\mathcal{B}^c := \{1, \ldots, d_z\}^2 \setminus S_\mathcal{B} \tag{14}$$

**Definition 12** ($C^k$-diffeomorphism)**.** *Let $A \subseteq \mathbb{R}^n$ and $B \subseteq \mathbb{R}^m$. A map $\boldsymbol{f} : A \to B$ is said to be a $C^k$-diffeomorphism if it is bijective, $C^2$ and has a $C^2$ inverse.*

**Remark 2.** *Differentiability is typically defined for functions that have an open domain in $\mathbb{R}^n$. However, in the definition above, the set $A$ might not be open in $\mathbb{R}^n$ and $B$ might not be open in $\mathbb{R}^m$. In the case of an arbitrary domain $A$, it is customary to say that a function $\boldsymbol{f} : A \subseteq \mathbb{R}^n \to \mathbb{R}^m$ is $C^k$ if there exists a $C^k$ function $\boldsymbol{g}$ defined on an open set $U \subseteq \mathbb{R}^n$ that contains $A$ such that $\boldsymbol{g}\big|_A = \boldsymbol{f}$ (i.e. $\boldsymbol{g}$ extends $\boldsymbol{f}$). With this definition, we have that a composition of $C^k$ functions is $C^k$, as usual. See for example p.199 of Munkres [55].*

The following lemma allows us to unambiguously define the $k$ first derivatives of a $C^k$ function $\boldsymbol{f} : A \to \mathbb{R}^m$ on the set $\overline{A^\circ}$.

**Lemma 6.** *Let $A \subseteq \mathbb{R}^n$ and $\boldsymbol{f} : A \to \mathbb{R}^m$ be a $C^k$ function. Then, its $k$ first derivatives is uniquely defined on $\overline{A^\circ}$ in the sense that they do not depend on the specific choice of $C^k$ extension.*

*Proof.* Let $\boldsymbol{g} : U \to \mathbb{R}^n$ and $\boldsymbol{h} : V \to \mathbb{R}^n$ be two $C^k$ extensions of $\boldsymbol{f}$ to $U \subseteq \mathbb{R}^n$ and $V \subseteq \mathbb{R}^n$ both open in $\mathbb{R}^n$. By definition,

$$\boldsymbol{g}(\boldsymbol{x}) = \boldsymbol{f}(\boldsymbol{x}) = \boldsymbol{h}(\boldsymbol{x}), \ \forall \boldsymbol{x} \in A \,. \tag{15}$$

The usual derivative is uniquely defined on the interior of the domain, so that

$$D\boldsymbol{g}(\boldsymbol{x}) = D\boldsymbol{f}(\boldsymbol{x}) = D\boldsymbol{h}(\boldsymbol{x}), \ \forall \boldsymbol{x} \in A^\circ \,. \tag{16}$$

Consider a point $\boldsymbol{x}_0 \in \overline{A^\circ}$. By definition of closure, there exists a sequence $\{\boldsymbol{x}_k\}_{k=1}^\infty \subseteq A^\circ$ s.t. $\lim_{k \to \infty} \boldsymbol{x}_k = \boldsymbol{x}_0$. We thus have that

$$\lim_{k \to \infty} D\boldsymbol{g}(\boldsymbol{x}_k) = \lim_{k \to \infty} D\boldsymbol{h}(\boldsymbol{x}_k) \tag{17}$$

$$D\boldsymbol{g}(\boldsymbol{x}_0) = D\boldsymbol{h}(\boldsymbol{x}_0) \,, \tag{18}$$

where we used the fact that the derivatives of $\boldsymbol{g}$ and $\boldsymbol{h}$ are continuous to go to the second line. Thus, all the $C^k$ extensions of $\boldsymbol{f}$ must have equal derivatives on $\overline{A^\circ}$. This means we can unambiguously define the derivative of $\boldsymbol{f}$ everywhere on $\overline{A^\circ}$ to be equal to the derivative of one of its $C^k$ extensions. Since $\boldsymbol{f}$ is $C^k$, its derivative $D\boldsymbol{f}$ is $C^{k-1}$, we can thus apply the same argument to get that the second derivative of $\boldsymbol{f}$ is uniquely defined on $\overline{\overline{A^\circ}^\circ}$. It can be shown that $\overline{\overline{A^\circ}^\circ} = \overline{A^\circ}$. One can thus apply the same argument recursively to show that the first $k$ derivatives of $\boldsymbol{f}$ are uniquely defined on $\overline{A^\circ}$. $\quad\square$

**Definition 13** ($C^k$-diffeomorphism onto its image)**.** *Let $A \subseteq \mathbb{R}^n$. A map $\boldsymbol{f} : A \to \mathbb{R}^m$ is said to be a $C^k$-diffeomorphism onto its image if the restriction $\boldsymbol{f}$ to its image $\tilde{\boldsymbol{f}} : A \to \boldsymbol{f}(A)$ is a $C^k$-diffeomorphism.*

**Remark 3.** *If $S \subseteq A \subseteq \mathbb{R}^n$ and $\boldsymbol{f} : A \to \mathbb{R}^m$ is a $C^k$-diffeomorphism on its image, then the restriction of $\boldsymbol{f}$ to $S$, i.e. $\boldsymbol{f}\big|_S$, is also a $C^k$ diffeomorphism on its image. That is because $\boldsymbol{f}\big|_S$ is clearly bijective, is $C^k$ (simply take the $C^k$ extension of $\boldsymbol{f}$) and so is its inverse (simply take the $C^k$ extension of $\boldsymbol{f}^{-1}$).*

## A.2 Relationship between additive decoders and the diagonal Hessian penalty

**Proposition 7** (Equivalence between additivity and diagonal Hessian)**.** *Let $\boldsymbol{f} : \mathbb{R}^{d_z} \to \mathbb{R}^{d_x}$ be a $C^2$ function. Then,*

$$\begin{array}{ll} \forall \boldsymbol{z} \in \mathbb{R}^{d_z}, \ \boldsymbol{f}(\boldsymbol{z}) = \sum_{B \in \mathcal{B}} \boldsymbol{f}^{(B)}(\boldsymbol{z}_B) & \forall k \in [d_x], \ \boldsymbol{z} \in \mathbb{R}^{d_z}, \ D^2 \boldsymbol{f}_k(\boldsymbol{z}) \text{ is} \\ \text{where } \boldsymbol{f}^{(B)} : \mathbb{R}^{|B|} \to \mathbb{R}^{d_x} \text{ is } C^2. & \Longleftrightarrow \quad \text{block diagonal with blocks in } \mathcal{B}. \end{array} \tag{19}$$

*Proof.* We start by showing the " $\Longrightarrow$ " direction. Let $B$ and $B'$ be two distinct blocks of $\mathcal{B}$. Let $i \in B$ and $i' \in B'$. We can compute the derivative of $\boldsymbol{f}_k$ w.r.t. $\boldsymbol{z}_i$:

$$D_i \boldsymbol{f}_k(\boldsymbol{z}) = \sum_{\bar{B} \in \mathcal{B}} D_i \boldsymbol{f}_k^{(\bar{B})}(\boldsymbol{z}_{\bar{B}}) = D_i \boldsymbol{f}_k^{(B)}(\boldsymbol{z}_B) \,, \tag{20}$$

where the last equality holds because $i \in B$ and not in any other block $\bar{B}$. Furthermore,

$$D_{i,i'}^2 \boldsymbol{f}_k(\boldsymbol{z}) = D_{i,i'}^2 \boldsymbol{f}_k^{(B)}(\boldsymbol{z}_B) = 0 \,, \tag{21}$$

where the last equality holds because $i' \notin B$. This shows that $D^2 \boldsymbol{f}_k(\boldsymbol{z})$ is block diagonal.

We now show the " $\Longleftarrow$ " direction. Fix $k \in [d_x]$, $B \in \mathcal{B}$. We know that $D_{B,B^c}^2 \boldsymbol{f}_k(\boldsymbol{z}) = 0$ for all $\boldsymbol{z} \in \mathbb{R}^{d_z}$. Fix $\boldsymbol{z} \in \mathbb{R}^{d_z}$. Consider a continuously differentiable path $\boldsymbol{\phi} : [0, 1] \to \mathbb{R}^{|B^c|}$ such that $\boldsymbol{\phi}(0) = 0$ and $\boldsymbol{\phi}(1) = \boldsymbol{z}_{B^c}$. As $D_{B,B^c}^2 \boldsymbol{f}_k(\boldsymbol{z})$ is a continuous function of $\boldsymbol{z}$, we can use the fundamental theorem of calculus for line integrals to get that

$$D_B \boldsymbol{f}_k(\boldsymbol{z}_B, \boldsymbol{z}_{B^c}) - D_B \boldsymbol{f}_k(\boldsymbol{z}_B, 0) = \int_0^1 \underbrace{D_{B,B^c}^2 \boldsymbol{f}_k(\boldsymbol{z}_B, \boldsymbol{\phi}(t))}_{=0} \boldsymbol{\phi}'(t) dt = 0 \,, \tag{22}$$

(where $D_{B,B^c}^2 \boldsymbol{f}_k(\boldsymbol{z}_B, \boldsymbol{\phi}(t)) \boldsymbol{\phi}'(t)$ denotes a matrix-vector product) which implies that

$$D_B \boldsymbol{f}_k(\boldsymbol{z}) = D_B \boldsymbol{f}_k(\boldsymbol{z}_B, 0) \,. \tag{23}$$

And the above equality holds for all $B \in \mathcal{B}$ and all $z \in \mathbb{R}^{d_z}$.

Choose an arbitrary $z \in \mathbb{R}^{d_z}$. Consider a continously differentiable path $\psi : [0,1] \to \mathbb{R}^{d_z}$ such that $\psi(0) = 0$ and $\psi(1) = z$. By applying the fundamental theorem of calculus for line integrals once more, we have that

$$\boldsymbol{f}_k(\boldsymbol{z}) - \boldsymbol{f}_k(0) = \int_0^1 D\boldsymbol{f}_k(\boldsymbol{\psi}(t))\boldsymbol{\psi}'(t)dt \tag{24}$$

$$= \int_0^1 \sum_{B \in \mathcal{B}} D_B \boldsymbol{f}_k(\boldsymbol{\psi}(t))\boldsymbol{\psi}'_B(t)dt \tag{25}$$

$$= \sum_{B \in \mathcal{B}} \int_0^1 D_B \boldsymbol{f}_k(\boldsymbol{\psi}(t))\boldsymbol{\psi}'_B(t)dt \tag{26}$$

$$= \sum_{B \in \mathcal{B}} \int_0^1 D_B \boldsymbol{f}_k(\boldsymbol{\psi}_B(t), 0)\boldsymbol{\psi}'_B(t)dt \,, \tag{27}$$

where the last equality holds by (23). We can further apply the fundamental theorem of calculus for line integrals to each term $\int_0^1 D_B \boldsymbol{f}_k(\boldsymbol{\psi}_B(t), 0)\boldsymbol{\psi}'_B(t)dt$ to get

$$\boldsymbol{f}_k(\boldsymbol{z}) - \boldsymbol{f}_k(0) = \sum_{B \in \mathcal{B}} (\boldsymbol{f}_k(\boldsymbol{z}_B, 0) - \boldsymbol{f}_k(0, 0)) \tag{28}$$

$$\implies \boldsymbol{f}_k(\boldsymbol{z}) = \boldsymbol{f}_k(0) + \sum_{B \in \mathcal{B}} (\boldsymbol{f}_k(\boldsymbol{z}_B, 0) - \boldsymbol{f}_k(0)) \tag{29}$$

$$= \sum_{B \in \mathcal{B}} \underbrace{\left( \boldsymbol{f}_k(\boldsymbol{z}_B, 0) - \frac{|\mathcal{B}| - 1}{|\mathcal{B}|} \boldsymbol{f}_k(0) \right)}_{\boldsymbol{f}_k^{(B)}(\boldsymbol{z}_B):=} \,. \tag{30}$$

and since $z$ was arbitrary, the above holds for all $z \in \mathbb{R}^{d_z}$. Note that the functions $\boldsymbol{f}_k^{(B)}(\boldsymbol{z}_B)$ must be $C^2$ because $\boldsymbol{f}_k$ is $C^2$. This concludes the proof. $\qquad \square$

### A.3 Additive decoders form a superset of compositional decoders [7]

Compositional decoders were introduced by Brady et al. [7] as a suitable class of functions to perform object-centric representation learning with identifiability guarantees. They are also interested in block-disentanglement, but, contrarily to our work, they assume that the latent vector $z$ is fully supported, i.e. $\mathcal{Z} = \mathbb{R}^{d_z}$. We now rewrite the definition of compositional decoders in the notation used in this work:

**Definition 14** (Compositional decoders, adapted from [7]). *Given a partition $\mathcal{B}$, a differentiable decoder $\boldsymbol{f} : \mathbb{R}^{d_z} \to \mathbb{R}^{d_x}$ is said to be compositional w.r.t. $\mathcal{B}$ whenever the Jacobian $D\boldsymbol{f}(\boldsymbol{z})$ is such that for all $i \in [d_z], B \in \mathcal{B}, \boldsymbol{z} \in \mathbb{R}^{d_z}$, we have*

$$D_B \boldsymbol{f}_i(\boldsymbol{z}) \neq \boldsymbol{0} \implies D_{B^c} \boldsymbol{f}_i(\boldsymbol{z}) = \boldsymbol{0} \,,$$

*where $B^c$ is the complement of $B \in \mathcal{B}$.*

In other words, each line of the Jacobian can have nonzero values only in one block $B \in \mathcal{B}$. Note that this nonzero block can change with different values of $z$.

The next result shows that additive decoders form a superset of $C^2$ compositional decoders (Brady et al. [7] assumed only $C^1$). Note that additive decoders are *strictly* more expressive than $C^2$ compositional decoders because some additive functions are not compositional, like Example 3 for instance.

**Proposition 8** (Compositional implies additive). *Given a partition $\mathcal{B}$, if $\boldsymbol{f} : \mathbb{R}^{d_z} \to \mathbb{R}^{d_x}$ is compositional (Definition 14) and $C^2$, then it is also additive (Definition 1).*

*Proof.* Choose any $i \in [d_x]$. Our strategy will be to show that $D^2 \boldsymbol{f}_i$ is block diagonal everywhere on $\mathbb{R}^{d_z}$ and use Proposition 7 to conclude that $\boldsymbol{f}_i$ is additive.

Choose an arbitrary $z_0 \in \mathbb{R}^{d_z}$. By compositionality, there exists a block $B \in \mathcal{B}$ such that $D_{B^c} f_i(z_0) = 0$. We consider two cases separately:

**Case 1** Assume $D_B f_i(z_0) \neq 0$. By continuity of $D_B f_i$, there exists an open neighborhood of $z_0$, $U$, s.t. for all $z \in U$, $D_B f_i(z) \neq 0$. By compositionality, this means that, for all $z \in U$, $D_{B^c} f_i(z) = 0$. When a function is zero on an open set, its derivative must also be zero, hence $DD_{B^c} f_i(z_0) = 0$. Because $f$ is $C^2$, the Hessian is symmetric so that we also have $D_{B^c} D f_i(z_0) = 0$. We can thus conclude that the Hessian $D^2 f_i(z_0)$ is such that all entries are zero except possibly for $D^2 f_i(z_0)_{B,B}$. Hence, $D^2 f_i(z_0)$ is block diagonal with blocks in $\mathcal{B}$.

**Case 2:** Assume $D_B f_i(z_0) = 0$. This means the whole row of the Jacobian is zero, i.e. $D f_i(z_0) = 0$. By continuity of $D f_i$, we have that the set $V := (D f_i)^{-1}(\{0\})$ is closed. Thus this set decomposes as $V = V^\circ \cup \partial V$ where $V^\circ$ and $\partial V$ are the interior and boundary of $V$, respectively.

**Case 2.1:** Suppose $z_0 \in V^\circ$. Then we can take a derivative so that $D^2 f_i(z_0) = 0$, which of course means that $D^2 f_i(z_0)$ is diagonal.

**Case 2.2:** Suppose $z_0 \in \partial V$. By the definition of boundary, for all open set $U$ containing $z_0$, $U$ intersects with the complement of $V$, i.e. $(D f_i)^{-1}(\mathbb{R}^{d_z} \setminus \{0\})$. This means we can construct a sequence $\{z_k\}_{k=1}^\infty \subseteq V^c$ which converges to $z_0$. By **Case 1**, we have that for all $k \geq 1$, $D^2 f_i(z_k)$ is block diagonal. This means that $\lim_{k \to \infty} D^2 f_i(z_k)$ is block diagonal. Moreover, by continuity of $D^2 f_i$, we have that $\lim_{k \to \infty} D^2 f_i(z_k) = D^2 f_i(z_0)$. Hence $D^2 f_i(z_0)$ is block diagonal.

We showed that for all $z_0 \in \mathbb{R}^{d_z}$, $D^2 f_i(z_0)$ is block diagonal. Hence, $f$ is additive by Proposition 7. $\square$

### A.4 Examples of local but non-global disentanglement

In this section, we provide examples of mapping $v : \hat{\mathcal{Z}}^{\text{train}} \to \mathcal{Z}^{\text{train}}$ that satisfy the *local* disentanglement property of Definition 4, but not the *global* disentanglement property of Definition 3. Note that these notions are defined for pairs of decoders $f$ and $\hat{f}$, but here we construct directly the function $v$ which is usually defined as $f^{-1} \circ \hat{f}$. However, given $v$ we can always define $f$ and $\hat{f}$ to be such that $f^{-1} \circ \hat{f} = v$: Simply take $f(z) := [z_1, \ldots, z_{d_z}, 0, \ldots, 0]^\top \in \mathbb{R}^{d_x}$ and $\hat{f} := f \circ v$. This construction however yields a decoder $f$ that is not sufficiently nonlinear (Assumption 2). Clearly the mappings $v$ that we provide in the following examples cannot be written as compositions of decoders $f^{-1} \circ \hat{f}$ where $f$ and $\hat{f}$ satisfy all assumptions of Theorem 2, as this would contradict the theorem. In Examples 5 & 6, the path-connected assumption of Theorem 2 is violated. In Example 7, it is less obvious to see which assumptions would be violated.

**Example 5** (Disconnected support with changing permutation)**.** *Let* $v : \hat{\mathcal{Z}} \to \mathbb{R}^2$ *s.t.* $\hat{\mathcal{Z}} = \hat{\mathcal{Z}}^{(1)} \cup \hat{\mathcal{Z}}^{(2)} \subseteq \mathbb{R}^2$ *where* $\hat{\mathcal{Z}}^{(1)} = \{z \in \mathbb{R}^2 \mid z_1 \leq 0 \text{ and } z_2 \leq 0\}$ *and* $\hat{\mathcal{Z}}^{(2)} = \{z \in \mathbb{R}^2 \mid z_1 \geq 1 \text{ and } z_2 \geq 1\}$. *Assume*

$$v(z) := \begin{cases} (z_1, z_2), & \text{if } z \in \hat{\mathcal{Z}}^{(1)} \\ (z_2, z_1), & \text{if } z \in \hat{\mathcal{Z}}^{(2)} \end{cases}. \tag{31}$$

***Step 1: $v$ is a diffeomorphism.*** *Note that $v$ is its own inverse. Indeed,*

$$v(v(z)) = \begin{cases} v(z_1, z_2) = (z_1, z_2), & \text{if } z \in \hat{\mathcal{Z}}^{(1)} \\ v(z_2, z_1) = (z_1, z_2), & \text{if } z \in \hat{\mathcal{Z}}^{(2)} \end{cases}.$$

*Thus, $v$ is bijective on its image. Clearly, $v$ is $C^2$, thus $v^{-1} = v$ is also $C^2$. Hence, $v$ is a $C^2$-diffeomorphism.*

***Step 2: $v$ is locally disentangled.*** *The Jacobian of $v$ is given by*

$$Dv(z) := \begin{cases} \begin{bmatrix} 1 & 0 \\ 0 & 1 \end{bmatrix}, & \text{if } z \in \hat{\mathcal{Z}}^{(1)} \\ \begin{bmatrix} 0 & 1 \\ 1 & 0 \end{bmatrix}, & \text{if } z \in \hat{\mathcal{Z}}^{(2)} \end{cases}, \tag{32}$$

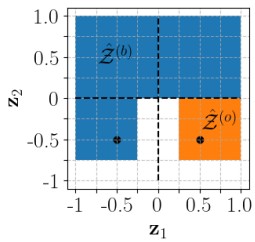

Figure 6: Illustration of $\hat{\mathcal{Z}} = \hat{\mathcal{Z}}^{(b)} \cup \hat{\mathcal{Z}}^{(o)}$ in Example 7 where $\hat{\mathcal{Z}}^{(b)}$ is the blue region and $\hat{\mathcal{Z}}^{(o)}$ is the orange region. The two black dots correspond to $(-1/2, -1/2)$ and $(1/2, -1/2)$, where the function $v_2(z_1, z_2)$ is evaluated to show that it is not constant in $z_1$.

*which is everywhere a permutation matrix, hence $v$ is locally disentangled.*

**Step 3: $v$ is not globally disentangled.** *That is because $v_1(z_1, z_2)$ depends on both $z_1$ and $z_2$. Indeed, if $z_2 = 0$, we have that $v_1(-1, 0) = -1 \neq 0 = v_1(0, 0)$. Also, if $z_1 = 1$, we have that $v_1(1, 1) = 1 \neq 2 = v_1(1, 2)$.*

**Example 6** (Disconnected support with fixed permutation). *Let $v : \hat{\mathcal{Z}} \to \mathbb{R}^2$ s.t. $\hat{\mathcal{Z}} = \hat{\mathcal{Z}}^{(1)} \cup \hat{\mathcal{Z}}^{(2)} \subseteq \mathbb{R}^2$ where $\hat{\mathcal{Z}}^{(1)} = \{z \in \mathbb{R}^2 \mid z_2 \leq 0\}$ and $\hat{\mathcal{Z}}^{(2)} = \{z \in \mathbb{R}^2 \mid z_2 \geq 1\}$. Assume $v(z) := z + \mathbb{1}(z \in \hat{\mathcal{Z}}^{(2)})$.*

**Step 1: $v$ is a diffeomorphism.** *The image of $v$ is the union of the following two sets: $\mathcal{Z}^{(1)} := v(\hat{\mathcal{Z}}^{(1)}) = \hat{\mathcal{Z}}^{(1)}$ and $\mathcal{Z}^{(2)} := v(\hat{\mathcal{Z}}^{(2)}) = \{z \in \mathbb{R}^2 \mid z_2 \geq 2\}$. Consider the map $w : \mathcal{Z}^{(1)} \cup \mathcal{Z}^{(2)} \to \hat{\mathcal{Z}}$ defined as $w(z) := z - \mathbb{1}(z \in \mathcal{Z}^{(2)})$. We now show that $w$ is the inverse of $v$:*

$$w(v(z)) = v(z) - \mathbb{1}(v(z) \in \mathcal{Z}^{(2)}) \tag{33}$$

$$= z + \mathbb{1}(z \in \hat{\mathcal{Z}}^{(2)}) - \mathbb{1}(z + \mathbb{1}(z \in \hat{\mathcal{Z}}^{(2)}) \in \mathcal{Z}^{(2)}). \tag{34}$$

*If $z \in \hat{\mathcal{Z}}^{(2)}$, we have*

$$w(v(z)) = z + \mathbb{1} - \mathbb{1}(z + \mathbb{1} \in \mathcal{Z}^{(2)}) \tag{35}$$

$$= z + \mathbb{1} - \mathbb{1}(z \in \hat{\mathcal{Z}}^{(2)}) = z. \tag{36}$$

*If $z \in \hat{\mathcal{Z}}^{(1)}$, we have*

$$w(v(z)) = z - \mathbb{1}(z \in \mathcal{Z}^{(2)}) = z. \tag{37}$$

*A similar argument can be made to show that $v(w(z)) = z$. Thus $w$ is the inverse of $v$. Both $v$ and its inverse $w$ are $C^2$, thus $v$ is a $C^2$-diffeomorphism on its image.*

**Step 2: $v$ is locally disentangled.** *This is clear since $Dv(z) = I$ everywhere.*

**Step 3: $v$ is not globally disentangled.** *Indeed, the function $v_1(z_1, z_2) = z_1 + \mathbb{1}(z \in \hat{\mathcal{Z}}^{(2)})$ is not constant in $z_2$.*

**Example 7** (Connected support). *Let $v : \hat{\mathcal{Z}} \to \mathbb{R}^2$ s.t. $\hat{\mathcal{Z}} = \hat{\mathcal{Z}}^{(b)} \cup \hat{\mathcal{Z}}^{(o)}$ where $\hat{\mathcal{Z}}^{(b)}$ and $\hat{\mathcal{Z}}^{(o)}$ are respectively the blue and orange regions of Figure 6. Both regions contain their boundaries. The function $v$ is defined as follows:*

$$v_1(z) := z_1 \tag{38}$$

$$v_2(z) := \begin{cases} \frac{(z_2+1)^2+1}{2}, & \text{if } z \in \hat{\mathcal{Z}}^{(b)} \\ e^{z_2}, & \text{if } z \in \hat{\mathcal{Z}}^{(o)} \end{cases}. \tag{39}$$

**Step 1: $v$ is a diffeomorphism.** *Clearly, $v_1$ is $C^2$. To show that $v_2$ also is, we must verify that $v_2(z)$ is $C^2$ at the frontier between $\hat{\mathcal{Z}}^{(b)}$ and $\hat{\mathcal{Z}}^{(o)}$, i.e. when $z \in [1/4, 1] \times \{0\}$.*

*$v_2(z)$ is continuous since*

$$\left.\frac{(z_2 + 1)^2 + 1}{2}\right|_{z_2=0} = 1 = e^{z_2}\big|_{z_2=0}. \tag{40}$$

$v_2(z)$ **is** $C^1$ **since**

$$\left.\left(\frac{(z_2+1)^2+1}{2}\right)'\right|_{z_2=0} = (z_2+1)|_{z_2=0} = 1 = e^{z_2}|_{z_2=0} = (e^{z_2})'|_{z_2=0} \, . \tag{41}$$

$v_2(z)$ **is** $C^2$ **since**

$$\left.\left(\frac{(z_2+1)^2+1}{2}\right)''\right|_{z_2=0} = 1|_{z_2=0} = 1 = e^{z_2}|_{z_2=0} = (e^{z_2})''|_{z_2=0} \, . \tag{42}$$

*We will now find an explicit expression for the inverse of $v$. Define*

$$w_1(z) := z_1 \tag{43}$$

$$w_2(z) := \begin{cases} \sqrt{2z_2-1}-1, & \text{if } z \in v(\hat{\mathcal{Z}}^{(b)}) \\ \log(z_2), & \text{if } z \in v(\hat{\mathcal{Z}}^{(o)}) \end{cases} \, . \tag{44}$$

*It is straightforward to see that $w(v(z)) = z$ for all $z \in \hat{\mathcal{Z}}$. One can also show that $w$ is $C^2$ at the boundary between both regions $v(\hat{\mathcal{Z}}^{(b)})$ and $v(\hat{\mathcal{Z}}^{(o)})$, i.e. when $z \in [1/4, 1] \times \{1\}$.*

*Since both $v$ and its inverse $w$ are $C^2$, $v$ is a $C^2$-diffeomorphism.*

**Step 2: $v$ is locally disentangled.** *The Jacobian of $v$ is*

$$Dv(z) := \begin{cases} \begin{bmatrix} 1 & 0 \\ 0 & z_2+1 \end{bmatrix}, & \text{if } z \in \hat{\mathcal{Z}}^{(b)} \\ \begin{bmatrix} 1 & 0 \\ 0 & e^{z_2} \end{bmatrix}, & \text{if } z \in \hat{\mathcal{Z}}^{(o)} \end{cases} \, , \tag{45}$$

*which is a permutation-scaling matrix everywhere on $\hat{\mathcal{Z}}$. Thus local disentanglement holds.*

**Step 3: $v$ is not globally disentangled.** *However, $v_2(z_1, z_2)$ is not constant in $z_1$. Indeed,*

$$v_2\left(-\frac{1}{2}, -\frac{1}{2}\right) = \left.\frac{(z_2+1)^2+1}{2}\right|_{z_2=-1/2} = \frac{5}{8} \neq e^{-1/2} = v_2\left(\frac{1}{2}, -\frac{1}{2}\right). \tag{46}$$

*Thus global disentanglement does not hold.*

## A.5 Proof of Theorem 1

**Proposition 9.** *Suppose that the data-generating process satisfies Assumption 1, that the learned decoder $\hat{f} : \mathbb{R}^{d_z} \to \mathbb{R}^{d_x}$ is a $C^2$-diffeomorphism onto its image and that the encoder $\hat{g} : \mathbb{R}^{d_x} \to \mathbb{R}^{d_z}$ is continuous. Then, if $\hat{f}$ and $\hat{g}$ solve the reconstruction problem on the training distribution, i.e. $\mathbb{E}^{\text{train}}||x - \hat{f}(\hat{g}(x))||^2 = 0$, we have that $f(\mathcal{Z}^{\text{train}}) = \hat{f}(\hat{\mathcal{Z}}^{\text{train}})$ and the map $v := f^{-1} \circ \hat{f}$ is a $C^2$-diffeomorphism from $\hat{\mathcal{Z}}^{\text{train}}$ to $\mathcal{Z}^{\text{train}}$.*

*Proof.* First note that

$$\mathbb{E}^{\text{train}}||x - \hat{f}(\hat{g}(x))||^2 = \mathbb{E}^{\text{train}}||f(z) - \hat{f}(\hat{g}(f(z)))||^2 = 0 \, , \tag{47}$$

which implies that, for $\mathbb{P}_z^{\text{train}}$-almost every $z \in \mathcal{Z}^{\text{train}}$,

$$f(z) = \hat{f}(\hat{g}(f(z))) \, .$$

But since the functions on both sides of the equations are continuous, the equality holds for all $z \in \mathcal{Z}^{\text{train}}$. This implies that $f(\mathcal{Z}^{\text{train}}) = \hat{f} \circ \hat{g} \circ f(\mathcal{Z}^{\text{train}}) = \hat{f}(\hat{\mathcal{Z}}^{\text{train}})$.

By Remark 3, the restrictions $f : \mathcal{Z}^{\text{train}} \to f(\mathcal{Z}^{\text{train}})$ and $\hat{f} : \hat{\mathcal{Z}}^{\text{train}} \to \hat{f}(\hat{\mathcal{Z}}^{\text{train}})$ are $C^2$-diffeomorphisms and, because $f(\mathcal{Z}^{\text{train}}) = \hat{f}(\hat{\mathcal{Z}}^{\text{train}})$, their composition $v := f^{-1} \circ \hat{f} : \hat{\mathcal{Z}}^{\text{train}} \to \mathcal{Z}^{\text{train}}$ is a well defined $C^2$-diffeomorphism (since $C^2$-diffeomorphisms are closed under composition). $\qquad\square$

**Theorem 1** (Local disentanglement via additive decoders). *Suppose that the data-generating process satisfies Assumption 1, that the learned decoder $\hat{\boldsymbol{f}} : \mathbb{R}^{d_z} \to \mathbb{R}^{d_x}$ is a $C^2$-diffeomorphism, that the encoder $\hat{\boldsymbol{g}} : \mathbb{R}^{d_x} \to \mathbb{R}^{d_z}$ is continuous, that both $\boldsymbol{f}$ and $\hat{\boldsymbol{f}}$ are additive (Definition 1) and that $\boldsymbol{f}$ is sufficiently nonlinear as formalized by Assumption 2. Then, if $\hat{\boldsymbol{f}}$ and $\hat{\boldsymbol{g}}$ solve the reconstruction problem on the training distribution, i.e. $\mathbb{E}^{\text{train}}||\boldsymbol{x} - \hat{\boldsymbol{f}}(\hat{\boldsymbol{g}}(\boldsymbol{x}))||^2 = 0$, we have that $\hat{\boldsymbol{f}}$ is locally $\mathcal{B}$-disentangled w.r.t. $\boldsymbol{f}$ (Definition 4).*

*Proof.* We can apply Proposition 9 and have that the map $\boldsymbol{v} := \boldsymbol{f}^{-1} \circ \hat{\boldsymbol{f}}$ is a $C^2$-diffeomorphism from $\hat{\mathcal{Z}}^{\text{train}}$ to $\mathcal{Z}^{\text{train}}$. This allows one to write

$$\boldsymbol{f} \circ \boldsymbol{v}(\boldsymbol{z}) = \hat{\boldsymbol{f}}(\boldsymbol{z}) \; \forall \boldsymbol{z} \in \hat{\mathcal{Z}}^{\text{train}} \tag{48}$$

$$\sum_{B \in \mathcal{B}} \boldsymbol{f}^{(B)}(\boldsymbol{v}_B(\boldsymbol{z})) = \sum_{B \in \mathcal{B}} \hat{\boldsymbol{f}}^{(B)}(\boldsymbol{z}_B) \; \forall \boldsymbol{z} \in \hat{\mathcal{Z}}^{\text{train}} . \tag{49}$$

Since $\mathcal{Z}^{\text{train}}$ is regularly closed and is diffeomorphic to $\hat{\mathcal{Z}}^{\text{train}}$, by Lemma 4, we must have that $\hat{\mathcal{Z}}^{\text{train}} \subseteq \overline{(\hat{\mathcal{Z}}^{\text{train}})^\circ}$. Moreover, the left and right hand side of (49) are $C^2$, which means they have uniquely defined first and second derivatives on $\overline{(\hat{\mathcal{Z}}^{\text{train}})^\circ}$ by Lemma 6. This means the derivatives are uniquely defined on $\hat{\mathcal{Z}}^{\text{train}}$.

Let $\boldsymbol{z} \in \hat{\mathcal{Z}}^{\text{train}}$. Choose some $J \in \mathcal{B}$ and some $j \in J$. Differentiate both sides of the above equation with respect to $\boldsymbol{z}_j$, which yields:

$$\sum_{B \in \mathcal{B}} \sum_{i \in B} D_i \boldsymbol{f}^{(B)}(\boldsymbol{v}_B(\boldsymbol{z})) D_j \boldsymbol{v}_i(\boldsymbol{z}) = D_j \hat{\boldsymbol{f}}^{(J)}(\boldsymbol{z}_J) . \tag{50}$$

Choose $J' \in \mathcal{B} \setminus \{J\}$ and $j' \in J'$. Differentiating the above w.r.t. $\boldsymbol{z}_{j'}$ yields

$$\sum_{B \in \mathcal{B}} \sum_{i \in B} \left[ D_i \boldsymbol{f}^{(B)}(\boldsymbol{v}_B(\boldsymbol{z})) D_{j,j'}^2 \boldsymbol{v}_i(\boldsymbol{z}) + \sum_{i' \in B} D_{i,i'}^2 \boldsymbol{f}^{(B)}(\boldsymbol{v}_B(\boldsymbol{z})) D_{j'} \boldsymbol{v}_{i'}(\boldsymbol{z}) D_j \boldsymbol{v}_i(\boldsymbol{z}) \right] = 0$$

$$\sum_{B \in \mathcal{B}} \left[ \sum_{i \in B} \left[ D_i \boldsymbol{f}^{(B)}(\boldsymbol{v}_B(\boldsymbol{z})) D_{j,j'}^2 \boldsymbol{v}_i(\boldsymbol{z}) + D_{i,i}^2 \boldsymbol{f}^{(B)}(\boldsymbol{v}_B(\boldsymbol{z})) D_{j'} \boldsymbol{v}_i(\boldsymbol{z}) D_j \boldsymbol{v}_i(\boldsymbol{z}) \right] + \right.$$

$$\left. \sum_{(i,i') \in B_<^2} D_{i,i'}^2 \boldsymbol{f}^{(B)}(\boldsymbol{v}_B(\boldsymbol{z}))(D_{j'} \boldsymbol{v}_{i'}(\boldsymbol{z}) D_j \boldsymbol{v}_i(\boldsymbol{z}) + D_{j'} \boldsymbol{v}_i(\boldsymbol{z}) D_j \boldsymbol{v}_{i'}(\boldsymbol{z})) \right] = 0 , \quad (51)$$

where $B_<^2 := B^2 \cap \{(i, i') \mid i' < i\}$. For the sake of notational conciseness, we are going to refer to $S_{\mathcal{B}}$ and $S_{\mathcal{B}}^c$ as $S$ and $S^c$ (Definition 11). Also, define

$$S_< := \bigcup_{B \in \mathcal{B}} B_<^2 . \tag{52}$$

Let us define the vectors

$$\forall i \in \{1, ...d_z\}, \; \vec{a}_i(\boldsymbol{z}) := (D_{j,j'}^2 \boldsymbol{v}_i(\boldsymbol{z}))_{(j,j') \in S^c} \tag{53}$$

$$\forall i \in \{1, ...d_z\}, \; \vec{b}_i(\boldsymbol{z}) := (D_{j'} \boldsymbol{v}_i(\boldsymbol{z}) D_j \boldsymbol{v}_i(\boldsymbol{z}))_{(j,j') \in S^c} \tag{54}$$

$$\forall B \in \mathcal{B}, \; \forall (i, i') \in B_<^2, \; \vec{c}_{i,i'}(\boldsymbol{z}) := (D_{j'} \boldsymbol{v}_{i'}(\boldsymbol{z}) D_j \boldsymbol{v}_i(\boldsymbol{z}) + D_{j'} \boldsymbol{v}_i(\boldsymbol{z}) D_j \boldsymbol{v}_{i'}(\boldsymbol{z}))_{(j,j') \in S^c} \tag{55}$$

This allows us to rewrite, for all $k \in \{1, ..., d_x\}$

$$\sum_{B \in \mathcal{B}} \left[ \sum_{i \in B} \left[ D_i \boldsymbol{f}_k^{(B)}(\boldsymbol{v}_B(\boldsymbol{z})) \vec{a}_i(\boldsymbol{z}) + D_{i,i}^2 \boldsymbol{f}_k^{(B)}(\boldsymbol{v}_B(\boldsymbol{z})) \vec{b}_i(\boldsymbol{z}) \right] + \sum_{(i,i') \in B_<^2} D_{i,i'}^2 \boldsymbol{f}_k^{(B)}(\boldsymbol{v}_B(\boldsymbol{z})) \vec{c}_{i,i'}(\boldsymbol{z}) \right] = 0 .$$
$$\tag{56}$$

We define

$$\boldsymbol{w}(\boldsymbol{z}, k) := ((D_i \boldsymbol{f}_k^{(B)}(\boldsymbol{z}_B))_{i \in B}, (D_{i,i}^2 \boldsymbol{f}_k^{(B)}(\boldsymbol{z}_B))_{i \in B}, (D_{i,i'}^2 \boldsymbol{f}_k^{(B)}(\boldsymbol{z}_B))_{(i,i') \in B_<^2})_{B \in \mathcal{B}} \tag{57}$$

$$\boldsymbol{M}(\boldsymbol{z}) := [[\vec{a}_i(\boldsymbol{z})]_{i \in B}, [\vec{b}_i(\boldsymbol{z})]_{i \in B}, [\vec{c}_{i,i'}(\boldsymbol{z})]_{(i,i') \in B_<^2}]_{B \in \mathcal{B}} , \tag{58}$$

which allows us to write, for all $k \in \{1, ..., d_z\}$

$$M(z)w(v(z), k) = 0. \tag{59}$$

We can now recognize that the matrix $W(v(z))$ of Assumption 2 is given by

$$W(v(z))^\top = [w(v(z), 1) \ \ldots \ w(v(z), d_x)] \tag{60}$$

which allows us to write

$$M(z)W(v(z))^\top = 0 \tag{61}$$

$$W(v(z))M(z)^\top = 0 \tag{62}$$

Since $W(v(z))$ has full column-rank (by Assumption 2 and the fact that $v(z) \in \mathcal{Z}^{\text{train}}$), there exists $q$ rows that are linearly independent. Let $K$ be the index set of these rows. This means $W(v(z))_{K,\cdot}$ is an invertible matrix. We can thus write

$$W(v(z))_{K,\cdot}M(z)^\top = 0 \tag{63}$$

$$(W(v(z))_{K,\cdot})^{-1}W(v(z))_{K,\cdot}M(z)^\top = (W(v(z))_{K,\cdot})^{-1}0 \tag{64}$$

$$M(z)^\top = 0, \tag{65}$$

which means, in particular, that, $\forall i \in \{1, \ldots, d_z\}, \vec{b}_i(z) = 0$, i.e.,

$$\forall i \in \{1, \ldots, d_z\}, \forall (j, j') \in S^c, D_j v_i(z) D_{j'} v_i(z) = 0 \tag{66}$$

Since the $v$ is a diffeomorphism, its Jacobian matrix $Dv(z)$ is invertible everywhere. By Lemma 5, this means there exists a permutation $\pi$ such that, for all $j$, $D_j v_{\pi(j)}(z) \neq 0$. This and (66) imply that

$$\forall (j, j') \in S^c, \ D_j v_{\pi(j')}(z) \underbrace{D_{j'} v_{\pi(j')}(z)}_{\neq 0} = 0, \tag{67}$$

$$\implies \forall (j, j') \in S^c, \ D_j v_{\pi(j')}(z) = 0. \tag{68}$$

To show that $Dv(z)$ is a $\mathcal{B}$-block permutation matrix, the only thing left to show is that $\pi$ respects $\mathcal{B}$. For this, we use the fact that, $\forall B \in \mathcal{B}, \forall (i, i') \in B^2_<, \vec{c}_{i,i'}(z) = 0$ (recall $M(z) = 0$). Because $\vec{c}_{i,i'}(z) = \vec{c}_{i',i}(z)$, we can write

$$\forall (i, i') \in S \text{ s.t. } i \neq i', \forall (j, j') \in S^c, D_{j'} v_{i'}(z) D_j v_i(z) + D_{j'} v_i(z) D_j v_{i'}(z) = 0. \tag{69}$$

We now show that if $(j, j') \in S^c$ (indices belong to different blocks), then $(\pi(j), \pi(j')) \in S^c$ (they also belong to different blocks). Assume this is false, i.e. there exists $(j_0, j'_0) \in S^c$ such that $(\pi(j_0), \pi(j'_0)) \in S$. Then we can apply (69) (with $i := \pi(j_0)$ and $i' := \pi(j'_0)$) and get

$$\underbrace{D_{j'_0} v_{\pi(j'_0)}(z) D_{j_0} v_{\pi(j_0)}(z)}_{\neq 0} + D_{j'_0} v_{\pi(j_0)}(z) D_{j_0} v_{\pi(j'_0)}(z) = 0, \tag{70}$$

where the left term in the sum is different of 0 because of the definition of $\pi$. This implies that

$$D_{j'_0} v_{\pi(j_0)}(z) D_{j_0} v_{\pi(j'_0)}(z) \neq 0, \tag{71}$$

otherwise (70) cannot hold. But (71) contradicts (68). Thus, we have that,

$$(j, j') \in S^c \implies (\pi(j), \pi(j')) \in S^c. \tag{72}$$

The contraposed is

$$(\pi(j), \pi(j')) \in S \implies (j, j') \in S \tag{73}$$

$$(j, j') \in S \implies (\pi^{-1}(j), \pi^{-1}(j')) \in S. \tag{74}$$

From the above, it is clear that $\pi^{-1}$ respects $\mathcal{B}$ which implies that $\pi$ respects $\mathcal{B}$ (Lemma 10). Thus $Dv(z)$ is a $\mathcal{B}$-block permutation matrix. $\qquad\square$

**Lemma 10** (B-respecting permutations form a group). *Let $\mathcal{B}$ be a partition of $\{1, \ldots, d_z\}$ and let $\pi$ and $\bar{\pi}$ be a permutation of $\{1, \ldots, d_z\}$ that respect $\mathcal{B}$. The following holds:*

1. *The identity permutation $e$ respects $\mathcal{B}$.*

2. *The composition $\pi \circ \bar{\pi}$ respects $\mathcal{B}$.*

3. *The inverse permutation $\pi^{-1}$ respects $\mathcal{B}$.*

*Proof.* The first statement is trivial, since for all $B \in \mathcal{B}$, $e(B) = B \in \mathcal{B}$.

The second statement follows since for all $B \in \mathcal{B}$, $\bar{\pi}(B) \in \mathcal{B}$ and thus $\pi(\bar{\pi}(B)) \in \mathcal{B}$.

We now prove the third statement. Let $B \in \mathcal{B}$. Since $\pi$ is surjective and respects $\mathcal{B}$, there exists a $B' \in \mathcal{B}$ such that $\pi(B') = B$. Thus, $\pi^{-1}(B) = \pi^{-1}(\pi(B')) = B' \in \mathcal{B}$. $\qquad\square$

### A.6 Sufficient nonlinearity v.s. sufficient variability in nonlinear ICA with auxiliary variables

In Section 3.1, we introduced the "sufficient nonlinearity" condition (Assumption 2) and highlighted its resemblance to the "sufficient variability" assumptions often found in the nonlinear ICA literature [30, 31, 33, 36, 37, 42, 73]. We now clarify this connection. To make the discussion more concrete, we consider the sufficient variability assumption found in Hyvärinen et al. [33]. In this work, the latent variable $\boldsymbol{z}$ is assumed to be distributed according to

$$p(\boldsymbol{z} \mid \boldsymbol{u}) := \prod_{i=1}^{d_z} p_i(\boldsymbol{z}_i \mid \boldsymbol{u}) \,. \tag{75}$$

In other words, the latent factors $\boldsymbol{z}_i$ are mutually conditionally independent given an observed auxiliary variable $\boldsymbol{u}$. Define

$$\boldsymbol{w}(\boldsymbol{z}, \boldsymbol{u}) := \left( \left( \frac{\partial}{\partial \boldsymbol{z}_i} \log p_i(\boldsymbol{z}_i \mid \boldsymbol{u}) \right)_{i \in [d_z]} \left( \frac{\partial^2}{\partial \boldsymbol{z}_i^2} \log p_i(\boldsymbol{z}_i \mid \boldsymbol{u}) \right)_{i \in [d_z]} \right) \in \mathbb{R}^{2d_z} \,. \tag{76}$$

We now recall the assumption of sufficient variability of Hyvärinen et al. [33]:

**Assumption 3** (Assumption of variability from Hyvärinen et al. [33, Theorem 1]). *For any $\boldsymbol{z} \in \mathbb{R}^{d_z}$, there exists $2d_z + 1$ values of $\boldsymbol{u}$, denoted by $\boldsymbol{u}^{(0)}, \boldsymbol{u}^{(1)}, \ldots, \boldsymbol{u}^{(2d_z)}$ such that the $2d_z$ vectors*

$$\boldsymbol{w}(\boldsymbol{z}, \boldsymbol{u}^{(1)}) - \boldsymbol{w}(\boldsymbol{z}, \boldsymbol{u}^{(0)}), \ldots, \boldsymbol{w}(\boldsymbol{z}, \boldsymbol{u}^{(2d_z)}) - \boldsymbol{w}(\boldsymbol{z}, \boldsymbol{u}^{(0)}) \tag{77}$$

*are linearly independent.*

To emphasize the resemblance with our assumption of sufficient nonlinearity, we rewrite it in the special case where the partition $\mathcal{B} := \{\{1\}, \ldots, \{d_z\}\}$. Note that, in that case, $q := d_z + \sum_{B \in \mathcal{B}} \frac{|B|(|B|+1)}{2} = 2d_z$.

**Assumption 4** (Sufficient nonlinearity (trivial partition)). *For all $\boldsymbol{z} \in \mathcal{Z}^{\text{train}}$, $\boldsymbol{f}$ is such that the following matrix has independent columns (i.e. full column-rank):*

$$\boldsymbol{W}(\boldsymbol{z}) := \left[ \left[ D_i \boldsymbol{f}^{(i)}(\boldsymbol{z}_i) \right]_{i \in [d_z]} \left[ D_{i,i}^2 \boldsymbol{f}^{(i)}(\boldsymbol{z}_i) \right]_{i \in [d_z]} \right] \in \mathbb{R}^{d_x \times 2d_z} \,. \tag{78}$$

One can already see the resemblance between Assumptions 3 & 4, e.g. both have something to do with first and second derivatives. To make the connection even more explicit, define $\boldsymbol{w}(\boldsymbol{z}, k)$ to be the $k$th row of $\boldsymbol{W}(\boldsymbol{z})$ (do not conflate with $\boldsymbol{w}(\boldsymbol{z}, \boldsymbol{u})$). Also, recall the basic fact from linear algebra that the column-rank is always equal to the row-rank. This means that $\boldsymbol{W}(\boldsymbol{z})$ is full column-rank if and only if there exists $k_1, \ldots, k_{2d_z} \in [d_x]$ such that the vectors $\boldsymbol{w}(\boldsymbol{z}, k_1), \ldots, \boldsymbol{w}(\boldsymbol{z}, k_{2d_z})$ are linearly independent. It is then easy to see the correspondance between $\boldsymbol{w}(\boldsymbol{z}, k)$ and $\boldsymbol{w}(\boldsymbol{z}, \boldsymbol{u}) - \boldsymbol{w}(\boldsymbol{z}, \boldsymbol{u}^{(0)})$ (from Assumption 3) and between the pixel index $k \in [d_x]$ and the auxiliary variable $\boldsymbol{u}$.

We now look at why Assumption 2 is likely to be satisfied when $d_x >> d_z$. Informally, one can see that when $d_x$ is much larger than $2d_z$, the matrix $\boldsymbol{W}(\boldsymbol{z})$ has much more rows than columns and thus it becomes more likely that we will find $2d_z$ rows that are linearly independent, thus satisfying Assumption 2.

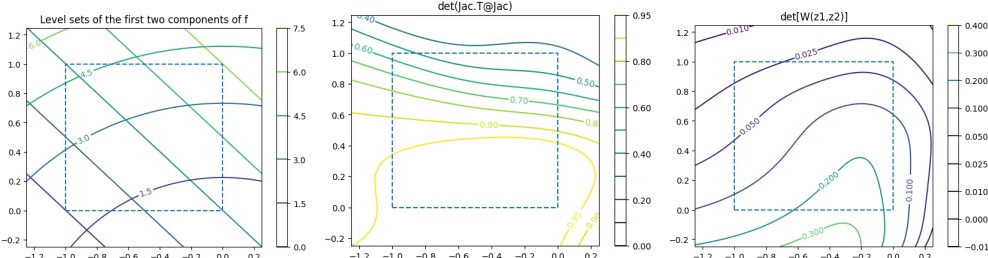

Figure 7: Numerical verification that $\boldsymbol{f} : [-1, 0] \times [0, 1] \to \mathbb{R}^4$ from Example 8 is injective (**left**), has a full rank Jacobian (**middle**) and satisfies Assumption 2 (**right**). The **left** figure shows that $\boldsymbol{f}$ is injective on the square $[-1, 0] \times [0, 1]$ since one can recover $\boldsymbol{z}$ uniquely by knowing the values of $\boldsymbol{f}_1(\boldsymbol{z})$ and $\boldsymbol{f}_2(\boldsymbol{z})$, i.e. knowing the level sets. The **middle** figure reports the $\det(D\boldsymbol{f}(\boldsymbol{z})^\top D\boldsymbol{f}(\boldsymbol{z}))$ (columns of the Jacobian are normalized to have norm 1) and shows that it is nonzero in the square $[-1, 0] \times [0, 1]$, which means the Jacobian is full rank. The **right** figure shows the determinant of the matrix $\boldsymbol{W}(\boldsymbol{z})$ (from Assumption 2, but with normalized columns), we can see that it is nonzero everywhere on the square $[-1, 0] \times [0, 1]$. We normalized the columns of $D\boldsymbol{f}$ and $\boldsymbol{W}$ so that the determinant is between 0 and 1.

## A.7  Examples of sufficiently nonlinear additive decoders

**Example 8** (A sufficiently nonlinear $\boldsymbol{f}$ - Example 3 continued). *Consider the additive function*

$$\boldsymbol{f}(\boldsymbol{z}) := \begin{bmatrix} z_1 \\ z_1^2 \\ z_1^3 \\ z_1^4 \end{bmatrix} + \begin{bmatrix} (z_2 + 1) \\ (z_2 + 1)^2 \\ (z_2 + 1)^3 \\ (z_2 + 1)^4 \end{bmatrix} . \tag{79}$$

*We will provide a numerical verification that this function is a diffeomorphism from the square $[-1, 0] \times [0, 1]$ to its image that satisfies Assumption 2.*

*The Jacobian of $\boldsymbol{f}$ is given by*

$$D\boldsymbol{f}(\boldsymbol{z}) = \begin{bmatrix} 1 & 1 \\ 2z_1 & 2(z_2 + 1) \\ 3z_1^2 & 3(z_2 + 1)^2 \\ 4z_1^3 & 4(z_2 + 1)^3 \end{bmatrix} , \tag{80}$$

*and the matrix $\boldsymbol{W}(\boldsymbol{z})$ from Assumption 2 is given by*

$$\boldsymbol{W}(\boldsymbol{z}) = \begin{bmatrix} 1 & 0 & 1 & 0 \\ 2z_1 & 2 & 2(z_2 + 1) & 2 \\ 3z_1^2 & 6z_1 & 3(z_2 + 1)^2 & 6(z_2 + 1) \\ 4z_1^3 & 12z_1^2 & 4(z_2 + 1)^3 & 12(z_2 + 1)^2 \end{bmatrix} . \tag{81}$$

*Figure 7 presents a numerical verification that $\boldsymbol{f}$ is injective, has a full rank Jacobian and satisfies Assumption 2. Injective $\boldsymbol{f}$ with full rank Jacobian is enough to conclude that $\boldsymbol{f}$ is a diffeomorphism onto its image.*

**Example 9** (Smooth balls dataset is sufficiently nonlinear - Example 4 continued). *We implemented a ground-truth additive decoder $\boldsymbol{f} : [0, 5]^2 \to \mathbb{R}^{64*64*3}$ which maps to 64x64 RGB images consisting of two colored balls where $z_1$ and $z_2$ control their respective heights (Figure 8a). The analytical form of $\boldsymbol{f}$ can be found in our code base. The decoder $\boldsymbol{f}$ is implemented in JAX [6] which allows for its automatic differentiation to compute $D\boldsymbol{f}$ and $D^2\boldsymbol{f}$ (Figures 8b & 8c). This allows us to verify numerically that $\boldsymbol{f}$ is sufficiently nonlinear (Assumption 2). Recall that this assumption requires that $\boldsymbol{W}(\boldsymbol{z})$ (defined in Assumption 2) has independent columns everywhere. To test this, we compute $Vol(\boldsymbol{z}) := \sqrt{|\det(\boldsymbol{W}(\boldsymbol{z})^\top \boldsymbol{W}(\boldsymbol{z}))|}$ over a grid of values of $\boldsymbol{z}$ and verify that $Vol(\boldsymbol{z}) > 0$ everywhere (Figure 8d). Note that $Vol(\boldsymbol{z})$ corresponds to the 4D volume of the parallelepiped embedded in $\mathbb{R}^{64*64*3}$ spanned by the four columns of $\boldsymbol{W}(\boldsymbol{z})$. This volume is $> 0$ if and only if the columns are linearly independent. Note that we normalize the columns of $\boldsymbol{W}(\boldsymbol{z})$ so that they have a norm*

*of one. It follows that Vol($z$) is between $0$ and $1$ where $1$ means the vectors are orthogonal, i.e. maximally independent. The minimal value of Vol($z$) over the domain of $f$ is $\approx 0.97$, indicating that Assumption 2 holds.*

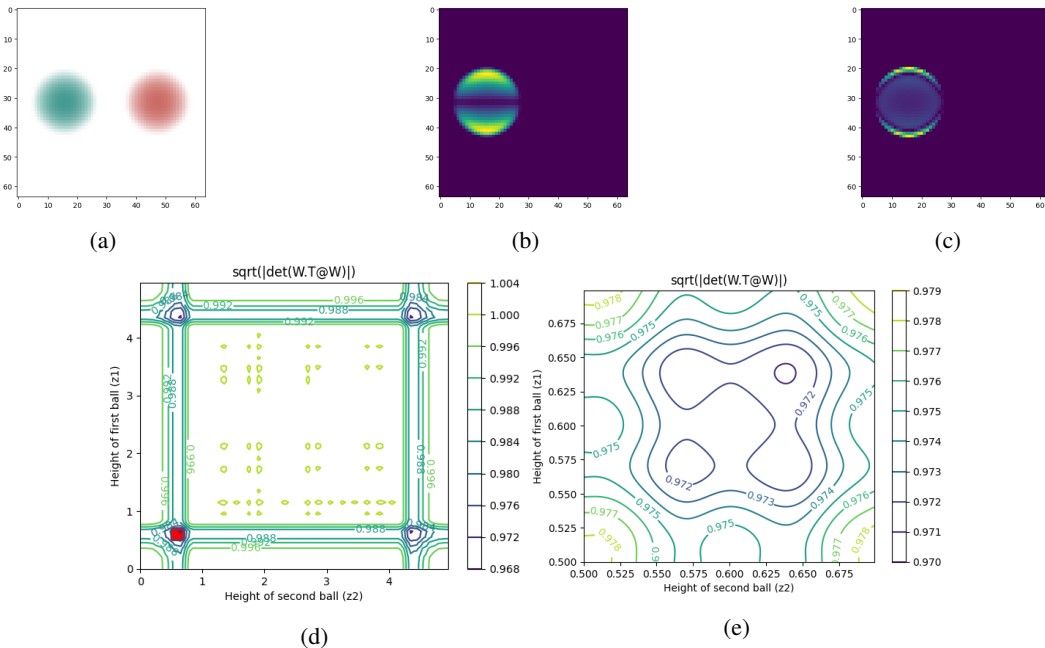

Figure 8: Figure (a) shows an image the synthetic dataset of Example 9. Figure (b) shows the derivative of the image w.r.t. $z_1$ (the height of the left ball) where the color intensity of each pixel corresponds to the Euclidean norm along the RGB axis. Figure (c) similarly shows the second derivative of the image w.r.t. $z_1$. Figure (d) is a contour plot of the function $\sqrt{|\det(W(z)^\top W(z))|}$ where $W(z)$ is defined in Assumption 2 (here columns are normalized to have unit norm). The smallest value of $\sqrt{|\det(W(z)^\top W(z))|}$ across domain is $\approx 0.97$, indicating that Assumption 2 is satisfied. See Example 9 and code for details. Figure 8e is a higher resolution rendering of the red region of Figure 8d (to make sure there is no singularity there).

## A.8 Proof of Theorem 2

We start with a simple definition:

**Definition 15** ($\mathcal{B}$-block permutation matrices)**.** *A matrix $A \in \mathbb{R}^{d \times d}$ is a $\mathcal{B}$-block permutation matrix if it is invertible and can be written as $A = CP_\pi$ where $P_\pi$ is the matrix representing the $\mathcal{B}$-respecting permutation $\pi$ ($P_\pi e_i = e_{\pi(i)}$) and $C \in \mathbb{R}^{d \times d}_{S_\mathcal{B}}$ (See Definitions 10 & 11).*

The following technical lemma leverages continuity and path-connectedness to show that the block-permutation structure must remain the same across the whole domain. It can be skipped at first read.

**Lemma 11.** *Let $\mathcal{C}$ be a connected topological space and let $M : \mathcal{C} \to \mathbb{R}^{d \times d}$ be a continuous function. Suppose that, for all $c \in \mathcal{C}$, $M(c)$ is an invertible $\mathcal{B}$-block permutation matrix (Definition 15). Then, there exists a $\mathcal{B}$-respecting permutation $\pi$ such that for all $c \in \mathcal{C}$ and all distinct $B, B' \in \mathcal{B}$, $M(c)_{\pi(B'),B} = 0$.*

*Proof.* The reason this result is not trivial, is that, even if $M(c)$ is a $\mathcal{B}$-block permutation for all $c$, the permutation might change for different $c$. The goal of this lemma is to show that, if $\mathcal{C}$ is connected and the map $M(\cdot)$ is continuous, then one can find a single permutation that works for all $c \in \mathcal{C}$.

First, since $\mathcal{C}$ is connected and $M$ is continuous, its image, $M(\mathcal{C})$, must be connected (by [56, Theorem 23.5]).

Second, from the hypothesis of the lemma, we know that

$$M(\mathcal{C}) \subseteq \mathcal{A} := \left( \bigcup_{\pi \in \mathfrak{S}(\mathcal{B})} \mathbb{R}^{d \times d}_{S_{\mathcal{B}}} P_\pi \right) \setminus \{\text{singular matrices}\}, \tag{82}$$

where $\mathfrak{S}(\mathcal{B})$ is the set of $\mathcal{B}$-respecting permutations and $\mathbb{R}^{d \times d}_{S_{\mathcal{B}}} P_\pi = \{M P_\pi \mid M \in \mathbb{R}^{d \times d}_{S_{\mathcal{B}}}\}$. We can rewrite the set $\mathcal{A}$ above as

$$\mathcal{A} = \bigcup_{\pi \in \mathfrak{S}(\mathcal{B})} \left( \mathbb{R}^{d \times d}_{S_{\mathcal{B}}} P_\pi \setminus \{\text{singular matrices}\} \right), \tag{83}$$

We now define an equivalence relation $\sim$ over $\mathcal{B}$-respecting permutation: $\pi \sim \pi'$ iff for all $B \in \mathcal{B}$, $\pi(B) = \pi'(B)$. In other words, two $\mathcal{B}$-respecting permutations are equivalent if they send every block to the same block (note that they can permute elements of a given block differently). We notice that

$$\pi \sim \pi' \implies \mathbb{R}^{d \times d}_{S_{\mathcal{B}}} P_\pi = \mathbb{R}^{d \times d}_{S_{\mathcal{B}}} P_{\pi'}. \tag{84}$$

Let $\mathfrak{S}(\mathcal{B})/\sim$ be the set of equivalence classes induce by $\sim$ and let $\Pi$ stand for one such equivalence class. Thanks to (84), we can define, for all $\Pi \in \mathfrak{S}(\mathcal{B})/\sim$, the following set:

$$V_\Pi := \mathbb{R}^{d \times d}_{S_{\mathcal{B}}} P_\pi \setminus \{\text{singular matrices}\}, \text{ for some } \pi \in \Pi, \tag{85}$$

where the specific choice of $\pi \in \Pi$ is arbitrary (any $\pi' \in \Pi$ would yield the same definition, by (84)). This construction allows us to write

$$\mathcal{A} = \bigcup_{\Pi \in \mathfrak{S}(\mathcal{B})/\sim} V_\Pi, \tag{86}$$

We now show that $\{V_\Pi\}_{\Pi \in \mathfrak{S}(\mathcal{B})/\sim}$ forms a partition of $\mathcal{A}$. Choose two distinct equivalence classes of permutations $\Pi$ and $\Pi'$ and let $\pi \in \Pi$ and $\pi' \in \Pi'$ be representatives. We note that

$$\mathbb{R}^{d \times d}_{S_{\mathcal{B}}} P_\pi \cap \mathbb{R}^{d \times d}_{S_{\mathcal{B}}} P_{\pi'} \subseteq \{\text{singular matrices}\}, \tag{87}$$

since any matrix that is both in $\mathbb{R}^{d \times d}_{S_{\mathcal{B}}} P_\pi$ and $\mathbb{R}^{d \times d}_{S_{\mathcal{B}}} P_{\pi'}$ must have at least one row filled with zeros. This implies that

$$V_\Pi \cap V_{\Pi'} = \emptyset, \tag{88}$$

which shows that $\{V_\Pi\}_{\Pi \in \mathfrak{S}(\mathcal{B})/\sim}$ is indeed a partition of $\mathcal{A}$.

Each $V_\Pi$ is closed in $\mathcal{A}$ (wrt the relative topology) since

$$V_\Pi = \mathbb{R}^{d \times d}_{S_{\mathcal{B}}} P_\pi \setminus \{\text{singular matrices}\} = \mathcal{A} \cap \underbrace{\mathbb{R}^{d \times d}_{S_{\mathcal{B}}} P_\pi}_{\text{closed in } \mathbb{R}^{d \times d}}. \tag{89}$$

Moreover, $V_\Pi$ is open in $\mathcal{A}$, since

$$V_\Pi = \mathcal{A} \setminus \underbrace{\bigcup_{\Pi' \neq \Pi} V_{\Pi'}}_{\text{closed in } \mathcal{A}}. \tag{90}$$

Thus, for any $\Pi \in \mathfrak{S}(\mathcal{B})/\sim$, the sets $V_\Pi$ and $\bigcup_{\Pi' \neq \Pi} V_{\Pi'}$ forms a *separation* (see [56, Section 23]). Since $M(\mathcal{C})$ is a connected subset of $\mathcal{A}$, it must lie completely in $V_\Pi$ or $\bigcup_{\Pi' \neq \Pi} V_{\Pi'}$, by [56, Lemma 23.2]. Since this is true for all $\Pi$, it must follow that there exists a $\Pi^*$ such that $M(\mathcal{C}) \subseteq V_{\Pi^*}$, which completes the proof. $\qquad\square$

**Theorem 2** (From local to global disentanglement). *Suppose that all the assumptions of Theorem 1 hold. Additionally, assume $\mathcal{Z}^{\text{train}}$ is path-connected (Definition 8) and that the block-specific decoders $\boldsymbol{f}^{(B)}$ and $\hat{\boldsymbol{f}}^{(B)}$ are injective for all blocks $B \in \mathcal{B}$. Then, if $\hat{\boldsymbol{f}}$ and $\hat{\boldsymbol{g}}$ solve the reconstruction problem on the training distribution, i.e. $\mathbb{E}^{\text{train}}||\boldsymbol{x} - \hat{\boldsymbol{f}}(\hat{\boldsymbol{g}}(\boldsymbol{x}))||^2 = 0$, we have that $\hat{\boldsymbol{f}}$ is (globally) $\mathcal{B}$-disentangled w.r.t. $\boldsymbol{f}$ (Definition 3) and, for all $B \in \mathcal{B}$,*

$$\hat{\boldsymbol{f}}^{(B)}(\boldsymbol{z}_B) = \boldsymbol{f}^{(\pi(B))}(\bar{\boldsymbol{v}}_{\pi(B)}(\boldsymbol{z}_B)) + \boldsymbol{c}^{(B)}, \text{for all } \boldsymbol{z}_B \in \hat{\mathcal{Z}}^{\text{train}}_B, \tag{8}$$

*where the functions $\bar{v}_{\pi(B)}$ are from Defintion 3 and the vectors $c^{(B)} \in \mathbb{R}^{d_x}$ are constants such that $\sum_{B \in \mathcal{B}} c^{(B)} = 0$. We also have that the functions $\bar{v}_{\pi(B)} : \hat{\mathcal{Z}}_B^{\mathrm{train}} \to \mathcal{Z}_{\pi(B)}^{\mathrm{train}}$ are $C^2$-diffeomorphisms and have the following form:*

$$\bar{v}_{\pi(B)}(z_B) = (f^{\pi(B)})^{-1}(\hat{f}^{(B)}(z_B) - c^{(B)}), \text{ for all } z_B \in \hat{\mathcal{Z}}_B^{\mathrm{train}}. \tag{9}$$

*Proof.* **Step 1 - Showing the permutation $\pi$ does not change for different $z$.** Theorem 1 showed local $\mathcal{B}$-disentanglement, i.e. for all $z \in \hat{\mathcal{Z}}^{\mathrm{train}}$, $Dv(z)$ has a $\mathcal{B}$-block permutation structure. The first step towards showing global disentanglement is to show that this block structure is the same for all $z \in \hat{\mathcal{Z}}^{\mathrm{train}}$ (*a priori*, $\pi$ could be different for different $z$). Since $v$ is $C^2$, its Jacobian $Dv(z)$ is continuous. Since $\mathcal{Z}^{\mathrm{train}}$ is path-connected, $\hat{\mathcal{Z}}^{\mathrm{train}}$ must also be since both sets are diffeomorphic. By Lemma 11, this means the $\mathcal{B}$-block permutation structure of $Dv(z)$ is the same for all $z \in \hat{\mathcal{Z}}^{\mathrm{train}}$ (implicitly using the fact that path-connected implies connected). In other words, there exists a permutation $\pi$ respecting $\mathcal{B}$ such that, for all $z \in \hat{\mathcal{Z}}^{\mathrm{train}}$ and all distinct $B, B' \in \mathcal{B}$, $D_B v_{\pi(B')}(z) = 0$.

**Step 2 - Linking object-specific decoders.** We now show that, for all $B \in \mathcal{B}$, $\hat{f}^{(B)}(z_B) = f^{(\pi(B))}(v_{\pi(B)}(z)) + c^{(B)}$ for all $z \in \hat{\mathcal{Z}}^{\mathrm{train}}$. To do this, we rewrite (50) as

$$D\hat{f}^{(J)}(z_J) = \sum_{B \in \mathcal{B}} Df^{(B)}(v_B(z))D_J v_B(z), \tag{91}$$

but because $B \neq \pi(J) \implies D_J v_B(z) = 0$ (block-permutation structure), we get

$$D\hat{f}^{(J)}(z_J) = Df^{(\pi(J))}(v_{\pi(J)}(z))D_J v_{\pi(J)}(z). \tag{92}$$

The above holds for all $J \in \mathcal{B}$. We simply change $J$ by $B$ in the following equation.

$$D\hat{f}^{(B)}(z_B) = Df^{(\pi(B))}(v_{\pi(B)}(z))D_B v_{\pi(B)}(z). \tag{93}$$

Now notice that the r.h.s. of the above equation is equal to $D(f^{(\pi(B))} \circ v_{\pi(B)})$. We can thus write

$$D\hat{f}^{(B)}(z_B) = D(f^{(\pi(B))} \circ v_{\pi(B)})(z), \text{ for all } z \in \hat{\mathcal{Z}}^{\mathrm{train}}. \tag{94}$$

Now choose distinct $z, z^0 \in \hat{\mathcal{Z}}^{\mathrm{train}}$. Since $\mathcal{Z}^{\mathrm{train}}$ is path-connected, $\hat{\mathcal{Z}}^{\mathrm{train}}$ also is since they are diffeomorphic. Hence, there exists a continuously differentiable function $\phi : [0,1] \to \hat{\mathcal{Z}}^{\mathrm{train}}$ such that $\phi(0) = z^0$ and $\phi(1) = z$. We can now use (94) together with the gradient theorem, a.k.a. the fundamental theorem of calculus for line integrals, to show the following

$$\int_0^1 D\hat{f}^{(B)}(\phi_B(z)) \cdot \phi_B(t)dt = \int_0^1 D(f^{(\pi(B))} \circ v_{\pi(B)})(\phi(z)) \cdot \phi(t)dt \tag{95}$$

$$\hat{f}^{(B)}(z_B) - \hat{f}^{(B)}(z_B^0) = f^{(\pi(B))} \circ v_{\pi(B)}(z) - f^{(\pi(B))} \circ v_{\pi(B)}(z^0) \tag{96}$$

$$\hat{f}^{(B)}(z_B) = f^{(\pi(B))} \circ v_{\pi(B)}(z) + \underbrace{(\hat{f}^{(B)}(z_B^0) - f^{(\pi(B))} \circ v_{\pi(B)}(z^0))}_{\text{constant in } z} \tag{97}$$

$$\hat{f}^{(B)}(z_B) = f^{(\pi(B))} \circ v_{\pi(B)}(z) + c^{(B)}, \tag{98}$$

which holds for all $z \in \hat{\mathcal{Z}}^{\mathrm{train}}$.

We now show that $\sum_{B \in \mathcal{B}} c^{(B)} = 0$. Take some $z^0 \in \hat{\mathcal{Z}}^{\mathrm{train}}$. Equations (49) & (98) tell us that

$$\sum_{B \in \mathcal{B}} f^{(B)}(v_B(z^0)) = \sum_{B \in \mathcal{B}} \hat{f}^{(B)}(z_B^0) \tag{99}$$

$$= \sum_{B \in \mathcal{B}} f^{(\pi(B))}(v_{\pi(B)}(z^0)) + \sum_{B \in \mathcal{B}} c^{(B)} \tag{100}$$

$$= \sum_{B \in \mathcal{B}} f^{(B)}(v_B(z^0)) + \sum_{B \in \mathcal{B}} c^{(B)} \tag{101}$$

$$\implies 0 = \sum_{B \in \mathcal{B}} c^{(B)} \tag{102}$$

**Step 3 - From local to global disentanglement.** By assumption, the functions $\boldsymbol{f}^{(B)} : \mathcal{Z}^{\text{train}}_B \to \mathbb{R}^{d_x}$ are injective. This will allow us to show that $\boldsymbol{v}_{\pi(B)}(\boldsymbol{z})$ depends only on $\boldsymbol{z}_B$. We proceed by contradiction. Suppose there exists $(\boldsymbol{z}_B, \boldsymbol{z}_{B^c}) \in \hat{\mathcal{Z}}^{\text{train}}$ and $\boldsymbol{z}^0_{B^c}$ such that $(\boldsymbol{z}_B, \boldsymbol{z}^0_{B^c}) \in \hat{\mathcal{Z}}^{\text{train}}$ and $\boldsymbol{v}_{\pi(B)}(\boldsymbol{z}_B, \boldsymbol{z}_{B^c}) \neq \boldsymbol{v}_{\pi(B)}(\boldsymbol{z}_B, \boldsymbol{z}^0_{B^c})$. This means

$$\boldsymbol{f}^{(\pi(B))} \circ \boldsymbol{v}_{\pi(B)}(\boldsymbol{z}_B, \boldsymbol{z}_{B^c}) + \boldsymbol{c}^{(B)} = \hat{\boldsymbol{f}}^{(B)}(\boldsymbol{z}_B) = \boldsymbol{f}^{(\pi(B))} \circ \boldsymbol{v}_{\pi(B)}(\boldsymbol{z}_B, \boldsymbol{z}^0_{B^c}) + \boldsymbol{c}^{(B)}$$

$$\boldsymbol{f}^{(\pi(B))}(\boldsymbol{v}_{\pi(B)}(\boldsymbol{z}_B, \boldsymbol{z}_B)) = \boldsymbol{f}^{(\pi(B))}(\boldsymbol{v}_{\pi(B)}(\boldsymbol{z}_B, \boldsymbol{z}^0_B))$$

which is a contradiction with the fact that $\boldsymbol{f}^{(\pi(B))}$ is injective. Hence, $\boldsymbol{v}_{\pi(B)}(\boldsymbol{z})$ depends only on $\boldsymbol{z}_B$. We also get an explicit form for $\boldsymbol{v}_{\pi(B)}$:

$$(\boldsymbol{f}^{\pi(B)})^{-1}(\hat{\boldsymbol{f}}^{(B)}(\boldsymbol{z}_B) - \boldsymbol{c}^{(B)}) = \boldsymbol{v}_{\pi(B)}(\boldsymbol{z}) \text{ for all } \boldsymbol{z} \in \mathcal{Z}^{\text{train}} . \tag{103}$$

We define the map $\bar{\boldsymbol{v}}_{\pi(B)}(\boldsymbol{z}_B) := (\boldsymbol{f}^{\pi(B)})^{-1}(\hat{\boldsymbol{f}}^{(B)}(\boldsymbol{z}_B) - \boldsymbol{c}^{(B)})$ which is from $\hat{\mathcal{Z}}^{\text{train}}_B$ to $\mathcal{Z}^{\text{train}}_{\pi(B)}$. This allows us to rewrite (98) as

$$\hat{\boldsymbol{f}}^{(B)}(\boldsymbol{z}_B) = \boldsymbol{f}^{(\pi(B))} \circ \bar{\boldsymbol{v}}_{\pi(B)}(\boldsymbol{z}_B) + \boldsymbol{c}^{(B)} , \text{ for all } \boldsymbol{z}_B \in \mathcal{Z}^{\text{train}}_B . \tag{104}$$

Because $\hat{\boldsymbol{f}}^{(B)}$ is also injective, we must have that $\bar{\boldsymbol{v}}_{\pi(B)} : \hat{\mathcal{Z}}^{\text{train}}_B \to \mathcal{Z}^{\text{train}}_{\pi(B)}$ is injective as well.

We now show that $\bar{\boldsymbol{v}}_{\pi(B)}$ is surjective. Choose some $\boldsymbol{z}_{\pi(B)} \in \mathcal{Z}^{\text{train}}_{\pi(B)}$. We can always find $\boldsymbol{z}_{\pi(B)^c}$ such that $(\boldsymbol{z}_{\pi(B)}, \boldsymbol{z}_{\pi(B)^c}) \in \mathcal{Z}^{\text{train}}$. Because $\boldsymbol{v} : \hat{\mathcal{Z}}^{\text{train}} \to \mathcal{Z}^{\text{train}}$ is surjective (it is a diffeomorphism), there exists a $\boldsymbol{z}^0 \in \hat{\mathcal{Z}}^{\text{train}}$ such that $\boldsymbol{v}(\boldsymbol{z}^0) = (\boldsymbol{z}_{\pi(B)}, \boldsymbol{z}_{\pi(B)^c})$. By (103), we have that

$$\bar{\boldsymbol{v}}_{\pi(B)}(\boldsymbol{z}^0_B) = \boldsymbol{v}_{\pi(B)}(\boldsymbol{z}^0) . \tag{105}$$

which means $\bar{\boldsymbol{v}}_{\pi(B)}(\boldsymbol{z}^0_B) = \boldsymbol{z}_{\pi(B)}$.

We thus have that $\bar{\boldsymbol{v}}_{\pi(B)}$ is bijective. It is a diffeomorphism because

$$\det D\bar{\boldsymbol{v}}_{\pi(B)}(\boldsymbol{z}_B) = \det D_B \boldsymbol{v}_{\pi(B)}(\boldsymbol{z}) \neq 0 \ \forall \boldsymbol{z} \in \hat{\mathcal{Z}}^{\text{train}} \tag{106}$$

where the first equality holds by (103) and the second holds because $\boldsymbol{v}$ is a diffeomorphism and has block-permutation structure, which means it has a nonzero determinant everywhere on $\hat{\mathcal{Z}}^{\text{train}}$ and is equal to the product of the determinants of its blocks, which implies each block $D_B \boldsymbol{v}_{\pi(B)}$ must have nonzero determinant everywhere.

Since $\bar{\boldsymbol{v}}_{\pi(B)} : \hat{\mathcal{Z}}^{\text{train}}_B \to \mathcal{Z}^{\text{train}}_{\pi(B)}$ bijective and has invertible Jacobian everywhere, it must be a diffeomorphism. $\qquad\square$

### A.9 Injectivity of object-specific decoders v.s. injectivity of their sum

We want to explore the relationship between the injectivity of individual object-specific decoders $\boldsymbol{f}^{(B)}$ and the injectivity of their sum, i.e. $\sum_{B \in \mathcal{B}} \boldsymbol{f}^{(B)}$.

We first show the simple fact that having each $\boldsymbol{f}^{(B)}$ injective is not sufficient to have $\sum_{B \in \mathcal{B}} \boldsymbol{f}^{(B)}$ injective. Take $\boldsymbol{f}^{(B)}(\boldsymbol{z}_B) = \boldsymbol{W}^{(B)} \boldsymbol{z}_B$ where $\boldsymbol{W}^{(B)} \in \mathbb{R}^{d_x \times |B|}$ has full column-rank for all $B \in \mathcal{B}$. We have that

$$\sum_{B \in \mathcal{B}} \boldsymbol{f}^{(B)}(\boldsymbol{z}_B) = \sum_{B \in \mathcal{B}} \boldsymbol{W}^{(B)} \boldsymbol{z}_B = [\boldsymbol{W}^{(B_1)} \ \cdots \ \boldsymbol{W}^{(B_\ell)}]\boldsymbol{z} , \tag{107}$$

where it is clear that the matrix $[\boldsymbol{W}^{(B_1)} \ \cdots \ \boldsymbol{W}^{(B_\ell)}] \in \mathbb{R}^{d_x \times d_z}$ is not necessarily injective even if each $\boldsymbol{W}^{(B)}$ is. This is the case, for instance, if all $\boldsymbol{W}^{(B)}$ have the same image.

We now provide conditions such that $\sum_{B \in \mathcal{B}} \boldsymbol{f}^{(B)}$ injective implies each $\boldsymbol{f}^{(B)}$ injective. We start with a simple lemma:

**Lemma 12.** *If $g \circ h$ is injective, then $h$ is injective.*

*Proof.* By contradiction, assume that $h$ is not injective. Then, there exists distinct $x_1, x_2 \in \text{Dom}(h)$ such that $h(x_1) = h(x_2)$. This implies $g \circ h(x_1) = g \circ h(x_2)$, which violates injectivity of $g \circ h$. $\qquad\square$

The following Lemma provides a condition on the domain of the function $\sum_{B\in\mathcal{B}} \boldsymbol{f}^{(B)}$, $\mathcal{Z}^{\text{train}}$, so that its injectivity implies injectivity of the functions $\boldsymbol{f}^{(B)}$.

**Lemma 13.** *Assume that, for all $B \in \mathcal{B}$ and for all distinct $\boldsymbol{z}_B, \boldsymbol{z}'_B \in \mathcal{Z}_B^{\text{train}}$, there exists $\boldsymbol{z}_{B^c}$ such that $(\boldsymbol{z}_B, \boldsymbol{z}_{B^c}), (\boldsymbol{z}'_B, \boldsymbol{z}_{B^c}) \in \mathcal{Z}^{\text{train}}$. Then, whenever $\sum_{B\in\mathcal{B}} \boldsymbol{f}^{(B)}$ is injective, each $\boldsymbol{f}^{(B)}$ must be injective.*

*Proof.* Notice that $\boldsymbol{f}(\boldsymbol{z}) := \sum_{B\in\mathcal{B}} \boldsymbol{f}^{(B)}(\boldsymbol{z}_B)$ can be written as $\boldsymbol{f} := \text{SumBlocks} \circ \bar{\boldsymbol{f}}(\boldsymbol{z})$ where

$$\bar{\boldsymbol{f}}(\boldsymbol{z}) := \begin{bmatrix} \boldsymbol{f}^{(B_1)}(\boldsymbol{z}_{B_1}) \\ \vdots \\ \boldsymbol{f}^{(B_\ell)}(\boldsymbol{z}_{B_\ell}) \end{bmatrix}, \text{ and } \text{SumBlocks}(\boldsymbol{x}^{(B_1)},\dots,\boldsymbol{x}^{(B_\ell)}) := \sum_{B\in\mathcal{B}} \boldsymbol{x}^{(B)} \tag{108}$$

Since $\boldsymbol{f}$ is injective, by Lemma 12 $\bar{\boldsymbol{f}}$ must be injective.

We now show that each $\boldsymbol{f}^{(B)}$ must also be injective. Take $\boldsymbol{z}_B, \boldsymbol{z}'_B \in \mathcal{Z}_B^{\text{train}}$ such that $\boldsymbol{f}^{(B)}(\boldsymbol{z}_B) = \boldsymbol{f}^{(B)}(\boldsymbol{z}'_B)$. By assumption, we know there exists a $\boldsymbol{z}_{B^c}$ s.t. $(\boldsymbol{z}_B, \boldsymbol{z}_{B^c})$ and $(\boldsymbol{z}'_B, \boldsymbol{z}_{B^c})$ are in $\mathcal{Z}^{\text{train}}$. By construction, we have that $\bar{\boldsymbol{f}}((\boldsymbol{z}_B, \boldsymbol{z}_{B^c})) = \bar{\boldsymbol{f}}((\boldsymbol{z}'_B, \boldsymbol{z}_{B^c}))$. By injectivity of $\bar{\boldsymbol{f}}$, we have that $(\boldsymbol{z}_B, \boldsymbol{z}_{B^c}) \neq (\boldsymbol{z}'_B, \boldsymbol{z}_{B^c})$, which implies $\boldsymbol{z}_B \neq \boldsymbol{z}'_B$, i.e. $\boldsymbol{f}^{(B)}$ is injective. $\square$

### A.10 Proof of Corollary 3

**Corollary 3** (Cartesian-product extrapolation). *Suppose the assumptions of Theorem 2 holds. Then,*

$$\text{for all } \boldsymbol{z} \in \text{CPE}_{\mathcal{B}}(\hat{\mathcal{Z}}^{\text{train}}), \ \sum_{B\in\mathcal{B}} \hat{\boldsymbol{f}}^{(B)}(\boldsymbol{z}_B) = \sum_{B\in\mathcal{B}} \boldsymbol{f}^{(\pi(B))}(\bar{\boldsymbol{v}}_{\pi(B)}(\boldsymbol{z}_B)). \tag{11}$$

*Furthermore, if $\text{CPE}_{\mathcal{B}}(\mathcal{Z}^{\text{train}}) \subseteq \mathcal{Z}^{\text{test}}$, then $\hat{\boldsymbol{f}}(\text{CPE}_{\mathcal{B}}(\hat{\mathcal{Z}}^{\text{train}})) \subseteq \boldsymbol{f}(\mathcal{Z}^{\text{test}})$.*

*Proof.* Pick $\boldsymbol{z} \in \text{CPE}(\hat{\mathcal{Z}}^{\text{train}})$. By definition, this means that, for all $B \in \mathcal{B}$, $\boldsymbol{z}_B \in \hat{\mathcal{Z}}_B^{\text{train}}$. We thus have that, for all $B \in \mathcal{B}$,

$$\hat{\boldsymbol{f}}^{(B)}(\boldsymbol{z}_B) = \boldsymbol{f}^{(\pi(B))} \circ \bar{\boldsymbol{v}}_{\pi(B)}(\boldsymbol{z}_B) + \boldsymbol{c}^{(B)}. \tag{109}$$

We can thus sum over $B$ to obtain

$$\sum_{B\in\mathcal{B}} \hat{\boldsymbol{f}}^{(B)}(\boldsymbol{z}_B) = \sum_{B\in\mathcal{B}} \boldsymbol{f}^{(\pi(B))} \circ \bar{\boldsymbol{v}}_{\pi(B)}(\boldsymbol{z}_B) + \underbrace{\sum_{B\in\mathcal{B}} \boldsymbol{c}^{(B)}}_{=0}. \tag{110}$$

Since $\boldsymbol{z} \in \text{CPE}(\hat{\mathcal{Z}}^{\text{train}})$ was arbitrary, we have

$$\text{for all } \boldsymbol{z} \in \text{CPE}(\hat{\mathcal{Z}}^{\text{train}}), \ \sum_{B\in\mathcal{B}} \hat{\boldsymbol{f}}^{(B)}(\boldsymbol{z}_B) = \sum_{B\in\mathcal{B}} \boldsymbol{f}^{(\pi(B))} \circ \bar{\boldsymbol{v}}_{\pi(B)}(\boldsymbol{z}_B) \tag{111}$$

$$\hat{\boldsymbol{f}}(\boldsymbol{z}) = \boldsymbol{f} \circ \bar{\boldsymbol{v}}(\boldsymbol{z}), \tag{112}$$

where $\bar{\boldsymbol{v}} : \text{CPE}_{\mathcal{B}}(\hat{\mathcal{Z}}^{\text{train}}) \to \text{CPE}_{\mathcal{B}}(\mathcal{Z}^{\text{train}})$ is defined as

$$\bar{\boldsymbol{v}}(\boldsymbol{z}) := \begin{bmatrix} \bar{\boldsymbol{v}}_{B_1}(\boldsymbol{z}_{\pi^{-1}(B_1)}) \\ \vdots \\ \bar{\boldsymbol{v}}_{B_\ell}(\boldsymbol{z}_{\pi^{-1}(B_\ell)}) \end{bmatrix}, \tag{113}$$

The map $\bar{\boldsymbol{v}}$ is a diffeomorphism since each $\bar{\boldsymbol{v}}_{\pi(B)}$ is a diffeomorphism from $\hat{\mathcal{Z}}_B^{\text{train}}$ to $\mathcal{Z}_{\pi(B)}^{\text{train}}$.

By (112) we get

$$\hat{\boldsymbol{f}}(\text{CPE}_{\mathcal{B}}(\hat{\mathcal{Z}}^{\text{train}})) = \boldsymbol{f} \circ \bar{\boldsymbol{v}}(\text{CPE}_{\mathcal{B}}(\hat{\mathcal{Z}}^{\text{train}})), \tag{114}$$

and since the map $\bar{\boldsymbol{v}}$ is surjective we have $\bar{\boldsymbol{v}}(\text{CPE}_{\mathcal{B}}(\hat{\mathcal{Z}}^{\text{train}})) = \text{CPE}_{\mathcal{B}}(\mathcal{Z}^{\text{train}})$ and thus

$$\hat{\boldsymbol{f}}(\text{CPE}_{\mathcal{B}}(\hat{\mathcal{Z}}^{\text{train}})) = \boldsymbol{f}(\text{CPE}_{\mathcal{B}}(\mathcal{Z}^{\text{train}})). \tag{115}$$

Hence if $\text{CPE}_{\mathcal{B}}(\mathcal{Z}^{\text{train}}) \subseteq \mathcal{Z}^{\text{test}}$, then $\boldsymbol{f}(\text{CPE}_{\mathcal{B}}(\mathcal{Z}^{\text{train}})) \subseteq \boldsymbol{f}(\mathcal{Z}^{\text{test}})$. $\square$

## A.11 Will all extrapolated images make sense?

Here is a minimal example where the assumption $\text{CPE}_{\mathcal{B}}(\mathcal{Z}^{\text{train}}) \not\subseteq \mathcal{Z}^{\text{test}}$ is violated.

**Example 10** (Violation of $\text{CPE}_{\mathcal{B}}(\mathcal{Z}^{\text{train}}) \not\subseteq \mathcal{Z}^{\text{test}}$). *Imagine $z = (z_1, z_2)$ where $z_1$ and $z_2$ are the $x$-positions of two distinct balls. It does not make sense to have two balls occupying the same location in space and thus whenever $z_1 = z_2$ we have $(z_1, z_2) \notin \mathcal{Z}^{\text{test}}$. But if $(1, 2)$ and $(2, 1)$ are both in $\mathcal{Z}^{\text{train}}$, it implies that $(1, 1)$ and $(2, 2)$ are in $\text{CPE}(\mathcal{Z}^{\text{train}})$, which is a violation of $\text{CPE}_{\mathcal{B}}(\mathcal{Z}^{\text{train}}) \subseteq \mathcal{Z}^{\text{test}}$.*

## A.12 Additive decoders cannot model occlusion

We now explain why additive decoders cannot model occlusion. Occlusion occurs when an object is partially hidden behind another one. Intuitively, the issue is the following: Consider two images consisting of two objects, A and B (each image shows both objects). In both images, the position of object A is the same and in exactly one of the images, object B partially occludes object A. Since the position of object $A$ did not change, its corresponding latent block $z_A$ is also unchanged between both images. However, the pixels occupied by object A do change between both images because of occlusion. The issue is that, because of additivity, $z_A$ and $z_B$ cannot interact to make some pixels that belonged to object A "disappear" to be replaced by pixels of object B. In practice, object-centric representation learning methods rely a masking mechanism which allows interactions between $z_A$ and $z_B$ (See Equation 1 in Section 2). This highlights the importance of studying this class of decoders in future work.

# B Experiments

## B.1 Training Details

**Loss Function.** We use the standard reconstruction objective of mean squared error loss between the ground truth data and the reconstructed/generated data.

**Hyperparameters.** For both the ScalarLatents and the BlockLatents dataset, we used the Adam optimizer with the hyperparameters defined below. Note that we maintain consistent hyperparameters across both the Additive decoder and the Non-Additive decoder method.

*ScalarLatents Dataset.*

- Batch Size: 64
- Learning Rate: $1 \times 10^{-3}$
- Weight Decay: $5 \times 10^{-4}$
- Total Epochs: 4000

*BlockLatents Dataset.*

- Batch Size: 1024
- Learning Rate: $1 \times 10^{-3}$
- Weight Decay: $5 \times 10^{-4}$
- Total Epochs: 6000

**Model Architecture.** We use the following architectures for Encoder and Decoder across both the datasets (ScalarLatents, BlockLatents). Note that for the ScalarLatents dataset we train with latent dimension $d_z = 2$, and for the BlockLatents dataset we train with latent dimension $d_z = 4$, which corresponds to the dimensionalities of the ground-truth data generating process for both datasets.

*Encoder Architecture:*

- RestNet-18 Architecture till the penultimate layer (512 dimensional feature output)
- Stack of 5 fully-connected layer blocks, with each block consisting of Linear Layer ( dimensions: $512 \times 512$), Batch Normalization layer, and Leaky ReLU activation (negative slope: 0.01).

- Final Linear Layer (dimension: $512 \times d_z$) followed by Batch Normalization Layer to output the latent representation.

*Decoder Architecture (Non-additive):*

- Fully connected layer block with input as latent representation, consisting of Linear Layer (dimension: $d_z \times 512$), Batch Normalization layer, and Leaky ReLU activation (negative slope: 0.01).
- Stack of 5 fully-connected layer blocks, with each block consisting of Linear Layer ( dimensions: $512 \times 512$), Batch Normalization layer, and Leaky ReLU activation (negative slope: 0.01).
- Series of DeConvolutional layers, where each DeConvolutional layer is follwed by Leaky ReLU (negative slope: 0.01) activation.
    - DeConvolution Layer ($c_{in}$: 64, $c_{out}$: 64, kernel: 4; stride: 2; padding: 1)
    - DeConvolution Layer ($c_{in}$: 64, $c_{out}$: 32, kernel: 4; stride: 2; padding: 1)
    - DeConvolution Layer ($c_{in}$: 32, $c_{out}$: 32, kernel: 4; stride: 2; padding: 1)
    - DeConvolution Layer ($c_{in}$: 32, $c_{out}$: 3, kernel: 4; stride: 2; padding: 1)

*Decoder Architecture (Additive):* Recall that an additive decoder has the form $\boldsymbol{f}(\boldsymbol{z}) = \sum_{B \in \mathcal{B}} \boldsymbol{f}^{(B)}(\boldsymbol{z}_B)$. Each $\boldsymbol{f}^{(B)}$ has the same architecture as the one presented above for the non-additive case, but the input has dimensionality $|B|$ (which is 1 or 2, depending on the dataset). Note that we do not share parameters among the functions $\boldsymbol{f}^{(B)}$.

## B.2 Datasets Details

We use the moving balls environment from Ahuja et al. [2] with images of dimension $64 \times 64 \times 3$, with latent vector ($\boldsymbol{z}$) representing the position coordinates of each balls. We consider only two balls. The rendered images have pixels in the range [0, 255].

**ScalarLatents Dataset.** We fix the x-coordinate of each ball to $0.25$ and $0.75$. The only factors varying are the y-coordinates of both balls. Thus, $\boldsymbol{z} \in \mathbb{R}^2$ and $\mathcal{B} = \{\{1\}, \{2\}\}$ where $\boldsymbol{z}_1$ and $\boldsymbol{z}_2$ designate the y-coordinates of both balls. We sample the y-coordinate of the first ball from a continuous uniform distribution as follows: $\boldsymbol{z}_1 \sim \text{Uniform}(0, 1)$. Then we sample the y-coordinate of the second ball as per the following scheme:

$$\boldsymbol{z}_2 \sim \begin{cases} \text{Uniform}(0, 1) & \text{if } \boldsymbol{z}_1 \leq 0.5 \\ \text{Uniform}(0, 0.5) & \text{else} \end{cases}$$

Hence, this leads to the L-shaped latent support, i.e., $\mathcal{Z}^{\text{train}} := [0, 1] \times [0, 1] \setminus [0.5, 1] \times [0.5, 1]$.

We use $50k$ samples for the test dataset, while we use $20k$ samples for the train dataset along with $5k$ samples (25% of the train sample size) for the validation dataset.

**BlockLatents Dataset.** For this dataset, we allow the balls to move in both the x, y directions, so that $\boldsymbol{z} \in \mathbb{R}^4$ and $\mathcal{B} = \{\{1, 2\}, \{3, 4\}\}$. For the case of **independent latents**, we sample each latent component independently and identically distributed according to a uniform distribution over $(0, 1)$, i.e. $z_i \sim \text{Uniform}(0, 1)$. We rejected the images that present occlusion, i.e. when one ball hides another one.[2]

For the case of **dependent latents**, we sample the latents corresponding to the first ball similarly from the same continuous uniform distribution, i.e, $z_1, z_2 \sim \text{Uniform}(0, 1)$. However, the latents of the second ball are a function of the latents of the first ball, as described in what follows:

$$z_3 \sim \begin{cases} \text{Uniform}(0, 0.5) & \text{if } 1.25 \times (\boldsymbol{z}_1^2 + \boldsymbol{z}_2^2) \geq 1.0 \\ \text{Uniform}(0.5, 1) & \text{if } 1.25 \times (\boldsymbol{z}_1^2 + \boldsymbol{z}_2^2) < 1.0 \end{cases}$$

---

[2]Note that, in the independent latents case, the latents are not actually independent because of the rejection step which prevents occlusion from happening.

$$z_4 \sim \begin{cases} \text{Uniform}(0.5, 1) & \text{if } 1.25 \times (z_1^2 + z_2^2) \geq 1.0 \\ \text{Uniform}(0, 0.5) & \text{if } 1.25 \times (z_1^2 + z_2^2) < 1.0 \end{cases}$$

Intuitively, this means the second ball will be placed in either the top-left or the bottom-right quadrant based on the position of the first ball. We also exclude from the dataset the images presenting occlusion.

Note that our dependent BlockLatent setup is same as the non-linear SCM case from Ahuja et al. [3].

We use $50k$ samples for both the train and the test dataset, along with $12.5k$ samples ($25\%$ of the train sample size) for the validation dataset.

**Disconnected Support Dataset.** For this dataset, we have setup similar to the **ScalarLatents** dataset; we fix the x-coordinates of both balls to $0.25$ and $0.75$ and only vary the y-coordinates so that $z \in \mathbb{R}^2$. We sample the y-coordinate of the first ball ($z_1$) from Uniform(0, 1). Then we sample the y-coordinate of the second ball ($z_2$) from either of the following continuous uniform distribution with equal probability; Uniform(0, 0.25) and Uniform(0.75, 1). This leads to a disconnected support given by $\mathcal{Z}^{\text{train}} := [0, 1] \times [0, 1] \setminus [0.25, 0.75] \times [0.25, 0.75]$.

We use $50k$ samples for the test dataset, while we use $20k$ samples for the train dataset along with $5k$ samples ($25\%$ of the train sample size) for the validation dataset.

## B.3 Evaluation Metrics

Recall that, to evaluate disentanglement, we compute a matrix of scores $(s_{B,B'}) \in \mathbb{R}^{\ell \times \ell}$ where $\ell$ is the number of blocks in $\mathcal{B}$ and $s_{B,B'}$ is a score measuring how well we can predict the ground-truth block $z_B$ from the learned latent block $\hat{z}_{B'} = \hat{g}_{B'}(x)$ outputted by the encoder. The final Latent Matching Score (LMS) is computed as $\text{LMS} = \arg\max_{\pi \in \mathfrak{S}_{\mathcal{B}}} \frac{1}{\ell} \sum_{B \in \mathcal{B}} s_{B,\pi(B)}$, where $\mathfrak{S}_{\mathcal{B}}$ is the set of permutations respecting $\mathcal{B}$ (Definition 2). These scores are always computed on the test set.

**Metric $\text{LMS}_{\text{Spear}}$:** As mentioned in the main paper, this metric is used for the **ScalarLatents** dataset where each block is 1-dimensional. Hence, this metric is almost the same as the mean correlation coefficient (MCC), which is widely used in the nonlinear ICA literature [30, 31, 33, 36, 42], with the only difference that we use Spearman correlation instead of Pearson correlation as a score $s_{B,B'}$. The Spearman correlation can capture nonlinear monotonous relations, unlike Pearson which can only capture linear dependencies. We favor Spearman over Pearson because our identifiability result (Theorem 2) guarantees we can recover the latents only up to permutation and element-wise invertible transformations, which can be nonlinear.

**Metric $\text{LMS}_{\text{tree}}$:** This metric is used for the **BlockLatents** dataset. For this metric, we take $s_{B,B'}$ to be the $R^2$ score of a Regression Tree with maximal depth of 10. For this, we used the class `sklearn.tree.DecisionTreeRegressor` from the `sklearn` library. We learn the parameters of the Decision Tree using the train dataset and then use it to evaluate $\text{LMS}_{\text{tree}}$ metric on the test dataset. For the additive decoder, it is easy to compute this metric since the additive structure already gives a natural partition $\mathcal{B}$ which matches the ground-truth. However, for the non-additive decoder, there is no natural partition and thus we cannot compute $\text{LMS}_{\text{tree}}$ directly. To go around this problem, for the non-additive decoder, we compute $\text{LMS}_{\text{tree}}$ for all possible partitions of $d_z$ latent variables into blocks of size $|B| = 2$ (assuming all blocks have the same dimension), and report the best $\text{LMS}_{\text{tree}}$. This procedure is tractable in our experiments due to the small dimensionality of the problem we consider.

## B.4 Boxplots for main experiments (Table 1)

Since the standard error in the main results (Table 1) was high, we provide boxplots in Figures 9 & 10 to have a better visibility on what is causing this. We observe that the high standard error for the Additive approach was due to bad performance for a few bad random initializations for the ScalarLatents dataset; while we have nearly perfect latent identification for the others. Figure 14e shows the latent space learned by the worst case seed, which somehow learned a disconnected support even if the ground-truth support was connected. Similarly, for the case of Independent BlockLatents, there are only a couple of bad random initializations and the rest of the cases have perfect identification.

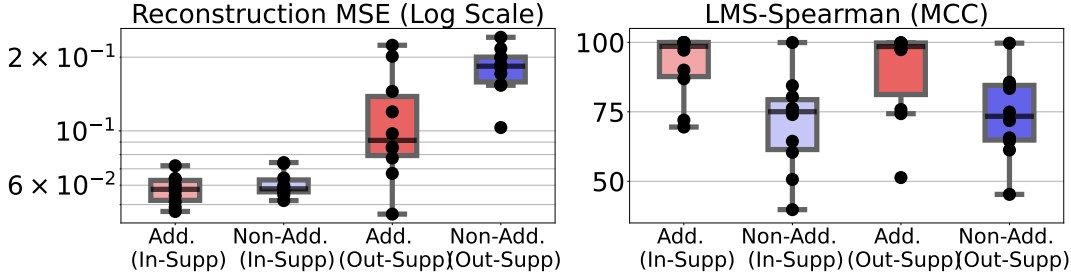

Figure 9: Reconstruction mean squared error (MSE) ($\downarrow$) and Latent Matching Score (LMS) ($\uparrow$) over 10 different random initializations for **ScalarLatents** dataset.

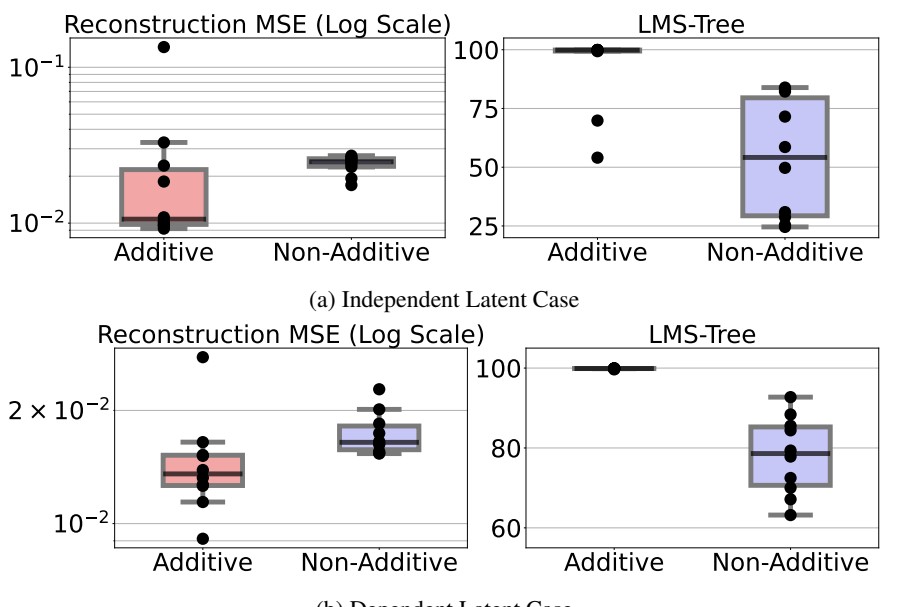

(a) Independent Latent Case

(b) Dependent Latent Case

Figure 10: Reconstruction mean squared error (MSE) ($\downarrow$) and Latent Matching Score (LMS) ($\uparrow$) for 10 different initializations for **BlockLatents** dataset.

## B.5  Additional Results: BlockLatents Dataset

To get a qualitative understanding of latent identification in the BlockLatents dataset, we plot the response of each predicted latent as we change a particular ground-truth latent factor. We describe the following cases of changing the ground-truth latents:

- **Ball 1 moving along x-axis:** We sample 10 equally spaced points for $z_1$ from $[0, 1]$; while keeping other latents fixed as follows: $z_2 = 0.25, z_3 = 0.50, z_4 = 0.75$. We will never have occlusion since the balls are separated along the y-axis $z_4 - z_2 > 0$.

- **Ball 2 moving along x-axis:** We sample 10 equally spaced points for $z_3$ from $[0, 1]$; while keeping other latents fixed as follows: $z_1 = 0.50, z_2 = 0.25, z_4 = 0.75$. We will never have occlusion since the balls are separated along the y-axis $z_4 - z_2 > 0$.

- **Ball 1 moving along y-axis:** We sample 10 equally spaced points for $z_2$ from $[0, 1]$; while keeping other latents fixed as follows: $z_1 = 0.25, z_3 = 0.75, z_4 = 0.50$. We will never have occlusion since the balls are separated along the x-axis $z_3 - z_1 > 0$.

- **Ball 2 moving along y-axis:** We sample 10 equally spaced points for $z_4$ from $[0, 1]$; while keeping other latents fixed as follows: $z_1 = 0.25, z_2 = 0.50, z_3 = 0.75$. We will never have occlusion since the balls are separated along the x-axis $z_3 - z_1 > 0$.

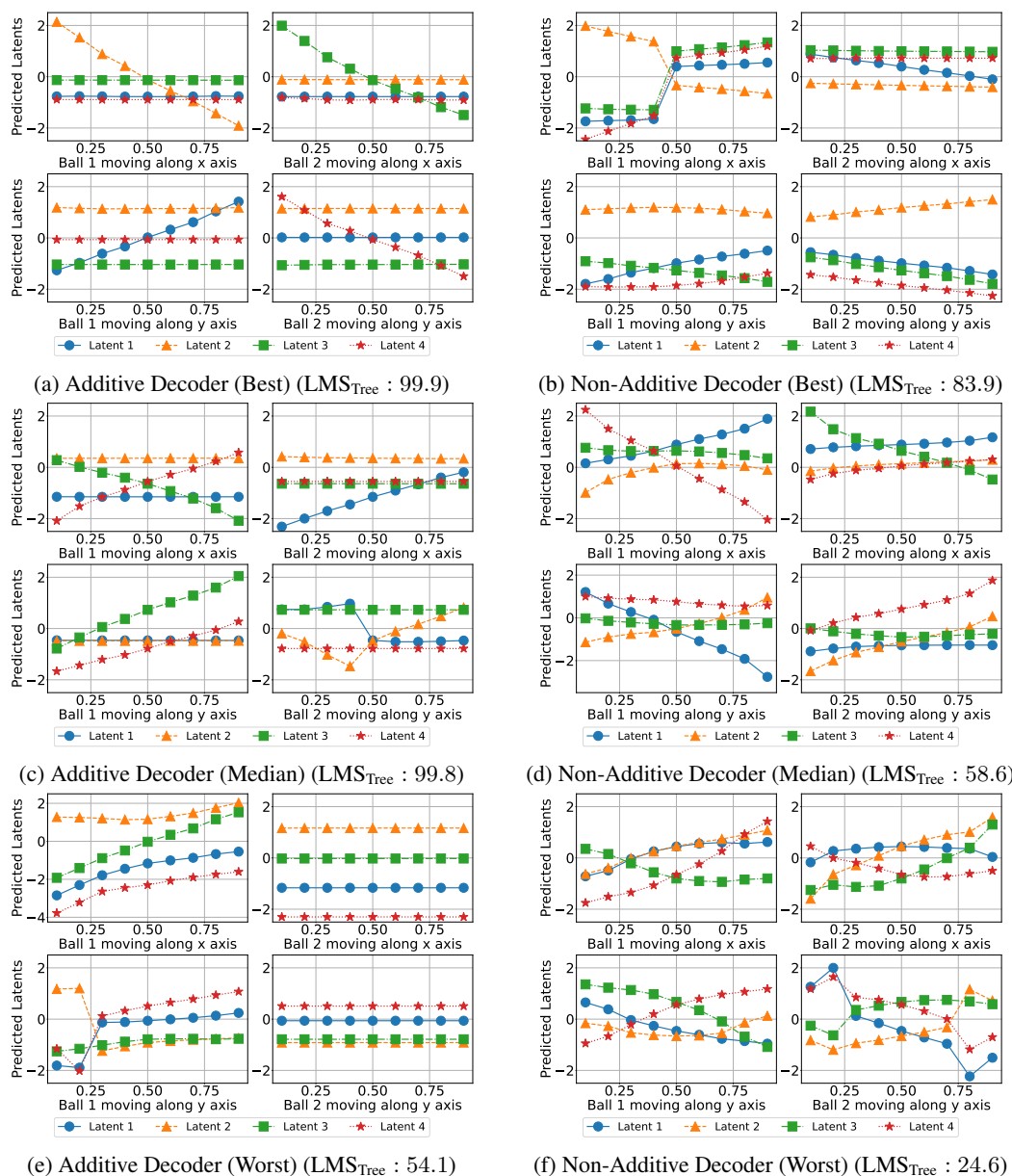

(a) Additive Decoder (Best) (LMS$_{\text{Tree}}$ : 99.9)

(b) Non-Additive Decoder (Best) (LMS$_{\text{Tree}}$ : 83.9)

(c) Additive Decoder (Median) (LMS$_{\text{Tree}}$ : 99.8)

(d) Non-Additive Decoder (Median) (LMS$_{\text{Tree}}$ : 58.6)

(e) Additive Decoder (Worst) (LMS$_{\text{Tree}}$ : 54.1)

(f) Non-Additive Decoder (Worst) (LMS$_{\text{Tree}}$ : 24.6)

Figure 11: Latent responses for the cases with the **best/median/worst** LMS$_{\text{Tree}}$ among runs performed on the **BlockLatent** dataset with independent latents. In each plot, we report the latent factors predicted from multiple images where one ball moves along only one axis at a time.

Figure 5 in the main paper presents the latent responses plot for the median LMS$_{\text{tree}}$ case among random initializations. In Figure 11, we provide the results for the case of best and the worst LMS$_{\text{tree}}$ among random seeds. We find that Additive Decoder fails for only for the worst case random seed, while Non-Additive Decoder fails for all the cases.

Additionally, we provide the object-specific reconstructions for the Additive Decoder in Figure 12. This helps us better understand the failure of Additive Decoder for the worst case random seed (Figure 12c), where the issue arises due to bad reconstruction error.

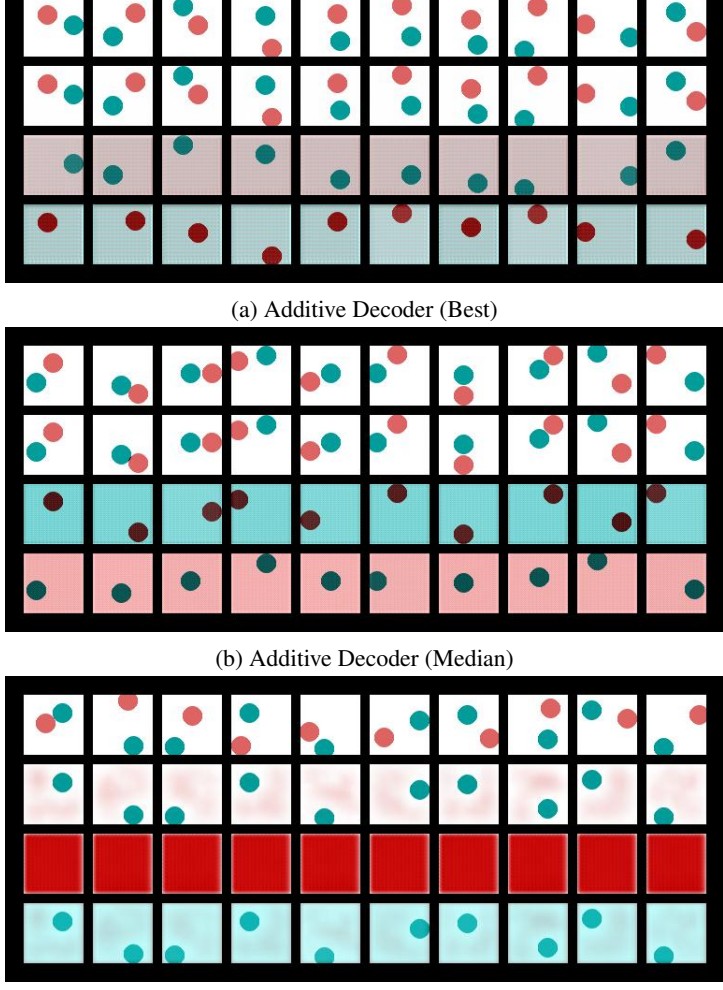

(a) Additive Decoder (Best)

(b) Additive Decoder (Median)

(c) Additive Decoder (Worst)

Figure 12: Object-specific renderings with the **best/median/worst** $LMS_{tree}$ among runs performed on the **BlockLatents** dataset with independent latents. In each plot, the first row is the original image, the second row is the reconstruction and the third and fourth rows are the output of the object-specific decoders. In the best and median cases, each object-specific decoder corresponds to one and only one object, e.g. the third row of the best case always corresponds to the red ball. However, in the worst case, there are issues with reconstruction as only one of the balls is generated. Note that the visual artefacts are due to the additive constant indeterminacy we saw in Theorem 2, which cancel each other as is suggested by the absence of artefacts in the reconstruction.

### B.6 Disconnected Support Experiments

Since path-connected latent support is an important assumption for latent identification with additive decoders (Theorem 2), we provide results for the case where the assumption is not satisfied. We experiment with the **Disconnected Support** dataset (Section B.2) and find that we obtain much worse $LMS_{Spear}$ as compared to the case of training with L-shaped support in the **ScalarLatents** dataset. Over 10 different random initializations, we find mean $LMS_{Spear}$ performance of 69.5 with standard error of 6.69.

For better qualitative understanding, we provide visualization of the latent support and the extrapolated images for the median $LMS_{Spear}$ among 10 random seeds in Figure 13. Somewhat surprisingly, the representation appears to be aligned in the sense that the first predicted latent corresponds to the blue ball while the second predicted latent correspond to the red ball. Also surprisingly, extrapolation

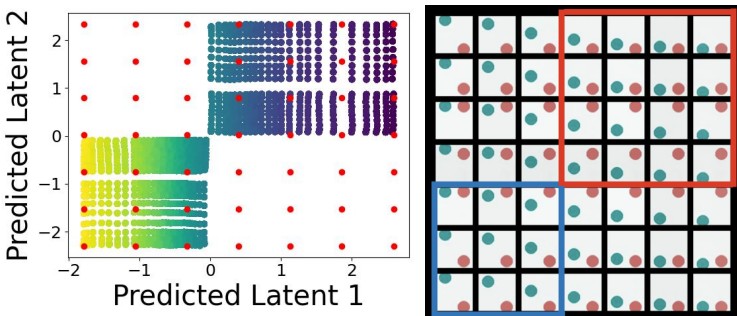

Figure 13: Learned latent space, $\hat{\mathcal{Z}}^{\text{train}}$, and the corresponding reconstructed images of the additive decoder with the **median** $\text{LMS}_{\text{Spear}}$ among runs performed on the **Disconnected Support** dataset. The red dots correspond to latent factors used to generate the images.

occurs (we can see images of both balls high). That being said, we observe that the relationship between the predicted latent 2 ($\hat{z}_2$) and y-coordinate of second (red) ball is not monotonic, which explains why the Spearman correlation is so low (Spearman correlation scores are high when there is a monotonic relationship between both variables).

## B.7 Additional Results: ScalarLatents Dataset

To get a qualitative understanding of extrapolation, we plot the latent support on the test dataset and sample a grid of equally spaced points from the support of each predicted latent on the test dataset. The grid represents the cartesian-product of the support of predicted latents and would contain novel combinations of latents that were unseen during training. We show the reconstructed images for each point from the cartesian-product grid to see whether the model is able to reconstruct well the novel latent combinations.

Figure 4 in the main paper presents visualizations of the latent support and the extrapolated images for the median $\text{LMS}_{\text{Spear}}$ case among random seeds. In Figure 14, we provide the results for the case of best and the worst $\text{LMS}_{\text{Spear}}$ among random seeds. We find that even for the best case (Figure 14b), Non-Additive Decoder does not generate good quality extrapolated images, while Additive Decoder generates extrapoalted images for the best and median case. The worst-case run for the Additive Decoder has disconnected support, which explains why it is not able to extrapolate.

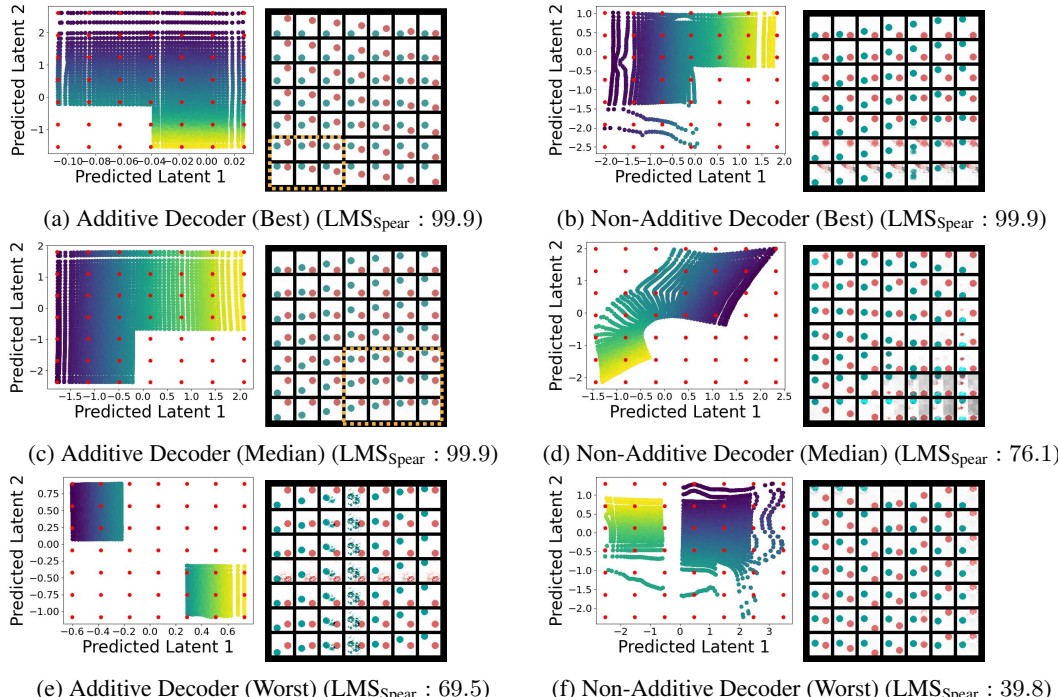

(a) Additive Decoder (Best) (LMS$_{\text{Spear}}$ : 99.9)  (b) Non-Additive Decoder (Best) (LMS$_{\text{Spear}}$ : 99.9)

(c) Additive Decoder (Median) (LMS$_{\text{Spear}}$ : 99.9)  (d) Non-Additive Decoder (Median) (LMS$_{\text{Spear}}$ : 76.1)

(e) Additive Decoder (Worst) (LMS$_{\text{Spear}}$ : 69.5)  (f) Non-Additive Decoder (Worst) (LMS$_{\text{Spear}}$ : 39.8)

Figure 14: Figure (a, c, e) shows the learned latent space, $\hat{\mathcal{Z}}^{\text{train}}$, and the corresponding reconstructed images of the additive decoder with the **best/median/worst** LMS$_{\text{Spear}}$ among runs performed on the **ScalarLatents** dataset. Figure (b, d, f) shows the same thing for the non-additive decoder. The red dots correspond to latent factors used to generate the images and the yellow square highlights extrapolated images.

