# OpenReview forum: "Additive Decoders for Latent Variables Identification and Cartesian-Product Extrapolation"
_NeurIPS.cc/2023/Conference — NeurIPS 2023 oral_

### Official Review · Reviewer_2rwH · 2023-06-27

**Soundness:** 3 good
**Presentation:** 3 good
**Contribution:** 3 good
**Rating:** 7
**Confidence:** 4

**Summary:**

This paper develops a theory to show how an additive decoder may be able to disentangle an image composed of several components. The proposed theory also shows how an additive decoder may produce novel images, possibly providing insights about the process performed by generative models.

Besides the theory, paper also performs some experiments showing that additive decoders may have certain advantages in performing those tasks.

**Strengths:**

Paper is well written.

The topic and the experiments are interesting.

It has a broad and practical view. Literature review is relatively good.

The proposed theory might turn into a useful contribution for the research community.

**Weaknesses:**

Experiments are interesting but limited and disconnected from existing experiments in the literature, in my view.

-----------------

There are a few publications that are relevant but not cited:

–Li, N., Raza, M.A., Hu, W., Sun, Z. and Fisher, R, Object-centric representation learning with generative spatial-temporal factorization, NeurIPS 2021.

–Yoon, J., Wu, Y.F., Bae, H. and Ahn, S., An investigation into pre-training object-centric representations for reinforcement learning, ICML 2023.

Both of the above papers have experiments on images that may be decomposed in the authors’ additive scenario and possibly be used as a baseline for comparison.

-----------------

“Reasonableness”, mentioned under section 3.2, seems to be an unclear definition underlying a significant portion of the theory. It appears that authors attempt to define the reasonableness, yet, it is not clear what is the difference between $Z^{test}$ and $Z^{train}$. Given the bijective assumption, the difference between $Z^{test}$ and $Z^{train}$ should refer to a specific region in the domain and range of the function. Yet, it is not clear what that difference is.

I understand it is hard to define the limits of the underlying manifold of relevant images - that is exactly the heart of the difficulty in developing useful theory for deep learning. Could authors expand on their “reasonable” assumption? Perhaps identifying the boundaries of the “reasonable” manifold is hard, but it may be helpful to describe and contrast what is unreasonable. Perhaps providing a discussion and citing some previous studies on the underlying manifold of images would be helpful as well, e.g.:

Cohen, U., Chung, S., Lee, D.D. and Sompolinsky, H., 2020. Separability and geometry of object manifolds in deep neural networks. Nature Communications, 11(1), p.746.


-----------------

There is an extrapolation study and a dataset called VAEC from the paper below.

Webb, T., Dulberg, Z., Frankland, S., Petrov, A., O’Reilly, R. and Cohen, J., Learning representations that support extrapolation. ICML 2020.

The images in the VAEC dataset are designed as an extrapolation task. Do authors think this task can fit into their framework?


-----------------

It seems that the notation D (for the Jacobian) is only defined in the appendix under Table 2. Since the notation is used in the main body of the paper, it would be useful to define it there. If instead of D, $\nabla$ was used for the Jacobian, I would have inferred what authors mean by it. However, I was not sure about D until I found it in the appendix.

-----------------

For assumption 2, it may be better to use “linearly independent” instead of “independent”.

Moreover, it might be better to explain in words that: this assumption is requiring the … matrices, to have full column rank.

Overall, the notion formalized in assumption 2 seems strange to me. Are authors familiar with the notion of curvature for functions and manifolds?

Is there any precedence for the notion of nonlinearity defined under assumption 2 for any class of functions? I am not sure how authors’ notion of nonlinearity relates to known notions of nonlinearity/curvature, for example, the notions of curvature in differential geometry. Why should assumption 2 be satisfied over the entire manifold?

In its current form, assumption 2 stacks the first and second order derivatives together and then requires that the stacked matrix to have full column rank. It is not clear to me why stacking of these matrices is necessary. If the unrolled second derivatives have full column rank, would it be necessary for the first derivatives to have full column rank? If each of the first and second order derivatives, individually have full column rank, would that be sufficient?


-----------------

This is not a weakness of the paper, but perhaps worth mentioning. It is common in the literature to use x as the input to a function/model, and use y or z as the output of a function/model. However, this paper uses z to denote the inputs and x to denote the output. This may sometimes be a bit confusing. But that is the authors’ choice, and this is just feedback.



**Questions:**

Please see the questions under weaknesses, especially the questions about assumption 2. (Assumption 2 and what is built on it is my main concern about this work. I hope authors can be more clear and more convincing about their approach.)

Have authors considered expanding their own experiments to something more sophisticated? For example, the shape of objects could be not just circles, but several other types: diamonds, triangles, rectangles, etc. The size of the objects could vary as well.

Do authors think their extrapolation method can be applied to the VAEC dataset (or some modification of it)? The point is to connect this paper’s experiments to existing experiments in the literature. For example, when an object in the VAEC dataset is enlarged, the enlarged object can be considered an addition of two smaller objects which would fit the authors’ additive framework. Currently, this paper’s experiments seem to be isolated from the literature. If VAEC is not suitable, authors may want to consider some other datasets from the literature.


**Limitations:**

I did not see a discussion on limitations.

---

> ### Author Rebuttal · Authors · 2023-08-09
>
> **Regarding uncited relevant works:**
>
> Object-centric representation learning is a vast field and our literature review did not cover all works. We believe many OCRL datasets have this nice additive structure and many OCRL approaches have the inductive bias of additive decoder, which underlines the relevance of our work. We will add these works to our review.
>
> We want to reemphasize the point we addressed in our general response: we are not proposing any novel approach for object-centric learning; instead we analyze additivity as an inductive bias already present in many OCRL decoders. Hence, our analysis is applicable to several OCRL methods. The only baseline we considered was non-additive decoder as we are concerned with understanding the impact of additivity on latent identification and extrapolation. Hence, using the suggested papers by Nanbo et al. and Yoon et al. as baselines does not align with the goal of our analysis.
>
> **“The images in the VAEC dataset are designed as an extrapolation task. Do authors think this task can fit into their framework?”**
>
> The VAEC dataset contains a single object, and we believe the attributes like size, location, and brightness cannot be disentangled/identified using additive decoders as the effect of these latent variables on the resulting image is not additive. For example, in our image dataset we could disentangle the latents (position) of a ball from those of the other ball, however, we cannot disentangle the x-y positions of each ball. Hence, the additive decoder inductive bias cannot help us to recover the position of the object in the VAEC dataset, similar to the non-identifiability of position of a particular ball in our dataset. This also implies the question of extrapolation in the VAEC dataset is out of scope from our analysis as we provide guarantees for extrapolation w.r.t latent factors that have additive relationship in the true data generation process.
>
> We now attempt to demonstrate that the “scale” factor of variation is not additive, as suggested by the reviewer: Let z_1 = size of object and z_2 = position of object and let’s forget about the other factors for the sake of the argument. We now want to write the images as x = f1(z_1) + f2(z_2). If we want f1(z_1) to correspond to “adding mass to object”, the issue is that “where to add the mass in the image” will depend on “where the object is located”, which is given by z_2. This means f1(z_1) could not implement “adding mass” since, to do so, it would have to depend on z_2 (which would violate additivity).
>
> **“It appears that authors attempt to define the reasonableness”**
>
> Clarification: We do not attempt to define reasonableness. Instead, we denote by Z^test the set of latent factors that are *reasonable* for a given application, without explicitly defining what it means to be reasonable. This is similar in spirit to the ground-truth decoder f which captures what are the "natural factors of variations" in the given application. Most works (including ours) do not try to define what makes some factors of variations "natural", they just denote by f the mapping between the "natural factors" and the images x, whatever these natural factors might be in a given application.
>
> **“It is not clear what is the difference between Z^test and Z^train”**
>
> Z^test is the set of all reasonable latent vectors. Z^train is the set of latent vectors observed during training. This distinction is made in Assumption 1.
>
> **“Perhaps identifying the boundaries of the “reasonable” manifold is hard, but it may be helpful to describe and contrast what is unreasonable.”**
>
> In Appendix A.10, we give an example of an unreasonable z in the context of occlusion, i.e. z \not\in Z^test.
>
> **“It seems that the notation D (for the Jacobian) is only defined in the appendix under Table 2”**
>
> The notation is introduced for the first time on page 2 in the paragraph titled “Notation”. We did not use \nabla to avoid the confusion with the gradient of a scalar-valued function, since here we are concerned with the derivative of vector-valued functions. A solution to improve clarity would be to recall the meaning of D the first time it is used in the paper. We’ll make sure to add this clarification in the next revision.
>
> **“For assumption 2, it may be better to use “linearly independent” instead of “independent””**
>
> Agreed, we implemented the change in the text.
>
> **“Is there any precedence for the notion of nonlinearity defined under assumption 2 for any class of functions?”**
>
> Not that we know of. However this assumption is very closely related to previous works in nonlinear ICA (see discussion in Appendix A.5)
>
> **Assumption 2 and notions of curvature in differential geometry.**
>
> Connection between Assumption 2 and known notions of curvature in differential geometry is a very interesting question worth investigating in the future as it might lead to a better understanding of these types of assumptions in nonlinear ICA. However we consider this to be out of scope for this work.
>
> **“Why should assumption 2 be satisfied over the entire manifold?”**
>
> It comes from the nature of the proof. The argument is local in the sense that we start by showing "local disentanglement" in Theorem 1 (Jacobian of v is a block-permutation scaling everywhere) and then use further assumptions to show global disentanglement in Theorem 2. For the local part of the proof, we need assumption 2 to hold for all z.
>
> **“If each of the first and second order derivatives, individually have full column rank, would that be sufficient?”**
>
> Again this comes from the nature of our proof, which requires W(z) to have full column-rank everywhere. Having the first and second derivatives individually have full rank would not be sufficient in our current proof (although we do not exclude the possibility that a proof exists that requires only this weaker condition).

---

> > ### Comment · Reviewer_2rwH · 2023-08-14
> >
> > I believe the authors' response and the discussions included in the appendix address most of my questions/concerns. The 1-page pdf file also provides useful clarifications about the authors’ work.
> >
> > The remaining point, in my view, is the requirement in assumption 2 which is not related to known notions of nonlinearity. However, authors explain in the rebuttal (and in the appendix) that this assumption somehow relates to previous assumptions in the OCRL literature. They further explain in the pdf that this assumption is satisfied for the models they have used in their experiments. As this is a conference paper, I find these arguments convincing and I’m raising my score.
> >
> > I suggest authors expand further on their examples 2 and 3, and explain the implications of their assumption 2 for neural networks, e.g., which models would not satisfy this requirement, which models would satisfy it, what is the minimal neural network architecture that would satisfy the requirement in assumption 2, etc.

---

> > > ### Author Response · Authors · 2023-08-15
> > >
> > > Thank you for engaging with the rebuttal and adjusting your evaluation.
> > >
> > > With the additional space, we should be able to add details about example 2 and 3, thanks for the suggestion.
> > >
> > > Clarification: Assumption 2 is not about the model used for learning. It is an assumption about the data-generating process. Here, since the dataset is synthetic, we can test the assumption, however in general it might not be possible.

---

### Official Review · Reviewer_C5F5 · 2023-07-03

**Soundness:** 3 good
**Presentation:** 3 good
**Contribution:** 3 good
**Rating:** 7
**Confidence:** 3

**Summary:**

The paper analyses the statistical identifiability of latent variables in an autoencoder with a so-called additive decoder. It is shown that under this class of decoders, the blocks of latent dimensions associated with the additive decoder can be identified. This result is further related to the ability of a model to extrapolate.

**Strengths:**

(These may be subject to change depending on the answers to my questions below)

1. The paper is generally very well written in terms of giving intuitions behind the presented math (see the counter view in Weakness 1 below).
2. The paper touches on an important topic (identifiability), and the focus on additive decoders is both interesting, relevant, and novel.
3. The paper does a nice job of providing proof sketches, which is helpful since space constraints prevent the authors from including actual proofs in the main text.
4. The paper does a very nice job of connecting assumptions and results to existing work in various branches of the literature. This is very helpful.
5. Finally, I want to emphasize that the theoretical findings are both novel and interesting.


**Weaknesses:**

(These may be subject to change depending on the answers to my questions below)

1. The mathematics is often phrased sufficiently convoluted that the phrased intuitions are required (see Strength 1 above). E.g. I found definition 3 to be nearly unreadable, and I could not verify if the clearly phrased intuitions (lines 177-178) actually describe the math (I trust that it does, but it's a problem that it is so difficult to verify).
2. The 'additive decoder' construction seems quite similar to mixture models for which decades of work exist regarding identifiability. I was surprised to not see this link even briefly touched upon.
3. I found the extrapolation part of the paper to be less convincing than the identifiability part. Bluntly put, I got lost in the many assumptions made (Corollary 3 holds under the assumptions of Theorem 2, which hold under the assumptions of Theorem 1 which holds under Assumption 1) that I was unable to tell which were the important assumptions for the particular corollary. Thus, I lost my intuition, and the following discussion (Lines 381-304) seems rather speculative. Fere I struggle to determine what's what.

**Questions:**

1. It seems to me that additive decoders are effectively a form of mixture model with non-trivial components. Here, we know quite a bit about conditions under which components can be identified up to permutation. This seems quite similar to the presented results. Can you elaborate on this connection? Am I misunderstanding something?
2. In definition 2, you write "let $\mathcal{B}$ be a partition..." Should this have been $\mathcal{B}$ be the set of partitions..."? Otherwise, I struggle to understand what $B \in \mathcal{B}$ actually means. But perhaps I misunderstood something.
3. Can we agree that $v := f^{-1} \circ \hat{f}$ is a diffeomorphism simply because assumption 1 states that $f$ is a diffeomorphism or is there more to this? (I ask as the statement about $v$ appears in several places in the paper)
4. How do you ensure that $f$ actually is a diffeomorphism? Here I mainly very about $f$ self-intersecting. I can see how it is easy to ensure that $f$ is an immersion, but in my reading of the paper you seem to require that $f$ is an embedding. Did I get this part right?
5. I did not understand Assumption 2 at all. Can you explain it to me?
6. In Theorem 2 it is assumed that $f$ is injective, which seems like a rather strong assumption. Can this be loosened?
7. In line 271 it is stated $Z$ is *typically* a subset of $CPE(Z)$. What is meant by "typically"?
8. I didn't quite understand the motivation behind the Cartesian-product extension (CPE). According to the intuition of Fig. 3, the CPE makes an axis-aligned extension, but isn't the entire issue regarding identifiability that we cannot assign much meaning to the axes (the axes concern the parametrization and not the underlying support)? Then, why is it natural to extend along the axes? (To be clear, I do see the point of making an extension when studying extrapolation, so my question is more why the said extension should be axis-aligned).

**Limitations:**

There is no need for a discussion regarding societal impact, etc., in a purely theoretical paper such as the present.

I wish the paper had had a greater discussion regarding the many assumptions made throughout the paper. Such a discussion would be akin to a "limitations" section often found in more empirical/methodological papers, and I don't see why a theoretical paper should not openly be discussing the limitations of the analysis (i.e. if the assumptions are appropriate).

---

> ### Author Rebuttal · Authors · 2023-08-09
>
> **Connection to mixture models:**
>
> We assume that by “mixture models” the reviewer refers to models of the form $p(x) = \sum_k p(c=k)p(x | c=k)$ where $c$ is the random component index. After some thinking, we do not see a clear connection between mixture models and our additive decoders. A possible point of confusion is that a mixture model is a sum (more precisely a convex combination) of *distributions* while the additive decoder $x = \sum_k f^{(k)}(z_k)$ is a sum of *random variables*. Both models are very different, since the distribution of a sum of random variables is the *convolution* of their respective distributions (assuming the RVs are independent), not the *convex combination*. Please let us know if the confusion persists or if we misunderstood your point.
>
> **Confusion around assumptions of Corollary 3:**
>
> Essentially, in each result, we always require the assumptions of the previous results. Each result essentially adds more assumptions to obtain stronger guarantees. We’ll add that comment in the next revision.
>
> **Lines 281 to 304 seems speculative:**
>
> Lines 281-288: We believe these to be objective. We describe a procedure we actually implemented in the experiments of Figure 4.
>
> Lines 289-297: See answer to Reviewer 2rwH under *“It appears that authors attempt to define the reasonableness”* for clarifications.
>
> Lines 298-304: We believe it is clear that if a decoder f_hat is disentangled w.r.t. a ground-truth decoder f on the training domain, nothing prevents both decoders to have very different behavior outside the training domain, i.e. no extrapolation, unless these decoders are restricted in some way, either via architectural choice (like in this paper) or via some other means like implicit regularization of the optimizer or loss function.
>
> **Meaning of the term “partition”:**
>
> In mathematics, a “partition of a set A” refers to a set of subsets of A that are mutually disjoint and covers A. This is a very common terminology, see e.g. https://en.wikipedia.org/wiki/Partition_of_a_set. An element of a partition is usually referred to as a “block”. So here, $\mathcal{B}$ is a partition and $B \in \mathcal{B}$ is a block.
>
> **Why is v = f^{-1} \circ f_hat a diffeomorphism?**
>
> We need both f and f_hat to be diffeomorphisms (these assumptions can be found in Assumption 1 and the statement of Theorem 1, respectively).
>
> **How to ensure that f is a diffeomorphism?**
>
> We would like to clarify that f is the ground-truth decoder so we cannot force it to be diffeomorphism, we have no choice but to assume that it is a diffeomorphism. That being said, Theorem 1 requires that f_hat, the learned decoder, is diffeomorphism. In practice, we do not enforce this even if our theory prescribes it. This doesn't seem to be a problem, but might be worth exploring in future works.
>
> **“I did not understand Assumption 2 at all. Can you explain it to me?”**
>
> This is a rather technical assumption which makes it difficult to formulate in words. But the main point is that it prevents f from being linear as Example 2 shows. Keep in mind that this is an assumption about the data-generating process. Relating this assumptions to notions of curvature in differential geometry as proposed by Reviewer 2rwH might help us understand this assumption better, but is left as future work.
>
> **“In Theorem 2 it is assumed that f is injective, which seems like a rather strong assumption. Can this be loosened?”**
>
> Just to clarify: the ground-truth decoder f was assumed to be a diffeomorphism (and hence injective) much earlier in Assumption 1. Loosening this common assumption is an important open question in the field. Part of the difficulty is in defining the relationship between the learned representation and the ground-truth representation which here is given by v. Without injectivity, many z could correspond to a single z_hat.
>
> **“In line 271 it is stated Z is typically a subset of CPE(Z). What is meant by "typically"?”**
>
> Quick correction: we wrote that “Z is typically a *proper* subset of CPE(Z)” (“proper” means Z != CPE(Z)). We simply meant that Z is going to be a subset of CPE(Z) unless Z = CPE(Z). One could argue that, in some sense, there are much more sets Z such that Z != CPE(Z) than there are sets such that Z = CPE(Z). That’s what we meant by “typically”. To avoid confusion, we replaced this sentence by “The Cartesian-product extension of Z, CPE(E), is indeed an extension of Z since Z is a subset of CPE(Z)”.
>
> **Why the extension should be axis-aligned?**
>
> Our identifiability result shows that the additive structure in the decoder induces natural axes (the axes that make the map from z to x additive). Given this, it is expected to have extrapolation that is aligned with these natural axes.

---

> > ### Comment · Reviewer_C5F5 · 2023-08-14
> > **Thank you for the follow-up**
> >
> > Thank you for the follow. Indeed, your analysis was spot on for my confusion regarding links to mixture models. I will increase my score.

---

> > > ### Author Response · Authors · 2023-08-15
> > >
> > > Glad we could resolve this confusion and thanks for adjusting your evaluation.

---

### Official Review · Reviewer_JkNs · 2023-07-07

**Soundness:** 3 good
**Presentation:** 3 good
**Contribution:** 2 fair
**Rating:** 6
**Confidence:** 2

**Summary:**

This paper presents the identification theory for the additive mixing function. Specifically, they transfer the existing nonlinear ICA conditions from the distribution (i.e., sufficient variability) to the nonlinear mixing function (i.e., sufficient nonlinearity). Under this model, they make the connection to extrapolation for generative models. Synthetic data experiments are designed to demonstrate their arguments.

**Strengths:**

1. This paper is well written — both the assumptions and the implications are adequately discussed and thus easy to understand.
2. The block-wise identification result is novel as a result of the sufficient nonlinearity condition inspired by the sufficient variability condition in prior work.
3. The connection to the extrapolation is interesting and yields a valuable understanding of current large models.

**Weaknesses:**

1. Some key assumptions, although discussed, are still evasive in their restrictiveness. The most notable is the sufficient nonlinearity assumption, which appears very restrictive. How to enforce this for the estimation model is challenging.
2. The primary assumption, namely additivity, can be very stringent. It is hard to believe this would hold for any realistic data-generating process. This also somehow trivializes the significance of the extrapolation part. I would also like to learn about the relation to recent work [1].
3. The experimental results lack detailed explanation. I struggle to make sense of the visualizations: what are the colored shades mean in Figure 4, and what does the color mean for the dots? Are the generating processes identical in the additive and the non-additive cases, i.e., does the only difference lie in the estimation model?

[1]. https://arxiv.org/abs/2305.14229

**Questions:**

I would like to learn about the authors' response to the weaknesses listed above, which may give me a clearer perspective on the paper's contribution.

**Limitations:**

Please see the weakness section.

---

> ### Author Rebuttal · Authors · 2023-08-09
>
> **On the restrictiveness of the sufficient nonlinearity assumption:**
>
> At line 218, we refer to Appendix A.5 to provide intuition for why the sufficient nonlinearity assumption is likely to be satisfied when d_x >> d_z. However, we notice that this explanation is lacking from our submission. In the next revision, we will add this explanation to Appendix A.5:
>
> We now look at why Assumption 2 is likely to be satisfied when d_x >> d_z. Informally, one can see that when d_x is much larger than d_z, the matrix W(z) has much more rows than columns and thus it becomes more likely that we will find enough rows that are linearly independent, thus satisfying Assumption 2.
>
> **Regarding how to enforce Assumption 2:**
>
> We note that Theorem 1 and 2 never require to enforce Assumption 2 in the learned model. This is an assumption about the data generating process, not about the learned model, so there is no need to enforce it during training.
>
> **On the restrictiveness of the additivity assumption:**
>
> We agree that the additivity assumption can be rather restrictive and is unlikely to hold for realistic data. Nevertheless we argued in the paper that standard OCRL decoders are “almost” additive and that our analysis shed light on why these methods work on realistic data. Of course, this is only a first step and more work is needed to understand these approaches theoretically.
>
> **Regarding [Brady et al., 2023]:**
>
> This work came out almost exactly as we were submitting this work, and thus we did not include it in our related work section. We will add it to the next revision. Brady et al. studies “compositional decoders” which by definition have Jacobians that, for all z, have always at most one nonzero value per row (note that the nonzero elements can have different locations for different values of z). Example 3 in our paper is an example of function that is additive but not compositional, since the Jacobian has more than one nonzero element per row. Moreover, we suspect, although aren’t fully convinced yet, that compositional functions are additive, since we couldn’t come up with an example of a compositional function that is not additive. If that turns out to be the case, that would mean that additive decoders form a (strict) superset of compositional decoders. We will keep working on this and update you during the discussion period in case we find a definitive answer.
>
> Other important distinctions between our work and theirs is that (i) we consider very general domains for the latent vector z (Brady et al. study only fully supported latent vectors) and (ii) we prove additive decoders can extrapolate (their work has no discussion of extrapolation).
>
> **Some details lacking in experiments:**
>
> Thanks for bringing this to our attention. The different shades of color correspond to the value of one of the ground-truth latent factors. This is useful to assess disentanglement visually. The red dots correspond to the latent factors used to generate the images. In figure 4, the only difference is the estimation model, the data is the same. We’ll make sure to add these clarifications in the next revision.

---

> > ### Comment · Reviewer_JkNs · 2023-08-14
> >
> > I appreciate the detailed response from the authors -- many thanks! I have raised my rating to reflect this.
> >
> > I would be interested in learning from the authors about the relationship between additive and compositional functions, if they happen to have further updates on this.

---

> > > ### Author Response · Authors · 2023-08-15
> > >
> > > Thanks for engaging with our rebuttal!
> > >
> > > Regarding the relationship between additive and compositional decoders, we now almost have a complete proof that compositional implies additive. We are stuck on a small technical detail. In the worst case, adding a mild regularity assumption should do the trick. Essentially, we (almost) showed that a compositional decoder has a Hessian with a single nonzero element that lies on its diagonal. This of course means that they have diagonal Hessians and thus are additive (Appendix A.2 shows that additivity is equivalent to diagonal Hessian). This is interesting has it makes the connection between both function classes very transparent.
> > >
> > > We will make sure to give more updates before the end of the discussion period to confirm (or infirm) everything.

---

> > > > ### Author Response · Authors · 2023-08-17
> > > >
> > > > We now have a complete proof that **($C^2$) compositional decoders are additive**. This implies that the class of additive decoders is *strictly* more expressive than the class of $C^2$ compositional decoder ([Brady et al., 2023] assumes only $C^1$, see below for more on this). We provide a partial proof here. We’ll be glad to provide further details on request.
> > > >
> > > > We give a definition of compositional decoder adapted from Brady et al..
> > > >
> > > > **Def:** Given a partition $\mathcal{B}$,  a function $f$ is compositional w.r.t. $\mathcal{B}$ when, for all $i \in [d_z], z \in \mathbb{R}^{d_z}, B \in \mathcal{B}$, we have that $D_B f_i(z) \not= 0 \implies D_{B^c} f_i(z) = 0$, where $B^c = [d_z] \setminus B$.
> > > >
> > > > **Proof sketch:** Our strategy is to show that the Hessian of $f_i$ is block diagonal everywhere on $\mathbb{R}^{d_z}$ and then use Proposition 5 from Appendix A.2 to conclude that $f_i$ must be additive.
> > > >
> > > > For each $z_0 \in \mathbb{R}^{d_z}$, we know there exists a $B \in \mathcal{B}$ such that $D_{B^c}f_i(z_0) = 0$.
> > > >
> > > > In the case where $D_Bf_i(z_0) \not= 0$, we have by continuity of $Df_i$ that there exists an open neighborhood of $z_0$ on which $D_Bf_i(z) \not= 0$. By compositionality, we must also have that $D_{B^c}f_i(z) = 0$ on that neighborhood. This means that the derivative of $D_{B^c}f_i$ at $z_0$ is zero, i.e. $DD_{B^c}f_i(z_0) = 0$. Since $f$ is $C^2$, its Hessian is symmetric and thus $D_{B^c}Df_i(z_0) = 0$. We can thus conclude that $D^2f_i(z_0)$ is filled with zeros except possibly at the entries $B\times B$. Hence it is block diagonal.
> > > >
> > > > In the case where $D_Bf_i(z_0) = 0$, the argument is slightly more involved because we cannot necessarily take derivatives because $(D_Bf_i)^{-1}(\\{0\\})$ is a closed set (by continuity of $D_Bf_i$) and thus $z_0$ might be on its boundary. If $z_0$ is in the interior of $(D_Bf_i)^{-1}(\\{0\\})$, we can use an argument similar to above to show that $D^2 f_i(z_0) = 0$. If $z_0$ is on the boundary, by using the continuity of $D^2f_i$, we can show the Hessian is also going to be block-diagonal. (We made this last part of the argument more precise in our revision. We can provide these additional details on request.) $\blacksquare$
> > > >
> > > > **We would like to reiterate the differences between both works:**
> > > >
> > > > - We consider additive decoders which, as we just showed, are strictly more expressive than $C^2$ compositional decoders introduced in [Brady et al., 2023].
> > > > - We assume the decoder is $C^2$ whereas [Brady et al., 2023] assumes only C^1 (which is weaker).
> > > > - We consider very general domains for the latent vector z (Brady et al. study only fully supported latent vectors)
> > > > - We prove additive decoders can extrapolate (their work has no discussion of extrapolation).
> > > >
> > > > Note that we cannot say that our identifiability result is stronger than theirs because we assume $C^2$ decoders while they assume $C^1$. This also makes the comparison between our “sufficient nonlinear” assumption (which refers to second derivatives) and their “irreducibility assumption” (which we believe to be somewhat analogous) difficult.
> > > >
> > > > We'd like to correct a mistake we made in a previous message: We said that "compositional" means that the Jacobian cannot have more than one nonzero entry per row. This is true only in the special case where $\mathcal{B} := \{\{1\}, ..., \{d_z\}\}, but the Brady et al. allowed for more general partitions. The definition we gave above is correct.
> > > >
> > > > Feel free to ask if you have any questions, we'll be happy to clarify.

---

### Official Review · Reviewer_CUR1 · 2023-07-11

**Soundness:** 3 good
**Presentation:** 4 excellent
**Contribution:** 3 good
**Rating:** 7
**Confidence:** 3

**Summary:**

This paper extends the recently popular approach of constraining the nonlinear function to achieve identifiable disentanglement. Here the idea is that $\mathbf{f}$ is additive i.e. made of constituent functions that operate independently on non-overlapping partitions of the latent function. Identifiable disentanglement is achieved in this situation with very mild conditions on the latent distribution. These additive decoders can be seen as rudimentary version of the decoders used in object-centric representation learning and thus help explain their generalization performance. In particular, the paper shows that by exploring the full cartesian product of the latent symbols the model can generate images that are out of the training images' support (called cartesian-product extrapolation).

**Strengths:**

- very well written paper with clear and intuitive examples despite the technical topic
- a novel way in which we can understand disentanglement by restricting the nonlinearity from the point of view of OCRL is a nice new angle to this increasingly popular field
- by making assumption about the structure of $f$ the authors are able make very mild assumptions about the distribution of the latent factors which is in contrast to the much more distribution
- additive decoders and the relevant results here provide a nice simple baseline model upon which future works can build more realistic ocrl models
- extrapolation guarantees is a nice addition, something previous works have been missing, and is something that hopefully will be adopted by the community (though it's unclear how that could be done; see below)
- assumption of sufficient nonlinearity is an insightful result and its connection to previous literature is nicely illustrated (albeit in the appendix)

**Weaknesses:**

Since the latent variables have only very mild restrictions, the price is paid by having fairly strong restriction on the 'mixing' function i.e. block-wise additivity (likely problematic in many realistic situations such as images with occlusion) + requirements on nonlinearity. This is likely useful for OCRL (e.g. scene mixtures) but in general may be very restrictive in the more general nonlinear ICA/disentanglement and also abstracts away from the desired goal of traditional nonlinear ICA where the aim is to separate out sources from 'heavily mixed' signals.

Furthermore, the additivity also leads to a slightly problematic level identifiability results in the sense that each partition-block $z_b$ is identified only up to arbitrary invertible nonlinear transformation (equation 8. & 9. and $v_B$ specifically) -- this can be interpreted as there being a nonlinear ICA / mixing problem completely *unsolved* and thus unidentified for each block. So even though we are no longer left with the generic unidentifiability problem of $x = f(z)$, we are still left with complete unidentifiability in each block $B$, $ x_B = f_B (z_B)$. It feels like the term "partition-respecting permutation" and "B-disentanglement" are thus very specific to this approach and not really corresponding to the commonly accepted definitions of disentanglement in literature. Indeed the authors write "Thus, B-disentanglement means that the blocks of latent dimensions zB are disentangled from one 177 another, but that variables within a given block might remain entangled."

Related to this, for OCRL the model is quite simple, as admitted by the authors, and while they indeed may provide a good baseline (as mentioned above) it is hard to be sure whether the theorems fully explain their performance especially given these concerns about block-wise unidentifiability.

It would have been nice to see more extensive experiments and especially on more complex data.

**Questions:**

Do you believe the block-wise unidentifiability/entanglement is undesirable? Or do you believe my concerns above are not a problem? I'm talking practically speaking. What do you think is the impact of this on CPE? What if the blocks could also be fully disentangled -- what would this change (e.g. in CPE?). Again I'm interested in your thoughts on practical implications rather than theory.

You say that: "is “imitating” a block-specific ground-truth decoder". Could you please explain what you exactly mean by imitation? I find that quite vague.

You write that "We believe it illustrates the  fact that disentanglement alone is not sufficient to enable extrapolation and that one needs to restrict the hypothesis class of decoders in some way." These sound like very generic statements -- do you have confidence that they hold much beyond this setting?

"our in this work to understand “out-of-support” generation is a step towards understanding theoretically why modern generative models such as DALLE-2 [42] and 53 Stable Diffusion [43] can be creative" Could you please explain this more specifically? Do you believe additiveness to be important to this?

**Limitations:**

There is some good discussion of limitations in the paper e.g. "additive decoders make intuitive sense for OCRL, they are not expressive enough to represent the “masked decoders” typically used in", "Additionally, this  parameter sharing across f (B) enables modern methods to have a variable number of objects across  samples, an important practical point our theory does not cover." Appendix also has nice examples of what happens when some assumptions are violated.

---

> ### Author Rebuttal · Authors · 2023-08-09
>
> **Strength of the assumptions**
>
> We agree that the additive assumption is rather strong and won’t make sense in many applications other than OCRL. That being said, we believe our work is a nice starting point for other works to investigate more expressive function classes.
>
> **Issues raised by block-disentanglement**
>
> We believe that extending existing proof techniques to block identifiability is actually a strength, since it is more general and it includes the trivial partition {{1}, {2}, ..., {d_z}}, which would yield full disentanglement. The level of granularity of the partition controls the trade-off between expressivity and identifiability: The finer the partition, the more constrained is the function class and the tighter the identifiability guarantee. The optimal level of granularity is going to be task-dependent. Regarding the impact on the CPE, we have that the finer the partition is, the larger the CPE is going to be. To see this, consider the case where the Z^train is a sphere in 3D (so that d_z = 3). If B = {{1}, {2}, {3}}, we have that CPE is the smallest cube containing the sphere. If B = {{1,2}, {3}}, the CPE is a cylinder, which is a proper subset of the cube. We plan on adding a discussion on these considerations in the next revision.
>
> **“it is hard to be sure whether the theorems fully explain their performance especially given these concerns about block-wise unidentifiability.”**
>
> Note that, in our experiments, we always test for B-disentanglement. For the ScalarLatents dataset, B = {{1}, {2}} whereas for the BlockLatents datasets, B = {{1,2}, {3,4}}. In the first case, B-disentanglement is simply “full/complete disentanglement”, but for the last case, our evaluation does not penalize entanglement within a block. At line 328, we provide details on our evaluation metrics.
>
> **“Could you please explain what you exactly mean by imitation?”**
>
> We use “imitation” for lack of a better word at line 244 and 281. This terminology is borrowed from [Ahuja et al., 2021] which was used in a similar way for transition mechanisms. For example, in equation (8) which relates the f_hat^B with f^{\pi(B)}, the two functions are not equal, but they are closely related to one another via the map v_bar and the additive constant c, so one can think about f_hat^B as imitating f^{\pi(B)}. We provide a more explicit explanation at lines 245-248.
>
> [Ahuja et al., 2021] https://arxiv.org/abs/2110.15796
>
> **Regarding necessity of restricting decoder in order to enable extrapolation:**
>
> Without a restriction on the decoder class, there are no ways to guarantee what will happen outside the domain of the training set. And disentanglement is not enough since you might be disentangled on the training domain but crazy things might happen outside. Note that this restriction might be enforced via the choice of architecture (like we did in the present work) or by some implicit bias of the optimizer or the loss function etc… We use the phrase “restricted decoder” in its broadest sense here.

---

> > ### Comment · Reviewer_CUR1 · 2023-08-15
> >
> > Thanks for the comments -- I am mostly happen with them except below:
> >
> > **We believe that extending existing proof techniques to block identifiability is actually a strength, since it is more general and it includes the trivial partition {{1}, {2}, ..., {d_z}}, which would yield full disentanglement.**
> >
> > I am not convinced by this argument. If you have have partition {{1}, {2}, ..., {d_z}} then each $f^{(b)}(z_b)$ only takes as an input a single  random variable, no? so there is complete lack of nonlinear mixing (only a linear mixture is disentangled to find the individual components). As soon as the partitions are larger than size 1, the block-specific nonlinear mixture is immediately unidentifiable. Therefore, you are not solving nonlinear ICA, you are only disentangling the partitions from each other -- there representations you learn for each block may be completely arbitrary (bijective) transformations of the ground-truth representations. Is this correct?
> > ***
> > Figure 4 -- as someone who has red-green color blindness, the red square of extrapolations in Figure 4 is almost invisible (i only saw it once zooming in full screen). The colors in general are poorly chosen -- there exist several colour palettes that take these issues into account.
> > ***
> >
> > p.s. dodgy grammar in l.286 "i.e. it is the observation one would have obtain by evaluating"

---

> > > ### Author Response · Authors · 2023-08-15
> > >
> > > Thanks for seriously engaging with our rebuttal, we appreciate it. Below we address your concerns. We apologize for the lengthy answer, but we felt some point required careful explanations.
> > >
> > > **"If you have have partition {{1}, {2}, ..., {d_z}} then each $f^{(b)}(z_b)$ only takes as an input a single random variable, no?"**
> > >
> > > That is correct.
> > >
> > > **"so there is complete lack of nonlinear mixing (only a linear mixture is disentangled to find the individual components)"**
> > >
> > > It depends on what is meant by "complete lack of nonlinear mixing". The resulting data-manifold can still be highly nonlinear. However, it is true that the nature of the mixing between components is limited by the additivity. I believe the point you are making here reduces to the point you initially made that "additivity is restrictive". We agree with this. Many works have considered restricted function classes to improve identifiability like [Brady et al., 2023], [Buchholz et al., 2022], [Gresele et al., 2021]  and [Taleb & Jutten, 1999]. It is clear that these works as well as ours do not form a complete solution to the nonlinear ICA problem, which, in his original formulation, is known to be unsolvable [Hyvärinen & Pajunen, 1999]. That being said, some function classes will be useful for some applications, but not all. In this work we argued that additivity makes intuitive sense for object-centric representation learning (with caveats).
> > >
> > > **"As soon as the partitions are larger than size 1, the block-specific nonlinear mixture is immediately unidentifiable."**
> > >
> > > That is correct. The blocks of the partition will be disentangled from one another, but the variables within a given block can remain entangled.
> > >
> > > **"Therefore, you are not solving nonlinear ICA, you are only disentangling the partitions from each other"**
> > >
> > > The term "nonlinear ICA" sometimes mean different things in different context. Some people use it to refer to the original problem where the decoder is a general invertible map and the latents are independent [Taleb & Jutten, 1999] (which is unidentifiable). Some will use it to mean any setting where the mixing function is a general invertible map but will allow for richer latent distribution like with auxiliary variables [Khemakhem et al., 2020] or temporal dependencies [Lachapelle et al., 2022] for example. In this work, we use the term "nonlinear ICA" to mean any problem where the goal is to recover latent variables from some nonlinear mixture (which might be restricted). Of course, under this definition, additive decoders count as "nonlinear ICA" (since additive functions are nonlinear in general). Note that, although our identifiability result restricts the mixing function, we allow for much more general distribution over the latents (dependencies + general support shape) than the strictest interpretation of "nonlinear ICA" which assumes independent latents.
> > >
> > > **"there representations you learn for each block may be completely arbitrary (bijective) transformations of the ground-truth representations. Is this correct?"**
> > >
> > > Indeed, the block $\hat{z}\_B$ of a learned representation can be an arbitrary nonlinear transformation of some block of the ground-truth $z_{B'}$.
> > >
> > > **Poor choice of color in Figure 4**
> > >
> > > Sincerely sorry for the inconvenience, we'll make sure to fix this in the camera-ready version. You said that in general the colors are poorly chosen, are there any other specific places that caused trouble?
> > >
> > > [Brady et al., 2023] https://arxiv.org/abs/2305.14229
> > >
> > > [Buchholz et al., 2022] https://arxiv.org/abs/2208.06406
> > >
> > > [Gresele et al., 2021] https://arxiv.org/abs/2106.05200
> > >
> > > [Taleb & Jutten, 1999] https://ieeexplore.ieee.org/document/790661
> > >
> > > [Hyvärinen & Pajunen, 1999] https://www.cs.helsinki.fi/u/ahyvarin/papers/NN99.pdf
> > >
> > > [Khemakhem et al., 2020] https://arxiv.org/abs/1907.04809
> > >
> > > [Lachapelle et al., 2022] https://arxiv.org/abs/2107.10098

---

> > > > ### Comment · Reviewer_CUR1 · 2023-08-15
> > > >
> > > > I still find this problematic. With regards to above discussion about nonlinear ICA, and lack of identifiability of nonlinear mixtures, I can see where you are coming from, but still hold my opinion that this paper is not really doing nonlinear ICA. You say that:
> > > >
> > > > *"The term "nonlinear ICA" sometimes mean different things in different context.  [...] Some will use it to mean any setting where the mixing function is a general invertible map but will allow for richer latent distribution like with auxiliary variables [Khemakhem et al., 2020] or temporal dependencies [Lachapelle et al., 2022] for example"*
> > > >
> > > > Perhaps, but using the term ICA to refer to situations where the latent variables are not independent, logically does not really make sense and is poor usage, in my opinion, given what the abbreviation stands for. In Khemakhem (iVAE paper) the latent components are conditionally independent so I feel it's justified there. As a case in point, Khemakhem et al. also have works where variables are *not* independent -- in his phd thesis he calls those works 'identifiable representation learning', which I find much more appropriate. Similarly the prefix 'nonlinear' is also not really justified when one is **not** able to recover nonlinearly mixed components, but can merely recover the partitions (symbols/objects).
> > > >
> > > > **To accept this paper, I would require this to be discussed more openly** Currently you write: l.177  "Thus, B-disentanglement means that the blocks of latent dimensions zB are disentangled from one  another, but that variables within a given block might remain entangled." I don't think this is enough because a careless / casual reader might not easily realize how different this is from the typical nonlinear ICA. I think it would suffice however if you had a sentence or two along the lines "note that the B-disentanglement result is different from those in nonlinear ICA in that we do not recover each latent component, but rather disentangle the partitions, that is, variables within a given block might remain entangled". Key is to stress the difference to nonlinear ICA which is missing now.
> > > >
> > > > *Sincerely sorry for the inconvenience, we'll make sure to fix this in the camera-ready version. You said that in general the colors are poorly chosen, are there any other specific places that caused trouble?*
> > > > I just meant Figure 4 in general -- it is also hard for me to tell apart the two balls as they are red and greenish or something like that. Figure 5 is fine. I appreciate that you can not see the Figures through other peoples eyes so it can always be hard to find colors that suit everyone!

---

> > > > > ### Author Response · Authors · 2023-08-17
> > > > >
> > > > > Thanks for engaging with us and insisting on that point! After some thinking and discussion, we’ve come to agree with you that our usage of the term “ICA'' was too broad and that “ICA” should be reserved for methods where some form of statistical independence between the components is assumed for identification. Of course, under this definition, our approach does not qualify as ICA. To address this issue, we propose the following modifications:
> > > > >
> > > > > 1- Regarding your request to contrast more transparently with nonlinear ICA, we suggest adding this paragraph at L87 in the “Background & literature review”:
> > > > >
> > > > > **Relation to nonlinear ICA.** [Hyvärinen & Pajunen, 1999] showed that the standard nonlinear ICA problem where the observation $x$ is given by a general nonlinear transformation of *statistically independent latent factors* $z_i$ is unidentifiable. This motivated various extensions of nonlinear ICA where more structure on the factors is assumed [CITE nonlinear ICA works]. Our approach departs from the standard nonlinear ICA problem along three axes: (i) we restrict the mixing function to be additive, (ii) the factors do not have to be necessarily independent, and (iii) we can identify only the blocks $z_B$ as opposed to each $z_i$ individually up to element-wise transformations, unless $\mathcal{B} = \\{\\{1\\}, ...,\\{d_z\\}\\}$ (see Section 3.1).
> > > > >
> > > > > 2- We will also add the following clarification right after L178 in Section 3.1:
> > > > >
> > > > > “Note that, unless the partition is $\mathcal{B} = \\{\\{1\\}, …, \\{d_z\\}\\}$, this corresponds to a weaker form of disentanglement than what is typically seeked in nonlinear ICA, i.e. recovering each variable individually.”
> > > > >
> > > > > 3- We suggest replacing the following problematic sentence from the abstract:
> > > > >
> > > > > L10: “Our result provides a new setting where nonlinear independent component analysis (ICA) is possible and adds to our theoretical understanding of OCRL methods.”
> > > > >
> > > > > by
> > > > >
> > > > > “Our result adds to our theoretical understanding of OCRL methods and provides a new variation of nonlinear independent component analysis (ICA) where latent factors can be identified.”
> > > > >
> > > > > (The rationale behind keeping the keyword “ICA” in the abstract is that we believe this result will be of interest to this community.)
> > > > >
> > > > > Please let us know whether you find these modifications to be satisfactory or not.
> > > > >
> > > > > Note: We would like to clarify your point that “Similarly the prefix 'nonlinear' is also not really justified when one is not able to recover nonlinearly mixed components, but can merely recover the partitions (symbols/objects)”.  Strictly speaking, a function of the form $\sum_B f^{(B)}(z_B)$ can be nonlinear in $z$, even when the partition is trivial. That being said, we agree that a case can be made that the *mixing* itself is linear in the following sense: Additive decoders with the trivial partition can be written as $f(z) = S(F(z))$ where $F(z) = [f^{(1)}(z_1), …, f^{(d_z)}(z_{d_z})] \in \mathbb{R}^{d_x \times d_z}$ and $S: \mathbb{R}^{d_x \times d_z} \rightarrow \mathbb{R}^{d_x}$ is the linear operator consisting of summing the columns of $F(z)$. Of course, $F(z)$ can be nonlinear, but it does not mix the latent factors. The mixing occurs only in $S$, which is a linear operator. So although the decoder function $f(z) = S(F(z))$ is indeed nonlinear, the *mixing step* is linear.

---

### Official Review · Reviewer_7aiT · 2023-07-24

**Soundness:** 3 good
**Presentation:** 3 good
**Contribution:** 3 good
**Rating:** 7
**Confidence:** 3

**Summary:**

Motivated by the problem of disentanglement in generative models, this paper proposes a novel decoder architecture, so-called additive decoders, based on the addition of block-wise latent variables, where blocks of latents correspond to semantic factors.

An extrapolation property of their decoder is demonstrated, which essentially consists of forming new products of latent blocks unseen at train time.

A theoretical analysis of their decoder is carried out. Several new definitions are made, as well as few necessary assumptions. Two theorems are presented on "local disentanglement" and "global disentanglement".

An empirical investigation based on synthetic image data, consisting of two balls randomly placed, is carried out, showing a case where their decoder improves upon a baseline case.


**Strengths:**

The paper addresses the problem of disentanglement in an interesting way: through blocks of latent variables which contribute additively to the overall decoding. This is a natural and reasonable idea.

The motivation for this paper is good: restrict the decoder to be additive to address the problem of latent variable identifiability.

The proof technique appears to build on the well-established results (Hyvärinen, AISTATS 2019).


**Weaknesses:**

The paper is quite technical. This makes it challenging to ensure all technical details are correct. I did not find any errors.

The authors write that this work may help explain the "creativity" of mainstream generative models like DALLE-2 and Stale Diffusion. It is not so clear to me that their analysis will be helpful.

Regarding Assumption 2, which the authors write is "key" for Theorem 2, I am not clear on how realistic this assumption is in practice. I understand that it is motivated by similar assumptions made in the ICA literature. However I come away with the feeling that it may not have much practical relevance. The authors give a toy numerical example in Example 3, which is helpful. But for instance, has this assumption been verified for the additive decoders used in the Experiments (Section 4)?

The proposed additive decoder cannot handle occlusions, as noted by the authors in Section 4, and discussed in the appendix. This is a limitation, since real images may have occlusions.

Only small-scale synthetic data are included. It doesn't appear to me that training on larger scale data is feasible, but more discussion of this would be helpful. It's reasonable to develop theory on simple cases, perhaps even necessary in this cases, but we should be clear whether scaling the results can be expected.


**Questions:**

Can the authors further justify their statement that this work "has the potential of expanding our creativity in generative model" (line 368).

Can assumption 2 be verified for the decoders in used in the Experiments (Section 4)?

Is training on larger-scale data possible with this method? If not, why not?


**Limitations:**

Unfortunately I did not find any discussion of limitations in the main text.

I don't think this paper requires a discussion on societal impact.

---

> ### Author Rebuttal · Authors · 2023-08-09
>
> **Practical relevance of Assumption 2**
>
> Although we haven’t shown that Assumption 2 is necessary in the sense that it cannot be relaxed, we know that we cannot simply remove it and get the same guarantees since this would contradict the very well established non identifiability of linear ICA when latents are independent and non-gaussian. That’s because, without Assumption 2, the decoder would be allowed to be linear (See Example 2). So this assumption has at least some practical relevance.
>
> **Verifying Assumption 2 in the “ball dataset” (new experiments!)**
>
> This is a very good suggestion which we decided to investigate. The image generator we are using to generate these images takes only integer values which, strictly speaking, prevents us from even talking about the derivative of this generator. That being said, we created a very similar dataset which takes continuous values as input and can thus be differentiated (in fact, this generator is infinitely differentiable). This allowed us to verify numerically that Assumption 2 is satisfied. To make this verification, we considered a grid of values of z and for each point on that grid, we computed the matrix W(z) from Assumption 2, normalized its columns to have unit length, and computed $\sqrt{|det(W^t W)|}$ which gives the 4D volume of the parallelogram spanned by the 4 columns of W. This quantity should be 1 when columns are orthogonal (i.e. maximally linearly independent) and 0 when the columns are linearly dependent. We found the minimal value of $\sqrt{|det(W^t W)|}$ to be ~0.97, which indicates that the columns of W are linearly independent. **Please check the pdf linked in the general response for a visualization of the new dataset as well as the contour plot of sqrt(|det(W^t W)|) as a function of z. The pdf also contains an experiment in which we trained both additive and non-additive decoders. It leads to very similar conclusion to what we found in the paper.**

---

> > ### Comment · Reviewer_7aiT · 2023-08-14
> >
> > I found the author's rebuttal convincing. The new numerical experiment on explicitly verifying Assumption 2 is helpful. At least for toy data with well-conditioned matrices, this assumption can be verified in practice. I will raise my score.

---

> > > ### Author Response · Authors · 2023-08-15
> > >
> > > Thanks for engaging with our work and adjusting your evaluation.

---

### Author Rebuttal · Authors · 2023-08-09

We would like to thank all reviewers for engaging with our work and asking important and thought-provoking questions.

We were glad to see that many reviewers appreciated the clarity, relevance and novelty of our work. For example, **Reviewer C5F5** said that “The paper touches on an important topic (identifiability), and the focus on additive decoders is both interesting, relevant, and novel.” We were happy to read that **Reviewer CUR1** thought that “extrapolation guarantees is a nice addition, something previous works have been missing, and is something that hopefully will be adopted by the community”. **Reviewer 2rwH** said that the paper had a “broad and practical view” while **Reviewer 7aiT** thought the paper was well motivated. **Reviewer JkNs** wrote “The paper is generally very well written in terms of giving intuitions behind the presented math”. **Reviewer CUR1** thought that our result was “insightful” and that “its connection to previous literature is nicely illustrated”.

Some concerns were raised by more than one reviewer, so we decided to address them here in this general response. The other concerns are addressed in individual responses. **Please not that we added additional data and experiments in response to Reviewer 7aiT (see PDF).**

**Absence of large-scale/more complex datasets (7aiT, CUR1, 2rwH)**

We would like to emphasize that the bulk of our contribution is not to propose a novel architecture with the goal to compare to SOTA methods in object-centric representation learning. Instead, we bring theoretical insights as to why current OCRL methods work in the first place (with the caveats that these OCRL decoders are not exactly additive, but close to it, as discussed in the paper). We note that Peebles et al. [41] already showed convincingly that the diagonal Hessian penalty (which is equivalent to additivity, as we showed in Appendix A.2) improves disentanglement on more realistic datasets, although without any theory for why this worked. Our work can be seen as filling this gap. In addition, our identifiability result is interesting in and of itself to the nonlinear ICA community since it shows yet another case where nonlinear ICA is possible.

[41] W. Peebles, J. Peebles, J.-Y. Zhu, A. A. Efros, and A. Torralba. The hessian penalty: A weak prior for unsupervised disentanglement. In ECCV, 2020.


**Lack of discussion surrounding limitations. (7aiT, C5F5)**

We would like to point out that this is in opposition with Reviewer CUR1’s comment which pointed out to discussions of limitations in our work. For example, the paragraph on line 156 titled “Differences with OCRL in practice” discusses some limitations of our analysis like the fact that the standard masked decoder used in practice is not additive and the fact that we do not consider variable numbers of objects across images, like most OCRL methods can do. Also, on line 325, we refer to Appendix A.11 where we explain why additive decoders cannot represent occlusion. On line 357, we discuss the importance of the “connected support” assumption.

If the reviewers have specific ideas of limitations to discuss, we would be happy to include them in our next revision.

**How is this analysis going to help us understand creativity in more modern generative model? (7aiT, CUR1)**

Thanks for raising this. We agree we did not develop this point sufficiently in the paper, so we add clarifications here and plan on adjusting the next revision accordingly. We believe our analysis illustrates how studying the identifiability of a function class can help us understand how it will extrapolate. We believe this type of analysis is novel and hope that it will motivate other researchers to apply similar analyses to other function classes/learning algorithms (e.g. DALLE-2) to understand and maybe improve extrapolation capabilities. Whether additivity is going to be central to these analyses is left to be seen. It might not be and this does not matter since our point that “identifiability is an interesting lens to study extrapolation” remains.

---

### Decision · Program_Chairs · 2023-09-21

**Decision:**

Accept (oral)

**Comment:**

Motivated by the problem of disentanglement in generative models,
This paper proposes a novel decoder architecture for latent variables identification and "out-of-support" image generation in representation learning. The model is based on the addition of block-wise latent variables, where blocks of latents correspond to semantic factors.
Identifiable disentanglement is achieved with very mild conditions on the latent distribution. Additionally, the paper shows that the model can generate images that are out of the training images' support (called cartesian-product extrapolation).

I agree with the reviewers that the paper addresses an important problem in representation learning, which is learning disantangled representations, and that the proposed approach can have a significant impact in the community.